# High-Dimensional Analysis of Single-Layer Attention for Sparse-Token Classification

**Nicholas Barnfield**
Department of Statistics
Harvard University
Cambridge, MA, USA
`nbarnfield@g.harvard.edu`

**Hugo Cui**
Center of Mathematical Sciences and Applications
Harvard University
Cambridge, MA, USA
`hugo@cmsa.fas.harvard.edu`

**Yue M. Lu**
John A. Paulson School of Engineering and Applied Sciences
Harvard University
Cambridge, MA, USA
`yuelu@seas.harvard.edu`

## Abstract

When and how can an attention mechanism learn to selectively attend to informative tokens, thereby enabling detection of weak, rare, and sparsely located features? We address these questions theoretically in a sparse-token classification model in which positive samples embed a weak signal vector in a randomly chosen subset of tokens, whereas negative samples are pure noise. For a simple single-layer attention classifier, we show that in the long-sequence limit it can, in principle, achieve vanishing test error when the signal strength grows only logarithmically in the sequence length $L$, whereas linear classifiers require $\sqrt{L}$ scaling. Moving from representational power to learnability, we study training at finite $L$ in a high-dimensional regime, where sample size and embedding dimension grow proportionally. We prove that just two gradient updates suffice for the query weight vector of the attention classifier to acquire a nontrivial alignment with the hidden signal, inducing an attention map that selectively amplifies informative tokens. We further derive an exact asymptotic expression for the test error of the trained attention-based classifier, and quantify its capacity—the largest dataset size that is typically perfectly separable—thereby explaining the advantage of adaptive token selection over nonadaptive linear baselines.

Attention-based architectures (Vaswani et al., 2017) have proven in recent years to be a major driver of progress in a wide spectrum of learning tasks, ranging from language processing (Kenton & Toutanova, 2019; Brown et al., 2020) to computer vision (Dosovitskiy et al., 2021). A core strength of these models is the ability of attention layers to dynamically weigh the importance of different input tokens, enabling the model to selectively focus on the most relevant information. This flexibility makes transformers particularly effective at capturing subtle patterns and features within complex, high-dimensional data, even when such information is dispersed throughout the input sequence.

Despite the ubiquity of attention-based models in contemporary deep learning practice, a rigorous theoretical understanding of their working mechanism is still in its early stages. A large body of theoretical works has focused on understanding the benefits of attention in simple solvable models, e.g. (Geshkovski et al., 2023; Ahn et al., 2023; Von Oswald et al., 2023; Edelman et al., 2022; Hahn, 2020; Bordelon et al., 2024; Bietti et al., 2023; Maulen-Soto et al., 2025), with particular focus

devoted to single-layer architectures. Recently, a line of studies has demonstrated the advantages of attention-based architectures for *sparse* token regression tasks—settings where labels depend only on a small subset of input tokens (Oymak et al., 2023; Marion et al., 2024; Sanford et al., 2023; Wang et al., 2024; Mousavi-Hosseini et al., 2025; Zhang et al., 2025; Ren et al., 2024). In such tasks, attention mechanisms dynamically identify and prioritize the relevant tokens, significantly enhancing learning efficiency. In contrast, fully-connected architectures require exponentially more samples (Mousavi-Hosseini et al., 2025) or neurons (Sanford et al., 2023) as the input sequence length grows.

In many applications, however, the *sparsity* of informative features is frequently compounded by additional challenges, notably the *weakness* and *rarity* of the underlying signals. For example, cancer diagnosis from computed tomography scans involves detecting lesions—features that are typically subtle (weakness), appear in varying locations (sparsity), and occur infrequently (rarity). All these characteristics significantly complicate the detection problem. Motivated by scenarios of this kind, we examine a statistical classification problem in which positive samples contain weak signals embedded within a small, randomly selected subset of tokens. We analyze the capacity of a single attention layer to learn to adaptively identify and enhance these sparse, weak, and potentially rare signals. Specifically, our **main contributions** are as follows:

- In the limit of large sequence length $L$, we show that an attention model can detect signals that are exponentially weaker in $L$ than those detectable by non-adaptive linear classifiers.

- Moving from representational power to learnability, we study training at finite $L$ and derive an exact characterization—down to explicit constants—of the test error for the attention model after two gradient updates, followed by full optimization of the last-layer weights, in the limit of high-dimensional token embeddings with proportionally large sample size. These sharp asymptotic results quantify precisely how the test error depends on the number of samples, the sequence length, and the signal strength.

- Our analysis demonstrates that merely two gradient steps suffice for the attention model to develop meaningful internal representations. Consequently, the classifier can dynamically identify and selectively focus on the relevant subset of signal-bearing tokens—effectively amplifying the signal-to-noise ratio—and outperform linear classifiers.

- To provide a complementary perspective on the advantage of attention, we characterize the capacity of the attention model, defined as the maximal dataset size that can be perfectly fit with high probability, and compare it with the corresponding capacity of linear classifiers.

### RELATED WORKS

**Theoretical analysis of transformers and attention models.** The expressivity of attention-based architectures has been extensively studied in recent literature. Fu et al. (2024) established that a single multi-head attention layer with fixed weights can represent a broad class of permutation-invariant functions. Edelman et al. (2022) observed that the statistical capacity of bounded-norm attention models scales only logarithmically with sequence length, suggesting a strong inductive bias toward sparse functions dependent on only a subset of input tokens.

**Sparse token regression/classification tasks.** A special class of sparse functions is studied in further detail by Sanford et al. (2023), who consider a sequence-to-sequence task on length $L$ sequences, where outputs correspond to the average of a dynamically selected subset of $R < L$ tokens. Whereas fully-connected architectures require $\Omega(L)$ hidden units to represent such functions, attention models only need $\Omega(R)$ and can provably learn the task via gradient-based training on the population risk (Wang et al., 2024). Complementing these findings, Mousavi-Hosseini et al. (2025) establish corresponding results demonstrating significant separations in terms of sample complexity. Similarly, Marion et al. (2024); Duranthon et al. (2025) prove that a softmax attention layer can learn a single-token regression up to Bayes-optimal error, whereas linear attention fails, and linear regression on flattened samples performs poorly due to its inability to adapt to dynamic sparsity. Additionally, recent work by Zhang et al. (2025) analyzes a sparse classification task where the relevant token locations are fixed across samples. Closer to our work, Oymak et al. (2023) study a related classification task with the same model as the one considered here, and prove that it reaches a good accuracy after three steps of gradient descent, outperforming linear regression with average-pooling.

Our current work builds upon and significantly extends this line of research along multiple fronts. On the technical level, we crucially extend the analysis of sparse token tasks to *arbitrary* convex losses beyond the square loss which is considered in prior works (Marion et al., 2024; Mousavi-Hosseini et al., 2025; Wang et al., 2024; Oymak et al., 2023). Our extension importantly includes classical loss functions such as the logistic loss, of particular relevance for classification tasks. Furthermore, while most theoretical works have focused on studying the challenges posed by signal sparsity, we further address the often concurrent hurdles of signal rarity and weakness. We demonstrate that attention mechanisms can adaptively address all three challenges by dynamically selecting informative tokens and amplifying their signals. In these respects, our manuscript provides a fully rigorous and encompassing analysis of empirical risk minimization in a classification setting.

# 1 PROBLEM SETUP

**Sparse token classification.** We consider a binary classification task on $L \times d$ covariates, seen as sequences of $L$ tokens embedded in $d$ dimensions. Positive samples contain a weak signal added to a random subset of tokens; negative samples do not display the signal. The learning task consists of discriminating samples with the signal from those devoid thereof. In a similar spirit to the sparse-token regression/classification problems studied in (Sanford et al., 2023; Oymak et al., 2023; Wang et al., 2024; Marion et al., 2024; Mousavi-Hosseini et al., 2025), the difficulty of the task lies in the fact that the location of the signal varies from sample to sample—consequently, any successful classifier must dynamically detect and attend to the relevant tokens. Formally, let $\mathcal{D} = \{X_i, y_i\}_{i \in [n]}$ be the training data where each sample $X_i \in \mathbb{R}^{L \times d}$ has rows representing token embeddings, and the labels $y_i \in \{-1, +1\}$ are such that $\mathbb{P}(y_i = 1) =: \pi \in (0, 1)$. We assume that the token matrices $\{X_i\}_{i \in [n]}$ are independent and drawn from one of two probability distributions. Specifically, for negative samples (namely given $y_i = -1$),

$$X_i = Z_i, \tag{1}$$

where $Z_i \in \mathbb{R}^{L \times d}$ is a matrix whose entries are i.i.d. standard normal random variables. In contrast, for positive samples ($y_i = 1$),

$$X_i = \theta v_i \xi^\top + Z_i, \tag{2}$$

where $\xi$ is a fixed signal vector with $\|\xi\| = 1$, $\theta > 0$ is the parameter indicating the signal strength, and $v_i$ is a random binary-valued vector indicating the location of the hidden features: $v_i = \begin{bmatrix} \mathbb{1}_{1 \in R_i} & \cdots & \mathbb{1}_{L \in R_i} \end{bmatrix}^\top$. $R_i$ denotes the subset of tokens that contain the signal, and is assumed to have fixed cardinality $|R_i| = R \in \mathbb{N}$. The law of $v_i$ is thus supported on $\{x \in \{0,1\}^L : \sum_\ell x_\ell = R\}$, and we furthermore assume its marginals $p_j = \mathbb{P}(v_j = 1)$ for $j \in [L]$ to satisfy $\|p\| \leq CR/\sqrt{L}$ for some constant $C > 0$. This assumption essentially requires that the distribution is sufficiently spread out across tokens, and is not localized on any privileged tokens—thereby making its detection particularly challenging. In particular, when the law of the non-zero elements of $v$ is the uniform distribution on all subsets of $[L]$, $\|p\| = R/\sqrt{L}$. Therefore, an algorithm with the capacity to generalize on the task must be able to adaptively identify the subset $R_i$ containing the signal, if the sample is positive, in addition to learning the signal vector $\xi$. The latter point is further rendered non-trivial by the observation that in (2), the signal part $\theta v_i \xi^\top$ is of norm $\mathcal{O}(\theta \sqrt{R})$, which is considerably weaker than the background noise term $\|Z_i\| = \mathcal{O}(\sqrt{Ld})$ when $d$ and/or $L$ are large—thereby making the signal hard to detect. Note that this scaling differs from that considered in (Marion et al., 2024) where both terms are comparable in size—a regime corresponding to a more easily detectable signal in the limit of large dimension $d$.

Intuitively, the data distribution (2) could be interpreted as a simple model of a vision task, where each token corresponds to a patch of an input image (e.g. a computed tomography scan), and where the location of the feature $\xi$ signals the presence of a certain pattern (e.g. a lesion) at the corresponding position. This pattern is sparse ($R < L$), weak ($\|\theta v \xi^\top\| \ll \|Z\|$), and potentially rare (small $\pi$). The data distribution and task is similar in spirit to that considered in (Oymak et al., 2023), with however two important differences. While in (Oymak et al., 2023) the signal is present in all samples, in the current work the signal is totally absent from negative samples, posing the additional challenge of rarity. In addition, the relevant tokens $R_i$ are devoid of any noise in (Oymak et al., 2023) and contain only the clean signal. On the other hand, in (2) the weak signal $\xi$ is corrupted by the additive noise $Z_i$, posing the challenge of signal weakness.

## 1.1 TWO LINEAR CLASSIFIER BASELINES

We first introduce two simple linear classifiers that will serve as reference models, providing benchmarks against which the attention model—specified in the next subsection—will be evaluated.

**Vectorized linear classifier.** The first baseline flattens each matrix-valued input $X_i \in \mathbb{R}^{L \times d}$ into an $Ld$-dimensional feature vector $\text{vec}(X_i) = [(X_i^1)^\top \dots (X_i^L)^\top]^\top$, which is then fed to a linear classifier. Explicitly, the classifier is

$$\mathsf{L}_{w,b}^{\text{vec}}(X) = \text{sign}(\langle w, f_{\text{vec}}(X) \rangle + b) \quad \text{where} \ \ f_{\text{vec}}(X) = \text{vec}(X), \ w \in \mathbb{R}^{Ld}, b \in \mathbb{R}. \tag{3}$$

As noted in (Marion et al., 2024), the location of the signal within the vector would then be shifting from sample to sample due to the randomness of $R_i$—making it challenging for this vectorized linear classifier to pinpoint the relevant features.

**Pooled linear classifiers.** A possible remedy would be to instead average the input along its first dimension, rather than flattening it. More precisely, the classifier becomes

$$\mathsf{L}_{w,b}^{\text{pool}}(X) = \text{sign}(\langle w, f_{\text{pool}}(X) \rangle + b) \quad \text{where} \ \ f_{\text{pool}}(X) = \frac{1}{L} \sum_{k \in [L]} X^k, \ w \in \mathbb{R}^d. \tag{4}$$

While such an average-pooling featurization bypasses the challenge of dynamically shifting signal positions, it introduces another complication. Specifically, after averaging, the norm of the signal term $\|1_L^\top v_i \xi^\top / L\| = \mathcal{O}(R/L)$ can become significantly weaker compared to the background noise term $\|1_L^\top Z_i / L\| = \mathcal{O}(\sqrt{d/L})$, especially when $R$ is small and $L$ is large. In other words, the averaging procedure effectively reduces the signal-to-noise ratio. These intuitions will be made precise in the following section by Theorem 1 and Proposition 1, which show that a large signal strength $\theta$ is needed to counteract these limitations, in order for linear classifiers to generalize.

## 1.2 AN ATTENTION MODEL

Ideally, to remedy the issue of signal dilution suffered by the pooled linear classifier, a non-uniform, sample-dependent reweighting of the tokens should instead be deployed, selectively placing more weights on tokens that embed the signal. As we will discuss and formalize, such a reweighting can be readily implemented by an attention-based mechanism. This intuition motivates the principal model analyzed in this work: a single-layer attention-based architecture designed to tackle the sparse token classification task. Specifically, we consider the model

$$\mathsf{A}_{q,w,b}(X) = \text{sign}\big(\langle f_q(X), w \rangle + b\big), \qquad \text{with} \qquad f_q(X) = X^\top \text{softmax}(\beta X q). \tag{5}$$

This attention model $\mathsf{A}_{q,w,b}$ is parameterized by two trainable weight vectors $q, w \in \mathbb{R}^d$ and a trainable scalar bias $b \in \mathbb{R}$. In (5), the parameter $\beta$ represents the inverse temperature of the softmax activation. The formulation (5) is a simplified attention model widely studied in theoretical contexts (see, e.g., (Oymak et al., 2023; Marion et al., 2024)), in which the representation $f_q(X)$ can be viewed as analogous to the [CLS] token used for classification and readout in transformer architectures (Kenton & Toutanova, 2019). A detailed discussion connecting this simplified model with standard self-attention architectures can be found in (Marion et al., 2024; Tarzanagh et al., 2023). While we devote the theoretical study herein to the model (5), we compare empirically with more complex attention-based architectures in Appendix A.

**Dynamic reweighting and signal amplification.** An important feature of the model (5) is that the weight vector $w$ acts not directly on the raw input $X$, but instead on the attention-based feature:

$$f_q(X) = \sum_{k \in [L]} \frac{e^{\beta \langle X^k, q \rangle}}{\sum_{\ell \in [L]} e^{\beta \langle X^\ell, q \rangle}} X^k, \tag{6}$$

where each token $X^k$ is reweighted according to the scores $e^{\beta \langle X^k, q \rangle}$. Crucially, in contrast to the naive average-pooling (4) discussed in subsection 1.1 (which corresponds to the special case of $q = 0_d$), the attention scores dynamically adapt to the input tokens. Therefore, in principle, the attention mechanism can allocate greater weight to tokens containing the signal $\xi$, thus mitigating the diminished signal-to-noise ratio described following (4). Such improvement occurs when the internal attention parameter $q$ aligns non-trivially with the signal vector $\xi$; this alignment increases the inner product $\langle X^k, q \rangle$ and consequently enhances the attention weights (6) for the signal-bearing tokens. In Section 3 we formalize and rigorously prove this intuitive mechanism.

## 2 OPTIMAL TEST ERRORS IN THE LIMIT OF LONG SEQUENCES

Before analyzing how effectively the attention model (5) and the two baseline linear classifiers (3) and (4) perform when *trained* on the sparse classification task described in Section 1, it is instructive to first determine the conditions under which these models can, in principle, learn the task. In this section, we examine the *optimal test error* of the considered hypothesis classes, measuring their intrinsic ability to represent the sparse classification problem. Formally, the optimal test error for any predictor $\hat{y}_W(X)$ parametrized by some finite-dimensional parameters $W$ is defined as follows:

$$\mathcal{E}^*_{\text{test}}[\hat{y}] := \inf_W \mathcal{E}_{\text{test}}[\hat{y}_W] \quad \text{where} \quad \mathcal{E}_{\text{test}}[\hat{y}_W] := \mathbb{P}_{X,y}\left[\hat{y}_W(X) \neq y\right]. \tag{7}$$

The optimal test error corresponds to the smallest misclassification error achievable by the classifier, provided its parameters $W$ are selected optimally. Concretely, for the vectorized and pooled linear classifiers defined herein, $W$ is given by $(w, b) \in \mathbb{R}^{Ld} \times \mathbb{R}$ and $(w, b) \in \mathbb{R}^d \times \mathbb{R}$ respectively whereas for the attention model one has $W = (w, q, b) \in \mathbb{R}^d \times \mathbb{R}^d \times \mathbb{R}$. In this section, we view $\theta, R$ as sequences depending on $L$, and focus on the $L \to \infty$ regime.

**Proposition 1.** *Suppose that the limit* $\text{SNR} := \lim_{L \to \infty} \theta R / \sqrt{L}$ *exists. Then, the optimal test error of the pooled linear classifier* (4) *satisfies*

$$\lim_{L \to \infty} \mathcal{E}^*_{\text{test}}[\mathsf{L}^{\text{pool}}] = \begin{cases} 0 & \text{if } \text{SNR} = \infty, \\ (1 - \pi)\Phi(b^*) + \pi\Phi(-b^* - \text{SNR}) & \text{if } \text{SNR} \in (0, \infty) . \\ \min(\pi, 1 - \pi) & \text{if } \text{SNR} = 0 \end{cases} \tag{8}$$

*In the above display,* $b^* = -\frac{\text{SNR}}{2} - \frac{1}{\text{SNR}} \log(1/\pi - 1)$ *and* $\Phi(\cdot)$ *is the cumulative distribution function of the standard normal distribution.*

We note that a similar result appears in (Oymak et al., 2023) (Appendix A) for the pooled classifier, but for a different data distribution, and without a trainable bias.

**Theorem 1.** *Suppose that the limit* $\text{SNR} := \lim_{L \to \infty} \frac{\theta R}{\sqrt{L}}$ *exists. Then the optimal error of the vectorized classifier* (3) *satisfies*

$$\lim_{L \to \infty} \mathcal{E}^*_{\text{test}}[\mathsf{L}^{\text{vec}}] = \begin{cases} 0 & \text{if } \text{SNR} = \infty, \\ \min(\pi, 1 - \pi) & \text{if } \text{SNR} = 0 \end{cases} \tag{9}$$

*and* $\liminf_{L \to \infty} \mathcal{E}^*_{\text{test}}[\mathsf{L}^{\text{vec}}] > 0$ *if* $\text{SNR} \in (0, \infty)$.

The proofs of Proposition 1 and Theorem 1 are detailed in Appendix C. Note that since the pooled classifier can be viewed as a particular realization of the vectorized classifier with tied weights, the optimal error of the former (Proposition 1) upper bounds the optimal error of the latter (Theorem 1). Concretely, to generalize perfectly on sparse signals $R = \Theta(1)$, both the pooled and vectorized linear classifiers require a strong signal strength $\theta = \Omega(\sqrt{L})$. In this regime, the optimal weights of the pooled (resp. vectorized) classifier are proportional to the signal $\xi$ (resp. to a concatenation of $\xi$ $L$ times), and allow for vanishing test error. If the signal is weaker, namely $\theta = o(\sqrt{L})$, the model performs no better that the naive predictor that always outputs the majority label and $\mathcal{E}^*_{\text{test}} = \min(\pi, 1 - \pi)$. In the case where $\text{SNR} \in (0, \infty)$, Theorem 1 shows that the optimal test error is bounded away from zero by a strictly positive number. In sharp contrast to the linear classifiers, the attention model can perfectly classify data with a much smaller signal strength:

**Theorem 2.** *Consider the attention model* $\mathsf{A}$ *given in* (5). *In the limit* $L \to \infty$ *with* $R = \Theta(1)$, *suppose that the signal strength* $\theta$ *satisfies* $\liminf_{L \to \infty} \theta / \log L > 0$. *Then, one has* $\mathcal{E}^*_{\text{test}}[\mathsf{A}] = 0$.

The proof of Theorem 2 can be found in Appendix D. A direct consequence of Theorem 2 is that a significantly milder signal strength of order $\theta = \log L$ suffices for the attention model (5) to perfectly learn the sparse token classification task—provided it employs optimal parameters $q, w, b$. Similarly to the linear classifiers, perfect classification is in particular achieved for weights $q, \xi$ colinear to the signal $\xi$. Similar results appear in (Oymak et al., 2023) on the optimal error in a related task, but are restricted to a simpler noiseless case ($Z_i = 0$). While Theorems 1 and 2, and Proposition 1, paint a clear separation between the attention model and the two linear baselines in terms of representation power and oracle test errors, they leave the question of learnability largely

open. Furthermore, this clear-cut distinction, which happens in the large-$L$ limit, becomes less pronounced when the sequence length $L$ is finite. Thus, a more nuanced analysis of the training at finite sample complexity and sequence length is warranted. This is the objective of the following section.

## 3 PRECISE ASYMPTOTIC ANALYSIS OF THE LEARNING

In what follows, we turn our attention to the study of the training of the three models on finite datasets, aiming to precisely characterize the learning behavior of the attention model (5) and the two linear classifiers (3) and (4) in this regime. Such exact characterizations become tractable in the high-dimensional embedding limit, as demonstrated by a growing body of literature on high-dimensional attention mechanisms (Rende et al., 2024; Cui et al., 2024a; Troiani et al., 2025; Tiberi et al., 2024; Cui, 2025; Erba et al., 2024; Duranthon et al., 2025). We adopt in the remainder of this manuscript the following high-dimensional, finite-length scaling regime:

**Assumption 1** (High-dimensional, finite-length limit). *We consider the limit of large embedding dimension $d$ and comparably large number of samples $n$, namely $d, n \to \infty$ with fixed ratio $\alpha = n/d = \Theta(1)$. The chosen scaling $n \sim d$ is such that the detection of the weak signal $\xi$ from the background $Z$ is statistically possible (Lesieur et al., 2015), yet non-trivial. Meanwhile, the sequence length $L$, signal strength $\theta$, and sparsity $R$, along with all other parameters, remain finite and fixed.*

**Training procedure.** We now turn to the learning process. The attention model (5) can be trained to solve the sparse token classification task defined in subsection 1 by performing empirical risk minimization over the dataset $\mathcal{D} = \{X_i, y_i\}_{i \in [n]}$, formulated as follows:

$$\hat{q}, \hat{w}, \hat{b} \in \underset{q,w,b}{\arg\min} \hat{\mathcal{R}}_{\mathcal{D}}(q, w, b), \text{ with } \hat{\mathcal{R}}_{\mathcal{D}}(q, w, b) = \frac{1}{n} \sum_{(X,y) \in \mathcal{D}} \ell(\langle f_q(X), w \rangle + b; y) + \frac{\lambda}{2} \|w\|^2. \quad (10)$$

Here, $\ell : \mathbb{R} \times \{-1, 1\} \to \mathbb{R}$ is a loss function that is convex with respect to its first argument (for example, the logistic loss $\ell(z, y) = \log(1 + \exp(-yz))$ or the quadratic loss $\ell(z, y) = \frac{1}{2}(z - y)^2$). The empirical risk (10) also includes a ridge regularization of strength $\lambda$. Notably, compared to prior studies on sparse token tasks (Sanford et al., 2023; Wang et al., 2024; Mousavi-Hosseini et al., 2025; Marion et al., 2024; Oymak et al., 2023), our setting extends beyond the squared loss to general convex loss functions. A natural approach to solving the non-convex optimization problem (10) is to run gradient descent on the set of trainable parameters $q, w, b$. In fact, as demonstrated below, just *two* gradient steps are sufficient for the query weights $q$ to achieve an alignment with the signal $\xi$. This alignment enables the attention model (5) to develop internal representations capable of effectively identifying and amplifying the hidden signal. Specifically, we consider the following training procedure:

1. **Initialization.** Consider a partition of the training data $\mathcal{D} = \mathcal{D}_0 \cup \mathcal{D}_1$ into two disjoint sets of sizes $n_0$ and $n_1 = n - n_0$ respectively. We assume $\alpha_0 = n_0/d = \Theta(1)$, and $\alpha_1 = n_1/d = \Theta(1)$. Initialize the weights of the attention model (5) as $w^{(0)} = q^{(0)} = 0_d$, $b^{(0)} = 0$.
2. **First gradient step on $b, q, w$.** Perform a first gradient step on each of the trainable parameters on the risk $\hat{\mathcal{R}}_{\mathcal{D}_0}(q, w, b)$, using the training set $\mathcal{D}_0$ with learning rates $\eta_b, \eta_q, \eta_w$:

$$\begin{aligned} b^{(1)} &= b^{(0)} - \eta_b \frac{\mathrm{d}}{\mathrm{d}b} \hat{\mathcal{R}}_{\mathcal{D}_0}(q, w, b)\big|_{b^{(0)}, q^{(0)}, w^{(0)}}, \\ w^{(1)} &= w^{(0)} - \eta_w \nabla_w \hat{\mathcal{R}}_{\mathcal{D}_0}(q, w, b)\big|_{b^{(0)}, q^{(0)}, w^{(0)}}, \\ q^{(1)} &= q^{(0)} - \eta_q \nabla_q \hat{\mathcal{R}}_{\mathcal{D}_0}(q, w, b)\big|_{b^{(0)}, q^{(0)}, w^{(0)}}. \end{aligned}$$

$$(11)$$

More explicitly, we obtain the parameters

$$w^{(1)} = -\frac{\eta_w}{n_0 L} \sum_{i \le n_0} h_i(0, 0, 0) X_i^\top 1_L, \quad b^{(1)} = -\frac{\eta_b}{n_0} \sum_{i \le n_0} h_i(0, 0, 0), \quad q^{(1)} = 0_d, \quad (12)$$

where $1_L \in \mathbb{R}^L$ denotes a vector with all elements equal to 1, and we have also introduced the shorthand notation $h_i(q, w, b) = \ell'(\langle f_i(q), w \rangle + b; y_i)$.

3. **Second gradient step on** $q$. Note that after a first step, $q^{(1)}$ remains zero. For the attention model (5) to develop a non-trivial internal representation parametrized by $q \neq 0_d$, a second gradient step on $q$ is thus needed:

$$
\begin{aligned}
q^{(2)} &= q^{(1)} - \eta \nabla_q \hat{\mathcal{R}}_{\mathcal{D}_0}(q, w, b)\big|_{b^{(1)}, q^{(1)}, w^{(1)}} \\
&= -\frac{\eta_q \beta}{n_0 L} \sum_{i \leq n_0} h_i(0, w^{(1)}, b^{(1)}) X_i^\top \left( I - \frac{1_L 1_L^\top}{L} \right) X_i w^{(1)}.
\end{aligned}
\tag{13}
$$

4. **Full training of** $w, b$. Having developed a meaningful internal representation parametrized by $q^{(2)}$, the readout weight $w$ and bias $b$ are finally fully updated by empirical risk minimization on the retained data $\mathcal{D}_1$:

$$
\hat{w}, \hat{b} = \underset{w,b}{\operatorname{argmin}} \ \hat{\mathcal{R}}_{\mathcal{D}_1}(q^{(2)}, w, b).
\tag{14}
$$

The performance of the trained model $\mathsf{A}_{q^{(2)}, \hat{w}, \hat{b}}$ is measured by its training loss and test error

$$
\mathcal{E}_{\text{train}} = \hat{\mathcal{R}}_{\mathcal{D}_1}(q^{(2)}, \hat{w}, \hat{b}), \qquad \mathcal{E}_{\text{test}} = \mathbb{P}_{X,y}\left[ \mathsf{A}_{q^{(2)}, \hat{w}, \hat{b}}(X) \neq y \right].
\tag{15}
$$

The latter simply corresponds to the probability that the trained model misclassifies a fresh test sample $(X, y)$ at inference time.

The primary purpose of the dataset partitioning performed in step 1—splitting the data into two subsets, used respectively for steps 2–3 and step 4—is to simplify the subsequent analysis of step 4. This partitioning ensures statistical independence between the learned query weights $q^{(2)}$ and the dataset $\mathcal{D}_1$. In Appendix A, we note qualitatively similar performance when the data from steps 2–3 is reused for step 4. Adopting a more practical viewpoint, $\mathcal{D}_0$ can also be viewed as a *pre-training* dataset used to train the query weights $q$, which can then be frozen as the model is deployed on other datasets, with only the readout and bias $w, b$ being fine-tuned.

Similar stage-wise training protocols with sample splitting have previously been analyzed in the context of two-layer neural networks (Ba et al., 2022; Moniri et al., 2023; Cui et al., 2024b; Dandi et al., 2024; 2023), demonstrating how even a single gradient step on the first-layer weights can yield meaningful internal network features. Analogously, in our setting, two gradient steps on the query weights $q$ are already sufficient for the attention model to develop informative internal representations. For transformer models, similar few-step analyses were conducted for instance in (Bietti et al., 2024; Oymak et al., 2023), however without the final step of full empirical risk minimization. This final optimization of the output weight $w$ can be taken as an analog to transfer learning, thus lending to more practical insights for real training procedures.

We are now in a position to present our main technical results: a precise characterization of the test error (15) achieved by the attention model (5), trained using the four-stage procedure detailed in subsection 3. In the following sections, we first analyze step 3—demonstrating precisely how the query weights $q^{(2)}$ develop an alignment with the signal $\xi$, resulting in nontrivial attention weightings. We then examine how this learned attention mechanism leads to an improvement in the test error (15), as compared to the baseline linear classifiers (4) and (3) at the conclusion of step 4.

**Characterization of the attention weights after two gradient steps.** The first technical result characterizes how, at the end of step 3 (see subsection 3), the query weights $q = q^{(2)}$ develop a non-zero alignment with the signal vector $\xi$. As will be discussed in a subsequent subsection, this alignment allows the attention model (5) to develop internal representation adapted to the task.

**Theorem 3** (Characterization of the query weights $q^{(2)}$ after two gradient steps). *In the asymptotic limit of Assumption 1, $\|q^{(2)}\|$ and $\langle q^{(2)}, \xi \rangle$ converge in probability to deterministic limits, whose expressions are given in Appendix E.*

Theorem 3 precisely characterize the parameters of the attention model (5) at the conclusion of step 3 of the training procedure described in subsection 3. The detailed proof of Theorem 3 is provided in Appendix E. A direct consequence of Theorem 3 is that the alignment between the query weights after two gradient steps $q^{(2)}$ and the signal $\xi$, as captured by the cosine similarity $\langle q^{(2)}, \xi \rangle / \|q^{(2)}\|$, tends rapidly in absolute value to its maximal value of 1 as the sample complexity $\alpha_0$ is increased, at a $1/\alpha_0$ rate. This observation is formalized in the following Corollary.

**Corollary 1** (Cosine similarity). *In the asymptotic limit of Assumption 1, the cosine similarity $\langle q^{(2)}, \xi \rangle / \|q^{(2)}\|$ converges in probability to a limit $s_q$. If furthermore the right-hand side of (72) is non-zero, its absolute value admits the expansion $|s_q| = 1 - C/\alpha_0 + o(1/\alpha_0)$. The expression of the constant $C$ is detailed in Appendix E.*

**Characterization of final test and training errors.** Having described steps $1 - 3$ of the training procedure 3, we now focus on step 4, where given $q^{(2)}$, the readout weights $w$ and the bias $b$ are fully trained on the held-out data batch $\mathcal{D}_1$. We note that once $q$ is fixed, the empirical risk minimization (14) amounts to training a linear model with weights $w, b$ on the high-dimensional non-linear features $f_{q^{(2)}}(X)$ (6). While the behavior of such linear classifiers is in general very well understood in the asymptotic limit of Assumption 1 (Candès & Sur, 2020; Liang & Sur, 2022; Montanari et al., 2019; Mai et al., 2019; Loureiro et al., 2021; Mignacco et al., 2020a), such works very often build on the assumption of simple (e.g. Gaussian mixture) data distributions. In the present case however, the features $f_{q^{(2)}}(X)$ possess a highly non-trivial distribution, as they result from the non-linear attention mechanism. Fortunately, the softmax acts only on the low-dimensional projection $g \in \mathbb{R}^L$ of the tokens along the query weights $q^{(2)}$, which can be handled separately. The idea of the proof, detailed in Appendix F, proceeds from this observation. The final results are succinctly summarized in the following theorem, while the full technical statement is deferred to Appendix F.

**Theorem 4** (Test and training errors after step 4). *The test error and training loss associated to the empirical risk minimization (14) converge in probability in the limit of Assumption 1 to deterministic limits $\mathcal{E}_{\text{test}}[A]$ and $\mathcal{E}_{\text{train}}[A]$, whose expression are deferred for clarity to Appendix F.*

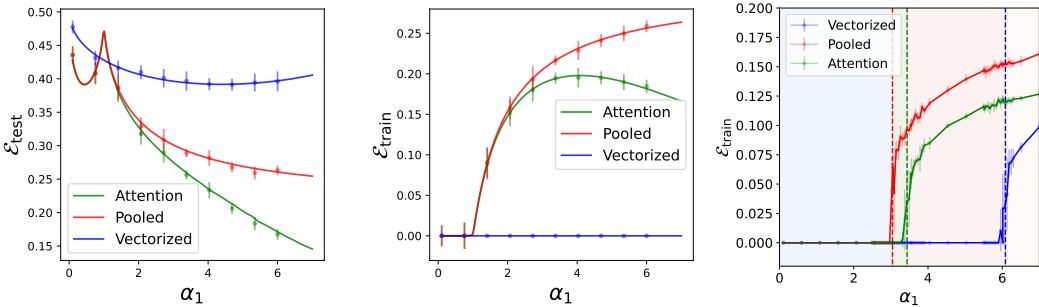

Figure 1: Test (**left**) and train (**middle**) errors achieved by the attention model (5) and the pooled (4) and vectorized (3) classifiers, for $L = 10, R = 1, \pi = 0.5, \theta = 5, \lambda = 10^{-5}, \eta_{b,q,w} = 0.1, \alpha_0 = \alpha_1$, trained with the square loss, as a function of the normalized number of samples $\alpha_1$. Solid lines correspond to the theoretical characterizations of Theorem 4. Dots represent numerical experiments in dimension $d = 1000$. Error bars represent one standard deviation over 8 trials. (**right**) Training loss $\mathcal{E}_{\text{train}}$ for the attention model (green), and the pooled (red) and vectorized (blue) linear classifiers, as a function of the sample complexity $\alpha_1$. $L = 2, R = 1, \theta = 2, \pi = 0.3$. The attention model has a unit norm query weight $q$ with alignment $\gamma = 0.99$ with the signal $\xi$. Dots correspond to numerical simulations in dimension $d = 2000$; error bars represent one standard deviation over 20 trials. Dashed lines: theoretical prediction of the separability thresholds, as given in Conjecture 1.

Theorem 4 provides an exact characterization—precise down to explicit constants—of the test error attained by the attention model (5), trained according to the procedure described in subsection 3, within the high-dimensional limit specified by Assumption 1. The resulting expression is formulated in terms of a small set of scalar summary statistics, which are determined as solutions to a system of self-consistent equations. While the latter still possess a rather intricate form, they can considerably simplify in some simple cases, yielding valuable insights. We detail such an instance in the following, for the case of a square loss in the ridgeless limit. Let us remark that while (Oymak et al., 2023) also provide error bounds for a three-gradient-steps protocol, Theorem 4 offers tight error characterizations, exact down to explicit constants. While the same work also reports sharp characterizations (Theorem 8) for the special case of the square loss, those results are restricted to a much simpler learning protocol that involves neither gradient steps nor empirical risk minimization, and that necessitates further oracle information on the set of relevant tokens.

**Baseline classifiers.** Having characterized the test error and training loss of the attention model, we now turn to the case of the two linear classifiers $\mathsf{L}_{w,b}^{\mathrm{pool}}, \mathsf{L}_{w,b}^{\mathrm{pool}}$, whose parameters $w, b$ are trained on the dataset $\mathcal{D}_1$ through the empirical risk minimization

$$\hat{w}, \hat{b} \in \operatorname*{argmin}_{w,b} \hat{\mathcal{R}}_{\mathcal{D}_1}(w, b), \text{ with } \hat{\mathcal{R}}_{\mathcal{D}_1}(w, b) = \frac{1}{n} \sum_{(X,y) \in \mathcal{D}} \ell(\langle f(X), w \rangle + b; y) + \frac{\lambda}{2} \|w\|^2, \quad (16)$$

where $f \in \{f_{\mathrm{pool}}, f_{\mathrm{vec}}\}$, and $\ell$ is an arbitrary strictly convex loss function. As for the attention model, a tight characterization can be reached for the associated test error and training loss, leveraging the observation that the distribution of the features $f_{\mathrm{vec}}(X), f_{\mathrm{pool}}(X)$ are in fact simple Gaussian mixtures with respectively $\binom{L}{R} + 1$ and 2 isotropic clusters. The test error and training loss of generalized linear classifiers in the high-dimensional limit of Assumption 1 for such data distribution has been characterized in prior works (Mignacco et al., 2020a; Loureiro et al., 2021). We briefly summarize the corresponding results below.

**Theorem 5** (Errors for the linear classifiers). *[(Loureiro et al., 2021)] In the asymptotic limit of Assumption 1, the test error and training loss for the pooled (resp. vectorized) linear classifier converge in probability to limits $\mathcal{E}_{\mathrm{train}}[\mathsf{L}^{\mathrm{pool}}]$ and $\mathcal{E}_{\mathrm{test}}[\mathsf{L}^{\mathrm{pool}}]$ (resp. $\mathcal{E}_{\mathrm{train}}[\mathsf{L}^{\mathrm{vec}}]$ and $\mathcal{E}_{\mathrm{test}}[\mathsf{L}^{\mathrm{vec}}]$).*

We defer the precise exposition of the expressions of $\mathcal{E}_{\mathrm{train}}[\mathsf{L}^{\mathrm{vec}}], \mathcal{E}_{\mathrm{test}}[\mathsf{L}^{\mathrm{vec}}]$ to Appendix G. For completeness, and to help readers connect and compare the proofs of Theorems 5 and 4, we also present in the same Appendix an alternate sketch of proof using the same leave-one-out approach as that leveraged in the proof of Theorem 4.

**Comparison of the three models.** The theoretical predictions for the training and test errors from Theorem 4 and 5—for both the attention model (5) and the linear baselines (4)(3)—are compared with numerical simulations in dimension $d = 1000$ in Fig. 1, demonstrating excellent agreement. The figure clearly illustrates how the learned attention mechanism leads to superior test performance compared to the linear classifiers, which lack this adaptive representation capability.

To garner further quantitative insights from the technical results of Theorem 4 and 5, let us focus on the particular case of a quadratic loss function $\ell(z, y) = 1/2(y - z)^2$, in the limit of vanishing regularization $\lambda = 0^+$. In this setting, the characterizations of Theorems 4 and 5 considerably simplify, revealing further insights, which we describe in the following Corollary.

**Corollary 2** (Ridgeless quadratic loss). *For a quadratic loss function $\ell(z, y) = 1/2(y - z)^2$, and $\lambda = 0$, the asymptotic limits $\mathcal{E}_{\mathrm{test}}[\mathsf{A}], \mathcal{E}_{\mathrm{test}}[\mathsf{L}^{\mathrm{pool}}]$, and $\mathcal{E}_{\mathrm{test}}[\mathsf{L}^{\mathrm{vec}}]$ characterized in Theorems 4 and 5 tend to their $\alpha_1 \to \infty$ limits $\mathcal{E}_{\mathrm{test}}^\infty[\mathsf{A}]$ and $\mathcal{E}_{\mathrm{test}}^\infty[\mathsf{L}^{\mathrm{pool}}] = \mathcal{E}_{\mathrm{test}}^\infty[\mathsf{L}^{\mathrm{vec}}]$ at a rate $1/\alpha_1$.*

A number of interesting conclusions can be garnered from Corollary 2. First, all three test errors tend to their respective $\alpha_1 \to \infty$ limit at the same $1/\alpha_1$ rate, as the sample complexity $\alpha_1$ is increased. Furthermore, the two linear classifiers $\mathsf{L}^{\mathrm{pool}}, \mathsf{L}^{\mathrm{vec}}$ tend to a common limit $\mathcal{E}_{\mathrm{test}}^\infty[\mathsf{L}]$. This finding somewhat echoes the intuition from Theorem 1, which already suggested that both models share similar oracle—and thus plausibly infinite sample complexity—behaviors. Lastly, one may naturally wonder which of the limiting test errors $\mathcal{E}_{\mathrm{test}}^\infty[\mathsf{A}], \mathcal{E}_{\mathrm{test}}^\infty[\mathsf{L}]$ is lower—in particular, whether the attention model always achieves a lower error provided it is given sufficient data. The answer is more nuanced, and crucially depends on the alignment $s_q$ (see Corollary 1) between the query weights $q^{(2)}$ and the signal $\xi$ achieved after step 3 of the training protocol. As shown in Fig. 5 in Appendix F, $\mathcal{E}_{\mathrm{test}}^\infty[\mathsf{A}] > \mathcal{E}_{\mathrm{test}}^\infty[\mathsf{L}]$ can hold in some settings for $s_q$ sufficiently small. In simple words, when the query weights have insufficiently aligned with the signal—e.g. as a result of insufficient data $\alpha_0$ or bad choice of the hyperparameters $\eta_{w,b}$—, the attention suffers from a misaligned internal representation, and achieves a worse error than the simpler linear classifiers. For moderate and large $s_q$ on the other hand, $\mathcal{E}_{\mathrm{test}}^\infty[\mathsf{A}] < \mathcal{E}_{\mathrm{test}}^\infty[\mathsf{L}]$ and the attention profits from the advantage of the dynamical reweighting implemented by its internal representation.

**Capacity.** The previous subsection compared the three models in terms of their test errors. We adopt in this subsection a complementary perspective, and analyze the *capacity* $\alpha^\star$ of the models $\mathsf{A}_{q^{(2)},w,b}, \mathsf{L}_{w,b}^{\mathrm{pool}}$ and $\mathsf{L}_{w,b}^{\mathrm{vec}}$, defined as the (normalized) maximal number of training samples that can typically be fitted by the models to vanishing training loss. More formally, let $\hat{y} \in \{\mathsf{A}, \mathsf{L}^{\mathrm{pool}}, \mathsf{L}^{\mathrm{vec}}\}$ be one of the three models, and let $\mathcal{E}_{\mathrm{train}}[\hat{y}](\alpha_1)$ designate the asymptotic training loss characterized

in Theorems 4 and 5, in the limit of vanishing regularization $\lambda \to 0$, for the logistic loss $\ell(y, z) = \log(1 + \exp(-yz))$. The *capacity* of the model $\hat{y}$ is then formally defined as

$$\alpha_{\hat{y}}^{\star} = \sup_{\alpha \geq 0} \{\mathcal{E}_{\text{train}}[\hat{y}](\alpha) = 0\} \tag{17}$$

For $\alpha < \alpha_{\hat{y}}^{\star}$, the training set is small enough so that it can with high probability be perfectly separated by the model and $\mathcal{E}_{\text{train}}[\hat{y}](\alpha) = 0$. At large sample complexities $\alpha > \alpha_{\hat{y}}^{\star}$, such perfect classification becomes typically impossible, resulting in a positive training loss $\mathcal{E}_{\text{train}}[\hat{y}](\alpha) > 0$. The capacity of a model captures how easily it can classify samples from a given data distribution, with a higher capacity thus intuitively reflecting a higher adequacy of the model to the task. An analytical expression for the capacity can be extracted from the characterizations of the training loss $\mathcal{E}_{\text{train}}$ provided by Theorems 4 and 5, which we report in the following Conjecture.

**Conjecture 1.** *The capacities of the models* $\mathsf{L}^{\text{pool}}, \mathsf{L}^{\text{vec}}, \mathsf{A}_{q^{(2)},w,b}$ *admit the following expressions:*

$$\alpha_{\text{vec}}^{\star} = \max_{s \in [0,1], \mathfrak{b}} \frac{L(1 - s^2)}{\int_0^\infty \left[\pi \Phi'\left(\mathfrak{b} + \frac{\theta R}{\sqrt{L}}s + u\right) + (1-\pi)\Phi'(u - \mathfrak{b})\right] u^2 \, du}, \quad \alpha_{\mathsf{A}}^{\star} = \max_{m_q, m_\xi, \mathfrak{b}} \frac{1}{\mathbb{E}\left[c_z^3 \int_0^\infty \Phi'\left(\frac{c_z^2 u + y(b + c_q m_q + c_\xi m_\xi)}{c_z}\right) u^2 \, du\right]}. \tag{18}$$

*and* $\alpha_{\text{pool}}^{\star} = \alpha_{\text{vec}}^{\star}/L$. *The expectation in the expression of* $\alpha_{\mathsf{A}}^{\star}$ *bears on* $y, c_z, c_\xi, c_q$ *whose joint law is detailed in Lemma 1, and depends in particular on* $\langle q^{(2)}, \xi \rangle$.

The derivation of the expressions (18) is detailed in Appendix I. Because they involve some heuristic step, we state the result as a conjecture. The capacity of linear classifiers has been studied in a rich line of prior works, e.g. (Candès & Sur, 2020; Mignacco et al., 2020a; Loureiro et al., 2021), impulsed by the seminal work of (Cover, 2006), albeit no analytical expressions have been to our awareness reported for the data distribution considered in the present work. Such results are on the other hand scarce for attention-based models. Conjecture 1 contributes to bridging this gap, by reporting an analytical expression for the capacity of the simple attention model considered in the present work. The theoretical predictions (18) are plotted in Fig. 1, where they are overlayed upon numerical evaluations of the training loss $\mathcal{E}_{\text{train}}$, for the three models, revealing good agreement. In the probed setting, $\alpha_{\text{vec}}^{\star} > \alpha_{\mathsf{A}}^{\star} > \alpha_{\text{pool}}^{\star}$, the attention model displays a higher capacity than the pooled classifier, while the higher capacity of the vectorized classifier can be explained from its operating in a $L-$times higher dimensional space. As we discussed above, this higher capacity of the attention model intuitively hints at a better suitability to the considered data distribution.

Finally, we note that this ordering can vary depending on the parameters of the problem, and crucially on the alignment $s_q$ achieved by the attention model between its query weights $q^{(2)}$ and the signal $\xi$, as characterized in Corollary 1. We discuss in Appendix I how a small $s_q$—resulting, for instance, from insufficient pretraining data $\alpha_0$ or bad choice of hyperparameters $\eta_{w,b}$—can result in the attention having a lower capacity than the pooled classifier, namely $\alpha_{\mathsf{A}}^{\star} < \alpha_{\text{pool}}^{\star}$. This echoes a similar observation at the level of the test error made in the previous subsection, and discussed in Appendix F.

## CONCLUSION

We study the sparse token classification task of detecting a sparse, weak, and rare signal embedded in sequential data. For long sequences, we rigorously establish a clear performance separation between linear and attention-based classifiers, showing that attention-based models require significantly weaker signals to achieve perfect generalization. For finite sequences, we provide a sharp analysis of the learning for a simple attention model in a high-dimensional limit. Specifically, our study demonstrates how merely two gradient steps suffice for the attention mechanism to learn meaningful internal representations, enabling the model to dynamically identify and focus on tokens containing the relevant signal. Moreover, we derive a sharp characterization of the resulting test error, quantifying precisely the performance gain achieved by the attention model relative to the linear classifier baselines. Finally, we put these results in perspective by analyzing the capacity of the three models.

ACKNOWLEDGMENTS

We thank Odilon Duranthon, Lenka Zdeborová, Claire Boyer, Pierre Marion, and Bruno Loureiro for insightful discussions. NB acknowledges support from the *Fonds de recherche du Québec — Nature et technologies* (FRQNT) in the form of a Master's Training Scholarship. HC acknowledges support from the Center of Mathematical Sciences and Applications (CMSA) of Harvard University. Y.M.L is supported by a Harvard College Professorship, the Harvard FAS Dean's Fund for Promising Scholarship, and DARPA under grant DIAL-FP-038.

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

# A    IMPARTING TO MORE COMPLEX ARCHITECTURES

We considered so far the simple attention model in (5) and the training procedure described in sub-section 3 to allow for a tractable analysis. In this last section, we provide some synthetic numerical experiments evidencing the parallels between our setup and more complex attention mechanisms. For comparison, we consider three models that build upon the attention model (5) which we refer to as $\mathcal{M}_0$. For weight matrices $W_Q, W_K, W_V \in \mathbb{R}^{d \times d}$, let $Q = XW_Q, K = XW_K, V = XW_V$, and define the attention weights $A = \mathrm{softmax}\left(QK^\top / \sqrt{d}\right)$. Akin to the classical self-attention mechanism considered in the seminal work of Vaswani et al. (2017), let $\mathcal{M}_1$ and $\mathcal{M}_2$ be the models with outputs:

$$\mathcal{M}_1: \quad f(X) = \frac{1}{L} \sum_{i=1}^{L} (AV)_i \xrightarrow[\text{output}]{} \mathrm{sign}\big(\langle f(X), w \rangle + b\big), \tag{19}$$

$$\mathcal{M}_2: \quad h(X) = \phi(W_h f(X) + c) \xrightarrow[\text{output}]{} \mathrm{sign}\big(\langle h(X), w \rangle + b\big),$$

where $\phi = \mathrm{ReLU}$ and $(W_h, c, w, b)$ are learnable weights. For a final comparison, we also consider a multi-head, multi-layer attention model $\mathcal{M}_3$ (4 heads and 2 layers) with linear activation and final output defined analogously to (19).

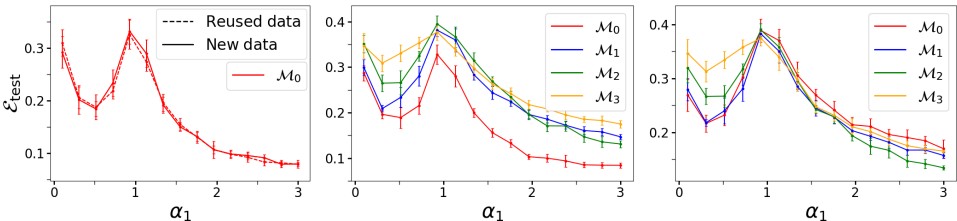

Figure 2:   Test errors achieved by the models $\{\mathcal{M}_i\}_{i=0}^3$ for $L = 10$, $R = 3$, $\pi = 0.5$, $\theta = 3$, and $\lambda = 10^{-2}$, trained on square loss using Adam optimizer (pre-training stage) before freezing inner model weights and optimizing readout weights (fine-tuning stage). Error after 2 epochs (**left**, **middle**) and 100 epochs (**right**) of pretraining are shown. Comparison between reusing pretraining data versus generating new data for finetuning (as in subsection 3) is also shown (**left**). Curves represent numerical experiments in dimension $d = 500$; error bars show one standard deviation over 8 trials.

We plot in Fig. 2 the learning curves of these different models. We employ mini-batch Adam (Kingma & Ba, 2015) instead of full-batch gradient descent, and vary the number of pretraining epochs. Fig. 2 (**left**) shows that using dataset $\mathcal{D}_1$ for the training of $w, b$, as we considered in 3, yields the same behavior as reusing the dataset $\mathcal{D}_0$ employed in the first pretraining steps. Qualitatively, in all probed settings the test curves for the model $\mathcal{M}_0$ display a strong likeness to the analytic curves provided in Fig. 1. This similarity in the shape of the loss curves between the more complex models and the simple model analyzed above can be observed even after 100 epochs of pretraining. For instance, the double-descent phenomenon of Fig. 1 remains present. As a point of contrast, when using only 2 epochs of pretraining, $\mathcal{M}_0$ out-performs the other models (Fig. 2, **middle**), which may be attributed to the much larger parameter spaces being optimized over by more complex models. Unsurprisingly, this observation flips with more pretraining (Fig. 2, **right**).

# B    AUXILIARY RESULTS

Throughout this appendix, for two random variables $X$ and $Y$, we write

$$X \stackrel{(d)}{=} Y$$

to mean that the two random variables are equal in distribution. For example, as is used often in our derivations, given a matrix $G \in \mathbb{R}^{m \times n}$ with i.i.d. $\mathcal{N}(0, 1)$ entries, an independent Gaussian vector $g \sim \mathcal{N}(0, I_m)$, and another independent random vector $u \in \mathbb{R}^n$, a basic fact is

$$Gu \stackrel{(d)}{=} \|u\|g.$$

Moreover, for two (possibly random) sequences $(a_n)$ and $(b_n)$, we write

$$a_n \asymp b_n$$

if $\lim_{n \to \infty} |a_n - b_n| = 0$, where the convergence may be taken in the almost-sure or in-probability sense depending on the context.

We first present a statistically equivalent representation of the feature vector $f_q(X)$ in the attention model defined in (6).

**Lemma 1.** *Let $g \in \mathbb{R}^L$ and $z \in \mathbb{R}^d$ be two independent random vectors with i.i.d. standard normal entries. Define two probability vectors*

$$s_+ := \mathsf{softmax}\big(\beta(\|q\|g + \langle q, \xi \rangle \theta v)\big) \qquad and \qquad s_- := \mathsf{softmax}(\beta\|q\|g). \tag{20}$$

*We have*

$$f_q(X)\big| \{y = +1\} \overset{(d)}{=} \frac{\langle g, s_+ \rangle q}{\|q\|} + \langle \theta v, s_+ \rangle \xi + \|s_+\| P_q^\perp z,$$

*and*

$$f_q(X)\big| \{y = -1\} \overset{(d)}{=} \frac{\langle g, s_{-1} \rangle q}{\|q\|} + \|s_-\| P_q^\perp z,$$

*where*

$$P_q^\perp = I - \frac{qq^\top}{\langle q, q \rangle}$$

*is the orthogonal projection onto the subspace orthogonal to $q$.*

*Proof.* By the rotational invariance of the isotropic Gaussian distributions, we can write

$$Z \overset{(d)}{=} \frac{gq^\top}{\|q\|} + \widetilde{Z} P_q^\perp, \tag{21}$$

where $\widetilde{Z}$ is an independent copy of $Z$. The result is straightforward after inserting the representation (21) into (6), which provides

$$f_q(X)\big| \{y = +1\} \overset{(d)}{=} \left( \frac{gq^\top}{\|q\|} + \widetilde{Z} P_q^\perp + \theta v \xi^\top \right)^\top \mathsf{softmax}\left(\beta\|q\|g + \beta\theta v \langle q, \xi \rangle\right)$$

$$\overset{(d)}{=} \frac{\langle g, s_+ \rangle q}{\|q\|} + \langle \theta v, s_+ \rangle \xi + \|s_+\| P_q^\perp z.$$

In the above, we have used the facts that $P_q^\perp q = 0_d$ and $\widetilde{Z} s_+ \overset{(d)}{=} \|s_+\|g$. The signal-less case (for $y = -1$) follows analogously. $\square$

The following result gives a simplified form for the test error that is valid for any $q, w \in \mathbb{R}^d$ and $b \in \mathbb{R}$. It will be used in the proofs of Theorems 2 and 4.

**Lemma 2.** *Let*

$$\mu_1 = \frac{\langle q, w \rangle}{\|q\|}, \qquad \mu_2 = \langle \xi, w \rangle, \qquad \mu_3 = \sqrt{\|w\|^2 - \mu_1^2}.$$

*The test error is*

$$\mathcal{E}_{\text{test}} = (1 - \pi) \cdot \mathbb{E}_g \left[ \Phi\left( \frac{b + \langle g, s_- \rangle \mu_1}{\mu_3 \|s_-\|} \right) \right] + \pi \cdot \mathbb{E}_g \left[ \Phi\left( \frac{-b - \langle \theta v, s_+ \rangle \mu_2 - \langle g, s_+ \rangle \mu_1}{\mu_3 \|s_+\|} \right) \right],$$

*where $s_+$ and $s_-$ are the two vectors defined in (20), and $\Phi(\cdot)$ is the cumulative distribution function of a standard normal distribution.*

*Proof.* By (5),

$$\mathcal{E}_{\text{test}} := (1 - \pi)\mathbb{P}\left( \langle f_q(X), w \rangle + b > 0 \,\big|\, y = -1 \right) + \pi\mathbb{P}\left( \langle f_q(X), w \rangle + b < 0 \,\big|\, y = +1 \right).$$

The result then follows from the statistical representations given by Lemma 1. $\square$

## C PROOFS OF PROPOSITION 1 AND THEOREM 1

**Proof of Proposition 1**

Notice that the pooled classifier corresponds to setting $q = 0$ in the attention model (6) and we have

$$\langle f_0(X), w \rangle + b \stackrel{(d)}{=} \frac{\|w\|}{\sqrt{L}} z + 1_{\{y=1\}} \frac{\theta R \langle w, \xi \rangle}{L} + b$$

for $z \sim \mathcal{N}(0, 1)$. Absorbing the factor $-\sqrt{L}/\|w\|$ by redefining the variable $b$, we obtain

$$
\begin{aligned}
\mathcal{E}^*_{\text{test}}[\hat{y}] &= \inf_{w \in \mathbb{R}^d, b \in \mathbb{R}} (1-\pi)\mathbb{P}(z > b) + \pi \mathbb{P}\left(z + \frac{\theta R \langle w, \xi \rangle}{\|w\|\sqrt{L}} < b\right) \\
&= \inf_{\rho \in [-1,1], b \in \mathbb{R}} (1-\pi)\mathbb{P}(z > b) + \pi \mathbb{P}\left(z + \frac{\rho \theta R}{\sqrt{L}} < b\right) \\
&= \inf_{b \in \mathbb{R}} (1-\pi)\mathbb{P}(z > b) + \pi \mathbb{P}\left(z + \frac{\theta R}{\sqrt{L}} < b\right) \\
&= \inf_{b \in \mathbb{R}} (1-\pi)\Phi(-b) + \pi \Phi\left(b - \frac{\theta R}{\sqrt{L}}\right).
\end{aligned}
\tag{22}
$$

Set $\ell_L = \theta R/\sqrt{L}$ and Denote $g_L(b) = (1-\pi)\Phi(-b) + \pi\Phi(b - \ell_L)$ the function over which the infimum is taken in (22). For any $L$, $g_L$ admits the derivative

$$g'_L(b) = \frac{e^{-\frac{b^2}{2}}}{\sqrt{2\pi}} \left[ -1 + \pi + \pi e^{-\frac{\ell_L^2}{2} + \ell_L b} \right]. \tag{23}$$

We assume without loss of generality that $\ell_L > 0$ since the asymptotic test error shall remain the same for when $\ell_L \to 0$ as $L \to \infty$. We have that the derivative $g'_L(b)$ is zero at

$$b_L^* = \frac{1}{2}\left(\ell_L - 2/\ell_L \log(\pi/1-\pi)\right), \tag{24}$$

where it switches sign from negative to positive. Therefore, the infimum in (22) is attained at $b_L^*$ and

$$
\begin{aligned}
\mathcal{E}^*_{\text{test}}[\hat{y}] &= (1-\pi)\Phi(-b_L^*) + \pi\Phi(b_L^* - \ell_L) \\
&= (1-\pi)\Phi\left(-\frac{1}{2}\left(\ell_L - 2/\ell_L \log(\pi/1-\pi)\right)\right) + \pi\Phi\left(\frac{1}{2}\left(\ell_L - 2/\ell_L \log(\pi/1-\pi)\right) - \ell_L\right).
\end{aligned}
\tag{25}
$$

Inspecting (25), by continuity of $\Phi$ we immediately see that when $\ell = \infty$ one has $\lim_{L \to \infty} \mathcal{E}^*_{\text{test}}[\hat{y}] = 0$ and when $\ell \in (0, \infty)$ we obtain the corresponding expression in the statement of Theorem 1. Notice that for $\ell = 0$,

$$\lim_{L \to \infty} -2/\ell_L \log(\pi/1-\pi) = \begin{cases} -\infty, & \pi > 1 - \pi \\ \infty, & \pi < 1 - \pi \\ 0, & \pi = 1/2 \end{cases}$$

and so, again examining (25), it follows that under this regime $\lim_{L \to \infty} \mathcal{E}^*_{\text{test}}[\hat{y}] = \min(\pi, 1 - \pi)$. $\square$

**Proof of Theorem 1**

Before dividing into the two separate cases of Theorem 1, we begin with a simplification of the optimal test error. Writing $w = (w_1, \ldots, w_L)$ with $w_\ell \in \mathbb{R}^d$, we have

$$\langle \text{vec}(X), w \rangle + b = \|w\|z + 1_{y=1}\theta \sum_{\ell=1}^{L} v_\ell \langle w_\ell, \xi \rangle + b$$

for $z \sim \mathcal{N}(0,1)$. Absorbing $\|w\|$ into $b$ gives

$$
\begin{aligned}
\mathcal{E}^*_{\text{test}}[\hat{y}] &= \inf_{w \in \mathbb{R}^{Ld}, b \in \mathbb{R}} (1-\pi)\mathbb{P}\left(z > b\right) + \pi\mathbb{P}\left(z + \theta \sum_{\ell=1}^{L} v_\ell \frac{\langle w_\ell, \xi \rangle}{\|w\|} < b\right) \\
&= \inf_{w \in S^{Ld-1}, b \in \mathbb{R}} (1-\pi)\mathbb{P}\left(z > b\right) + \pi\mathbb{P}\left(z + \theta \sum_{\ell=1}^{L} v_\ell \langle w_\ell, \xi \rangle < b\right) \\
&= \inf_{a \in S^{d-1} \cap \mathbb{R}^d_+, b \in \mathbb{R}} (1-\pi)\mathbb{P}\left(z > b\right) + \pi\mathbb{P}\left(z + \theta\langle v, a \rangle < b\right)
\end{aligned}
\tag{26}
$$

The last line follows as any optimal $w$ will be of the form $w_\ell = a_\ell \xi$ for $1 \leq \ell \leq L$ where $a_\ell \geq 0$ and $\|a\| = 1$.

With the representation for $\mathcal{E}^*_{\text{test}}[\hat{y}]$ given in (26), we now establish the separate results of the theorem.

1. Recalling the assumption

$$
\|p\| = O\left(\frac{R}{\sqrt{L}}\right) \quad \text{where } p_j = \mathbb{P}(v_j = 1) \text{ for } j \in [L],
\tag{27}
$$

there exists $C > 0$ such that $\|p\| \leq C(R/\sqrt{L})$ for all $L \geq 1$. To begin, defining the random variable $u = \theta\langle v, a \rangle$ and the decreasing function $f_b(x) = \Phi(-x - b)$, notice that (26) is equivalent to

$$
\inf_{a \in S^{d-1} \cap \mathbb{R}^d_+, b \in \mathbb{R}} (1-\pi)\Phi(b) + \pi\mathbb{E}[f_b(u)].
\tag{28}
$$

where the dependence on $a$ persists through $u$ and the expectation is taken with respect to $u$.

We first show that if $\ell^* = \infty$, one has $\mathcal{E}^*_{\text{test}}[\hat{y}] \to 0$ as $L \to \infty$. To this end, consider a "flat" solution $a = 1/\sqrt{L} \cdot \mathbf{1}_L$ and notice that this gives $u = \mathbb{E}[u] = \theta R/\sqrt{L}$. Thus, we have

$$
\mathbb{E}_u[f_b(u)] = f_b(\theta R/\sqrt{L}) = \Phi(-\theta R/\sqrt{L} - b).
$$

Taking $b = -\theta R/2\sqrt{L}$, we have

$$
\mathcal{E}^*_{\text{test}}[\hat{y}] \leq (1-\pi)\Phi(-\theta R/2\sqrt{L}) + \pi\Phi(-\theta R/2\sqrt{L}) \xrightarrow{L \to \infty} 0.
$$

Next, we show that $\mathcal{E}^*_{\text{test}}[\hat{y}] \to \min(\pi, 1-\pi)$ if $\ell^* = 0$. Observe that for $\nu > 0$,

$$
\begin{aligned}
\mathbb{E}_u[f_b(u)] &\geq \mathbb{E}_u[f_b(u)\mathbf{1}_{\{u \leq k\}}] \\
&\geq f_b(k)\mathbb{P}(u \leq k) \\
&\geq f_b(k)\left(1 - \frac{\mathbb{E}[u]}{k}\right)
\end{aligned}
\tag{29}
$$

where the second and third inequalities above are due to the monotonicity of $f_b$ and Markov's inequality respectively. Note that

$$
\mathbb{E}[u] = \theta\langle p, a \rangle \leq \theta\|p\| \leq C\frac{R}{\sqrt{L}}
$$

by our delocalization assumption. Setting $\nu_L = (C\theta R/\sqrt{L})^{1/2}$, from (28) and (29), we have

$$
\mathcal{E}^*_{\text{test}}[\hat{y}] \geq \inf_{b \in \mathbb{R}} (1-\pi)\Phi(b) + \pi\Phi(-\nu_L - b)(1 - \nu_L).
$$

Since $\Phi(-\nu_L - b)(1 - \nu_L) \xrightarrow{L \to \infty} \Phi(-b)$ uniformly in $b \in \mathbb{R}$, we have

$$
\liminf_{L \to \infty} \mathcal{E}^*_{\text{test}}[\hat{y}] \geq \liminf_{L \to \infty} \left(\inf_{b \in \mathbb{R}} (1-\pi)\Phi(b) + \pi\Phi(-\nu_L - b)(1 - \nu_L)\right)
\tag{30}
$$

$$
= \inf_{b \in \mathbb{R}} (1-\pi)\Phi(b) + \pi\Phi(-b)
\tag{31}
$$

$$
= \min(\pi, 1-\pi).
\tag{32}
$$

On the other hand,

$$\limsup_{L \to \infty} \mathcal{E}_{\text{test}}^*[\hat{y}] \leq \min(\pi, 1 - \pi)$$

as the upper bound above can be achieved by setting the original weight vector $w = 0_d$. This establishes that $\lim_{L \to \infty} \mathcal{E}_{\text{test}}^*[\hat{y}] = 0$ when $\ell^* = 0$. Finally, we consider the case where $\ell^* \in (0, \infty)$. Setting $\bar{u} = \max(u, -b)$, we have

$$\mathbb{E}[f_b(u)] \geq \mathbb{E}[f_b(\bar{u})] \geq f_b(\mathbb{E}[\bar{u}]) = \Phi(-\mathbb{E}[\bar{u}] - b) \tag{33}$$

where the first inequality is due to the monotonicity of $f_b$ and the second comes is by Jensen's inequality seeing that $f_b(x)$ is convex for $x \geq -b$. This provides a lower bound

$$\mathcal{E}_{\text{test}}^*[\hat{y}] \geq \inf_{a \in S^{d-1} \cap \mathbb{R}_+^d, b \in \mathbb{R}} (1 - \pi)\Phi(b) + \pi\Phi(-\mathbb{E}[\bar{u}] - b)$$

where we remark that $\mathbb{E}[\bar{u}]$ depends on both $a$ and $b$. Defining $g(b) = -\mathbb{E}[\bar{u}] - b$, one notices that $g$ is concave, piecewise linear, and non-increasing. As $\mathbb{E}[u] \leq \theta\|p\|$, one finds that the function

$$\tilde{g}(b) = \begin{cases} -\theta\|p\|, & b \leq 0 \\ -\theta\|p\| - b, & b > 0 \end{cases}$$

is a minorant for $g(b)$ and so

$$\mathcal{E}_{\text{test}}^*[\hat{y}] \geq (1 - \pi)\Phi(b) + \pi\Phi(\tilde{g}(b)) \tag{34}$$

$$\geq \begin{cases} \pi\Phi(-\theta\|p\|), & b \leq 0 \\ (1 - \pi)/2, & b > 0 \end{cases}. \tag{35}$$

Applying the delocalization bound on $\|p\|$ then yields the lower bound

$$\liminf_{L \to \infty} \mathcal{E}_{\text{test}}^*[\hat{y}] \geq \min\left(\frac{1 - \pi}{2}, \; \pi\Phi(-C\ell^*)\right) > 0$$

where $C > 0$ was such that $\|p\| \leq CR/\sqrt{L}$. This completes the proof for the first set of assumptions of Theorem 1.

2. We now turn the the uniformity assumptions, namely when $\pi = 1/2$ and $v$ has a uniform distribution on its support. Setting $G(a, b)$ to be the objective function of (28) and

$$g(b, t) = 1/2\Phi(-b) + 1/2\Phi(b - t)$$

for $t \geq 0$, observe that $\mathbb{E}[g(b, u)] = G(a, b)$ where we again recall that $u$ depends on $a$. Following the same minimization over $b$ in the proof of Proposition 1, we see that

$$\inf_{b \in \mathbb{R}} g(b, t) = \frac{\Phi(-t/2)}{2}$$

and so

$$\mathcal{E}_{\text{test}}^*[\hat{y}] = \inf_{a \in S^{d-1} \cap \mathbb{R}_+^d, b \in \mathbb{R}} G(a, b) \geq \inf_{a \in S^{d-1} \cap \mathbb{R}_+^d} \frac{\mathbb{E}[\Phi(-u/2)]}{2} \geq \inf_{a \in S^{d-1} \cap \mathbb{R}_+^d} \frac{\Phi(-\mathbb{E}[u/2])}{2}$$

where the last inequality follows from Jensen's inequality as $\Phi(-x)$ is convex for $x \geq 0$. By monotonicity of $\Phi(-(\cdot))$ and since the choice $a = 1/\sqrt{L} \cdot \mathbf{1}_L$ maximizes $\mathbb{E}[u]$, we have

$$\mathcal{E}_{\text{test}}^*[\hat{y}] \geq \frac{\Phi(-\ell_L/2)}{2}$$

where $\ell_L = \theta R/\sqrt{L}$. Here, one notices that the right-hand-side corresponds to the optimal test error found for the pooled classifier in (25) when $\pi = 1/2$. Notably, the above is indeed an equality which is seen by evaluating $G$ as the previously considered values $(a, b)$. Hence, the uniformity assumptions reduce the optimal test error for the vectorized classifier to those of the pooled classifier. One then obtains an analogous result to (9).

# D  PROOF OF THEOREM 2

We detail in this Appendix the proof of Theorem 2. The proof builds on the following intermediary proposition, which gives a sufficient condition for vanishing test error, when the query weights $q$ are constrained in norm.

**Proposition 2.** *Consider the attention model* A *(5). For $\tau > 0$, let $\tau\mathbb{B}^d = \{x \in \mathbb{R}^d : \|x\| \leq \tau\}$, we consider the optimal test error $\mathcal{E}_{\text{test}}^*[A, \tau] = \inf_{q \in \tau\mathbb{B}^d, w \in \mathbb{R}^d, b \in \mathbb{R}} \mathcal{E}_{\text{test}}[A_{q,w,b}]$, restraining the minimization on $q$ to vectors of norm less than or equal to $\tau$. In the limit $L \to \infty$, $R = \Theta(1)$, allowing the signal $\theta$ to depend on $L$, suppose that*

$$\lim_{L \to \infty} \frac{\theta e^{\beta\tau\theta}}{L} \to \infty \tag{36}$$

*Then, the attention model* A *achieves an optimal test error of $\mathcal{E}_{\text{test}}^*[A] = 0$.*

*Proof.* We remind that from Lemma 2, for any $q, w, b$ the test error can be expressed as

$$\mathcal{E}_{\text{test}}[A_{q,w,b}] = (1 - \pi)\mathbb{P}\left(\|P_q^\perp w\| \cdot \|s_-\|z < b + \frac{\langle w, q\rangle}{\|q\|}\langle g, s_-\rangle\right) \tag{37}$$

$$+ \pi\mathbb{P}\left(\|P_q^\perp w\| \cdot \|s_+\|z < -b - \frac{\langle w, q\rangle}{\|q\|}\langle g, s_+\rangle - \langle\theta v, s_+\rangle\langle\xi, w\rangle\right). \tag{38}$$

where

$$s_- = \text{softmax}(\beta\|q\|g) \quad \text{and} \quad s_+ = \text{softmax}(\beta(\|q\|g + \langle q, \xi\rangle\theta v)),$$

$P_q^\perp = I_d - qq^\top/\langle q, q\rangle$, and $z \sim N(0, 1)$ is independent of $g \sim \mathcal{N}(0, I_L)$. The probability $\mathbb{P}$ bears jointly over the random variables $v, g, z$. To derive an upper bound on the optimal test error, we can consider the special case $q = \tau\xi$, $w = \xi$. The expression of the test error then simplifies to

$$\mathcal{E}_{\text{test}}[A_{q,w,b}] = (1 - \pi)\mathbb{P}(0 < b + \langle g, s_-\rangle) + \pi\mathbb{P}(0 < -b - \langle g, s_+\rangle - \langle\theta v, s_+\rangle). \tag{39}$$

Note that

$$\langle g, s_-\rangle = \frac{\sum\limits_{i \in [L]} g_i e^{\beta\tau g_i}}{\sum\limits_{i \in [L]} e^{\beta\tau g_i}} = \frac{\beta\tau e^{\frac{\beta^2\tau^2}{2}} + \frac{1}{\sqrt{L}}z_2}{e^{\frac{\beta^2\tau^2}{2}} + \frac{1}{\sqrt{L}}z_1}. \tag{40}$$

We have introduced the random variables

$$z_1 = \frac{\sum\limits_{i \in [L]} e^{\beta\tau g_i} - Le^{\frac{\beta^2\tau^2}{2}}}{\sqrt{L}}, \qquad z_2 = \frac{\sum\limits_{i \in [L]} g_i e^{\beta\tau g_i} - L\beta\tau e^{\frac{\beta^2\tau^2}{2}}}{\sqrt{L}}. \tag{41}$$

From the central limit theorem, $z_1, z_2$ converge in distribution to standard Gaussian variables. By the same token, one can rewrite

$$\langle g, s_+ + \theta v\rangle = \frac{\frac{e^{\beta\tau\theta} - 1}{L}B + \frac{\theta e^{\beta\tau\theta}}{L}A + \beta\tau e^{\frac{\beta^2\tau^2}{2}} + \frac{1}{\sqrt{L}}z_2}{\frac{e^{\beta\tau\theta} - 1}{L}A + e^{\frac{\beta^2\tau^2}{2}} + \frac{1}{\sqrt{L}}z_1}, \tag{42}$$

introducing the random variables

$$A = \sum\limits_{i \in [R]} e^{\beta\tau g_i}, \qquad B = \sum\limits_{i \in [R]} g_i e^{\beta\tau g_i}. \tag{43}$$

Using the change of variables $b = -\beta\tau - \tilde{b}/\sqrt{L}$ allows to reach

$$\mathcal{E}_{\text{test}}[A_{\tau\xi,\xi,-\beta\tau-\tilde{b}/\sqrt{L}}] = (1 - \pi)\mathbb{P}\left(\langle g, s_-\rangle - \beta\tau > \frac{\tilde{b}}{\sqrt{L}}\right) + \pi\mathbb{P}\left(\langle g + \theta v, s_+\rangle - \beta\tau < \frac{\tilde{b}}{\sqrt{L}}\right) \tag{44}$$

$$= (1 - \pi)\mathbb{P}\left(\frac{z_2 - \beta\tau z_1}{e^{\frac{\beta^2\tau^2}{2}} + \frac{1}{\sqrt{L}}z_1} > \tilde{b}\right) \tag{45}$$

$$+ \pi\mathbb{P}\left(\frac{\theta\frac{e^{\beta\tau\theta}}{\sqrt{L}}A + \frac{e^{\beta\tau\theta} - 1}{\sqrt{L}}(B - \beta\tau A) + z_2 - \beta\tau z_1}{\frac{e^{\beta\tau\theta} - 1}{L}A + e^{\frac{\beta^2\tau^2}{2}} + \frac{1}{\sqrt{L}}z_1} < \tilde{b}\right). \tag{46}$$

Let $\epsilon > 0$. We first focus on the first term, which one can bound as

$$
\mathbb{P}\left( \frac{z_2 - \beta\tau z_1}{e^{\frac{\beta^2\tau^2}{2}} + \frac{1}{\sqrt{L}}z_1} > \tilde{b} \right) \leq \mathbb{P}\left( z_2 - \beta\tau z_1 > \tilde{b}\left( e^{\frac{\beta^2\tau^2}{2}} - 1 \right) \right) + \mathbb{P}(|z_1| > \sqrt{L}). \tag{47}
$$

Let $M = \sqrt{2}\mathrm{erfc}(1 - \epsilon/8)$, and let

$$
\tilde{b}_1 = \frac{\sqrt{2\left( 1 + \beta^2\tau^2 - 2\beta^2\tau^2 e^{\beta^2\tau^2}(2e^{\beta^2\tau^2} - 1) \right)}}{-1 + e^{\frac{\beta^2\tau^2}{2}}}\mathrm{erfc}(1 - \epsilon/8). \tag{48}
$$

and

$$
\tilde{b}_2 = \sqrt{2\left( 1 + \beta^2\tau^2 - 2\beta^2\tau^2 e^{\beta^2\tau^2}(2e^{\beta^2\tau^2} - 1) \right)}\mathrm{erfc}(1 - \epsilon/8). \tag{49}
$$

We now fix $\tilde{b} = \max(\tilde{b}_1, \tilde{b}_2)$. Let $L_1$ be such that for $L \geq L_1$,

$$
\mathbb{P}\left( z_2 - \beta\tau z_1 > \tilde{b}\left( e^{\frac{\beta^2\tau^2}{2}} - 1 \right) \right) \leq 1 - \Phi\left( \frac{\tilde{b}\left( e^{\frac{\beta^2\tau^2}{2}} - 1 \right)}{\sqrt{2\left( 1 + \beta^2\tau^2 - 2\beta^2\tau^2 e^{\beta^2\tau^2}(2e^{\beta^2\tau^2} - 1) \right)}} \right) + \frac{\epsilon}{8},
$$
$$\tag{50}$$

$$
\mathbb{P}(|z_1| > M) \leq 2 - 2\Phi(M) + \frac{\epsilon}{8}. \tag{51}
$$

The existence of such $L_1$ is guaranteed by the convergence in distribution of $z_1, z_2$ to joint normal Gaussian variables. Then for any $L > \max(M, L_1)$,

$$
\mathbb{P}\left( \frac{z_2 - \beta\tau z_1}{e^{\frac{\beta^2\tau^2}{2}} + \frac{1}{\sqrt{L}}z_1} > \tilde{b} \right) \leq \frac{\epsilon}{2}. \tag{52}
$$

Turning to the other term,

$$
\mathbb{P}\left( \frac{\theta\frac{e^{\beta\tau\theta}}{\sqrt{L}}A + \frac{e^{\beta\tau\theta}-1}{\sqrt{L}}(B - \beta\tau A) + z_2 - \beta\tau z_1}{\frac{e^{\beta\tau\theta}-1}{L}A + e^{\frac{\beta^2\tau^2}{2}} + \frac{1}{\sqrt{L}}z_1} < \tilde{b} \right) \leq \mathbb{P}(z_2 - \beta\tau z_1 < -\tilde{b}) + \mathbb{P}(|z_1| \geq \sqrt{L})
$$
$$\tag{53}$$

$$
+ \mathbb{P}\left( A < \frac{2\tilde{b}(e^{\frac{\beta^2\tau^2}{2}} + 1) - \frac{e^{\beta\tau\theta}-1}{\sqrt{L}}B}{\theta\frac{e^{\beta\tau\theta}}{\sqrt{L}} - \frac{e^{\beta\tau\theta}-1}{\sqrt{L}}(\beta\tau + \tilde{b}/\sqrt{L})} \right)
$$
$$\tag{54}$$

Let $L_2$ be such that for $L \geq L_2$,

$$
\mathbb{P}\left( z_2 - \beta\tau z_1 > \tilde{b}\left( e^{\frac{\beta^2\tau^2}{2}} - 1 \right) \right) \leq 1 - \Phi\left( \frac{\tilde{b}}{\sqrt{2\left( 1 + \beta^2\tau^2 - 2\beta^2\tau^2 e^{\beta^2\tau^2}(2e^{\beta^2\tau^2} - 1) \right)}} \right) + \frac{\epsilon}{8},
$$
$$\tag{55}$$

$$
\mathbb{P}(|z_1| > M) \leq 2 - 2\Phi(M) + \frac{\epsilon}{8}. \tag{56}
$$

Finally,

$$
\mathbb{P}\left( A < \frac{2\tilde{b}(e^{\frac{\beta^2\tau^2}{2}} + 1) - \frac{e^{\beta\tau\theta}-1}{\sqrt{L}}B}{\theta\frac{e^{\beta\tau\theta}}{\sqrt{L}} - \frac{e^{\beta\tau\theta}-1}{\sqrt{L}}(\beta\tau + \tilde{b}/\sqrt{L})} \right) \leq \mathbb{P}\left( A < \frac{2\tilde{b}(e^{\frac{\beta^2\tau^2}{2}} + 1) + \frac{e^{\beta\tau\theta}-1}{\sqrt{L}}\sqrt{\theta}}{\theta\frac{e^{\beta\tau\theta}}{\sqrt{L}} - \frac{e^{\beta\tau\theta}-1}{\sqrt{L}}(\beta\tau + \tilde{b}/\sqrt{L})} \right) + \mathbb{P}(|B| > \sqrt{\theta}).
$$
$$\tag{57}$$

Now, note that

$$
\frac{\theta e^{\beta\tau\theta}}{L} \xrightarrow{L\to\infty} \infty \implies \theta \xrightarrow{L\to\infty} \infty. \tag{58}
$$

From Markov's inequality,

$$\mathbb{P}(|B| > \sqrt{\theta}) \le \frac{\mathbb{E}[|B|]}{\sqrt{\theta}}. \tag{59}$$

Let $L_3$ be such that for all $L \ge L_3$,

$$\mathbb{P}(|B| > \sqrt{\theta}) < \frac{\epsilon}{8}. \tag{60}$$

Finally, let us introduce the shorthand

$$h := \frac{2\tilde{b}(e^{\frac{\beta^2\tau^2}{2}} + 1) + \frac{e^{\beta\tau\theta}-1}{\sqrt{L}}\sqrt{\theta}}{\theta\frac{e^{\beta\tau\theta}}{\sqrt{L}} - \frac{e^{\beta\tau\theta}-1}{\sqrt{L}}(\beta\tau + \tilde{b}/\sqrt{L})}. \tag{61}$$

Remark that $h \to 0$ as $L \to \infty$. Let $L_4$ be such that for $L \le L_4$, $h < 1$. Using Mill's inequality,

$$\mathbb{P}(A < h) \le \mathbb{P}(e^{\beta\tau g_1} < h) \le \frac{1}{2}\frac{1}{\frac{1}{\beta\tau}|\log h|}e^{-\frac{1}{2\beta^2\tau^2}\log h^2}. \tag{62}$$

The right hand side tends to 0: let $L_5$ be such that for all $L \le L_5$, it is smaller than $\epsilon/8$. In conclusion, summarizing, for any $L \ge \max(M, L_1, L_2.L_3, L_4.L_5)$,

$$0 \le \mathcal{E}^*_{\text{test}}[A, \tau] \le \mathcal{E}_{\text{test}}\left[A_{\tau\xi,\xi,\sqrt{2(1+\beta^2\tau^2-2\beta^2\tau^2e^{\beta^2\tau^2}(2e^{\beta^2\tau^2}-1))}\text{erfc}(1-\epsilon/8)\max\left(1,\frac{1}{e^{\beta^2\tau^2/2}-1}\right)}\right] < \epsilon. \tag{63}$$

Thus,

$$\mathcal{E}^*_{\text{test}}[A] \xrightarrow{L\to\infty} 0. \tag{64}$$

$\square$

This concludes the proof of Proposition 2. We now prove Theorem 2.

*Proof.* Suppose

$$\liminf \frac{\theta}{\log L} > 0. \tag{65}$$

There then exist $C > 0, L_0$, such that for all $L \ge L_0$, $\theta > C\log L$. Then, setting

$$\tau = \frac{1}{C\beta}, \tag{66}$$

observe that

$$\frac{\theta e^{\beta\tau\theta}}{\sqrt{L}} \ge C\log L\sqrt{L} \xrightarrow{L\to\infty} \infty. \tag{67}$$

From proposition 2,

$$0 \le \mathcal{E}^*_{\text{test}}[A] \le \mathcal{E}^*_{\text{test}}[A, \tau] \xrightarrow{L\to\infty} 0 \tag{68}$$

$\square$

## E  PROOF OF THEOREM 3

**Output Weights and Bias.**

We prove in this appendix Theorem 3, which we summarized in the main text. We first give the full statement.

**Assumption 2.** *The loss function is of the form $\ell(z, y) = \ell^*(yz)$ for some convex function $\ell^*(\cdot)$. This assumption is in particular satisfied by the logistic and quadratic losses on $\mathbb{R} \times \{-1, +1\}$. We further denote $C(\ell) = -y\partial_z\ell(z, y)|_{z=0}$. For the logistic (resp. quadratic) loss, $C(\ell) = 1/2$ (resp. $C(\ell) = 1$).*

**Theorem 3** (Characterization of the query weights $q^{(2)}$ after two gradient steps). *Let $w^{(1)}, b^{(1)}$ be the readout weights and bias of the attention model A (5) at the end of step 2 of the training procedure detailed in subsection 3. In the asymptotic limit of Assumption 1, the summary statistics $b^{(1)}, \|w^1\|$ and $\langle w^1, \xi \rangle$ converge in probability to deterministic limits, given by*

$$b^{(1)} \xrightarrow[d\to\infty]{P} C(\ell)\eta_b(2\pi - 1), \tag{69}$$

*while*

$$\|w^{(1)}\| \xrightarrow[d\to\infty]{P} \gamma_1 := \eta_w C(\ell)\sqrt{\frac{1}{\alpha_0 L} + (\pi\theta R/L)^2}, \quad \langle w^{(1)}, \xi \rangle \xrightarrow[d\to\infty]{P} \gamma_2 := \eta_w C(\ell)\frac{\theta\pi R}{L}. \tag{70}$$

*Similarly, let $q^{(2)}$ denote the query weights at the end of step 3. The summary statistics $\|q^{(2)}\|$ and $\langle q^{(2)}, \xi \rangle$ converge in probability to the limits*

$$\|q^{(2)}\| \xrightarrow[d\to\infty]{P} \tfrac{\eta_q\beta}{L}\left[(L-1)\gamma_1^2\left((L-1)E_1^2 + \tfrac{E_3}{\alpha_0}\right) + \theta^2\left(R - \tfrac{R^2}{L}\right)\left(\gamma_2^2\tfrac{E_4}{\alpha_0} + 2\gamma_2^2 E_2(L-1)E_1\right) + \theta^4\gamma_2^2\left(R - \tfrac{R^2}{L}\right)^2 E_2^2\right]^{\frac{1}{2}}, \tag{71}$$

*and*

$$\langle \xi, q^{(2)} \rangle \xrightarrow[d\to\infty]{P} \gamma := -\frac{\eta_q\beta\gamma_2}{L}\left[(L-1)E_1 + \theta^2(R - R^2/L)E_2\right]. \tag{72}$$

*Here, $E_1, E_2, E_3, E_4$ are constants whose expressions are given in the proof.*

For this bias, the Law of Large Numbers yields

$$b^{(1)} = -\frac{\eta_b}{n_0}\sum_{i\le n_0} h_i(0, 0, 0)$$

$$= \frac{\eta_b}{n_0}\sum_{i\le n_0} C(\ell)y_i \xrightarrow{P} C(\ell)\eta_b(2\pi - 1)$$

since $\mathbb{E}[y] = \mathbb{P}(y = 1) - \mathbb{P}(y = -1) = 2\pi - 1$. Now, decomposing the noise $Z_i$ by

$$Z_i = \begin{bmatrix} s_i^\top \\ U_i^\top \end{bmatrix} \in \mathbb{R}^{L\times d} \quad U_i \in \mathbb{R}^{d\times L-1}, \quad s_i \in \mathbb{R}^d,$$

and setting

$$\mathbf{S} = \begin{bmatrix} s_1 & \cdots s_n \end{bmatrix} \quad \text{and} \quad \mathbf{y} = \begin{bmatrix} y_1 \\ \vdots \\ y_n \end{bmatrix},$$

we have

$$w^{(1)} = -\frac{\eta_w}{nL}\sum_{i\le n_0} h_i(0, 0, 0)X_i^\top \mathbf{1}$$

$$\stackrel{(d)}{=} C(\ell)\cdot\frac{\eta_w}{L}\left(\sqrt{L}\mathbf{S}\mathbf{y} + \sum_{i\le n_0} y_i v_i^\top \mathbf{1}\,\theta R\xi\right)$$

$$\asymp C(\ell)\eta_w\left(\frac{\mathbf{S}\mathbf{y}}{n_0\sqrt{L}} + \frac{\pi\theta R\xi}{L}\right) \tag{73}$$

where one applies the Law of Large Numbers for the last line above. Using the representation of (73), we obtain

$$\langle w^{(1)}, \xi \rangle \xrightarrow[d\to\infty]{P} \gamma_2$$

since $\langle \xi, \mathbf{S}\mathbf{y}\rangle/n_0 \sim \mathcal{N}(0, \frac{1}{n_0})$, and

$$\|w^{(1)}\| \asymp C(\ell)\eta_w \sqrt{\frac{\|\mathbf{S}\mathbf{y}\|^2}{n_0^2 L} + (\pi\theta R/L)^2} \xrightarrow[d\to\infty]{P} \gamma_1$$

as $\langle \mathbf{S}\mathbf{y}, \mathbf{S}\mathbf{y}\rangle/n_0 \sim \chi_d^2$.

**Query Weights.** Setting

$$c_\mu := h_\mu(0, w^{(1)}, b^{(1)}) \quad \text{and} \quad A_\mu := X_\mu^\top (I - \mathbf{1}\mathbf{1}^\top/L)X_\mu$$

we have

$$q^{(2)} = -\frac{\eta_q \beta}{n_0 L} \sum_{\mu \le n_0} c_\mu A_\mu w^{(1)}. \tag{74}$$

It will become clear that we require only the first and second moments of $c_\mu$ conditional on $y_\mu$ to characterize $\|q\|$ and $\langle q, \xi\rangle$ for large $n_0$ and $d$. Specifically, set

$$E_1 = \mathbb{E}[c_\mu], \quad E_2 = \mathbb{E}[c_\mu|y=1], \quad E_3 = \mathbb{E}[c_\mu^2], \quad E_4 = \mathbb{E}[c_\mu^2|y=1]. \tag{75}$$

Concretely, $c_\mu$ is given by

$$c_\mu = \frac{\mathrm{d}}{\mathrm{d}z}\ell(z, y_\mu)|_{z=m_\mu} \tag{76}$$

where

$$m_\mu = \langle w^{(1)}, \underbrace{X_\mu^\top \mathbf{1}/L}_{\bar{x}_\mu}\rangle + b^{(1)}$$

To find the distribution of $m_\mu$, we write

$$w^{(1)} = w_{-\mu}^{(1)} + \Delta_\mu, \qquad \Delta_\mu = \frac{C(\ell)\eta_w}{n_0 L}\langle y_\mu, X_\mu^\top \mathbf{1}\rangle, \tag{77}$$

where $w_{-\mu}^{(1)}$ is obtained from all samples except $\mu$ and is therefore independent of $X_\mu$. Substituting (77) gives the exact identity

$$m_\mu = \underbrace{\langle \bar{x}_\mu, w_{-\mu}^{(1)}\rangle}_{\text{noise term}} + \underbrace{\langle \bar{x}_\mu, \Delta_\mu\rangle}_{\text{self term}} + b^{(1)}$$

$$= C(\ell)\frac{\eta_w}{n_0 L}\frac{\|X_\mu^\top \mathbf{1}\|^2}{L}y_\mu + C(\ell)\frac{\eta_w}{n_0 L}\langle \bar{x}_\mu, S_{\text{rest}}\rangle + C(\ell)\eta_b(2\pi - 1),$$

with $S_{\text{rest}} := \sum_{j\neq\mu} y_j X_j^\top \mathbf{1}$ (independent of $X_\mu$). The above representations lends to the following conditional distributions:

$$m_\mu \mid \{y_\mu = -1\} \sim \mathcal{N}\left(C(\ell)\eta_b(2\pi-1) - \frac{C(\ell)\eta_w}{\alpha}, \frac{C(\ell)^2\eta_w^2}{\alpha L^2},\right)$$

$$m_\mu \mid \{y_\mu = +1\} \sim \mathcal{N}\left(C(\ell)\eta_b(2\pi-1) + C(\ell)\eta_w\left(\frac{1}{\alpha} + \frac{\theta^2 R^2}{L^2}\right), \frac{C(\ell)^2\eta_w^2}{\alpha L^2}\right). \tag{78}$$

From the above, we see that marginally $m_\mu$ is Gaussian mixture. Knowing the distributions $m_\mu|y_\mu$ and $y_\mu$ facilitates the computation of $E_1, \ldots, E_4$. This can easily be done to machine precision — such as via Gauss–Hermite quadrature as an example.

Returning to another piece of (74), set $b_\mu := \mathbf{1}_{y_\mu} \cdot ((2\mathbf{1}\mathbf{1}^\top/\sqrt{L} - I_L)v)_{[2:L]} \in \mathbb{R}^{L-1}$ and decompose the Gaussian noise $U_\mu$ by

$$U_\mu = [g_\mu \quad V_\mu], \quad g_\mu \in \mathbb{R}^d, \quad V_\mu \in \mathbb{R}^{d\times L-2}.$$

We then decompose the feature gradient $A_\mu$ by

$$
\begin{aligned}
A_\mu &= U_\mu U_\mu^\top + \theta^2 (R \cdot 1_{\{y_\mu=1\}} - R^2 \cdot 1_{\{y_\mu=1\}}/L)\xi\xi^\top + \theta U_\mu b_\mu \xi^\top + \theta \xi b_\mu^\top U_\mu^\top \\
&= g_\mu g_\mu^\top + V_\mu V_\mu^\top + \theta^2 (R \cdot 1_{\{y_\mu=1\}} - R^2 \cdot 1_{\{y_\mu=1\}}/L)\xi\xi^\top \\
&\qquad\qquad + \theta\sqrt{R \cdot 1_{\{y_\mu=1\}} - R^2 \cdot 1_{\{y_\mu=1\}}/L} \cdot (g_\mu\xi^\top + \xi g_\mu^\top) \\
&= g_\mu g_\mu^\top + V_\mu V_\mu^\top + \theta^2 h_\mu^2 \xi\xi^\top + \theta h_\mu (g_\mu\xi^\top + \xi g_\mu^\top)
\end{aligned}
$$

for

$$
h_\mu := \|b_\mu\| = 1_{\{y_\mu=1\}}\sqrt{R - R^2/L}.
$$

Now, set

$$
G = [g_1 \quad \cdots \quad g_n] \in \mathbb{R}^{d\times n}, \quad c = \begin{bmatrix} c_1 \\ \vdots \\ c_n \end{bmatrix}, \quad h = \begin{bmatrix} h_1 \\ \vdots \\ h_n \end{bmatrix},
$$

and let $\Lambda_x := \mathrm{diag}(x) \in \mathbb{R}^{k\times k}$ for $x \in \mathbb{R}^k, k \in \mathbb{N}$. Observe that

$$
\sum_{\mu=1}^n c_\mu V_\mu V_\mu^\top = \sum_{j=1}^{L-2} V_j \Lambda_c V_j^\top
$$

where — abusing notation — $V_j \overset{\text{iid}}{\sim} V \in \mathbb{R}^{d\times n}$ and $V$ has i.i.d. $\mathcal{N}(0,1)$ entries. Making a final decomposition of the noise:

$$
G = \begin{bmatrix} \tilde{g}_s^\top \\ \tilde{G}_u^\top \end{bmatrix}, \qquad V = \begin{bmatrix} \tilde{v}_s^\top \\ \tilde{V}_u^\top \end{bmatrix}, \qquad \tilde{g}_s, \tilde{v}_s \in \mathbb{R}^n, \quad \tilde{G}_u, \tilde{V}_u \in \mathbb{R}^{n\times d-1},
$$

we obtain

$$
q^{(2)} = -\frac{\eta_q\beta}{n_0 L}\left(\sum_{\mu\le n_0} c_\mu A_\mu\right) w^{(1)}
$$

$$
= -\frac{\eta_q\beta}{n_0 L}\left[ G\Lambda_c G^\top + \underbrace{V\Lambda_c V^\top}_{L-2 \text{ ind. copies}} + \theta^2\left(\sum_\mu h_\mu^2 c_\mu\right)\xi\xi^\top + \theta G\Lambda_h c\xi^\top + (\theta G\Lambda_h c\xi^\top)^\top \right] w^{(1)}
$$

$$
\overset{(d)}{=} -\frac{\eta_q\beta}{n_0 L}\left[ H_{w^{(1)}}\left(\|w^{(1)}\|\left(G\Lambda_c\tilde{g}_s + \underbrace{V\Lambda_c\tilde{v}_s}_{L-2 \text{ ind. copies}}\right) + \theta\cdot\xi^\top w^{(1)}\cdot G\Lambda_h c\right) \right.
$$

$$
\left. + \left(\theta\|w^{(1)}\|c^\top\Lambda_h g_s + \theta^2(\xi^\top w^{(1)})\sum_\mu c_\mu h_\mu^2\right)\xi \right]
$$

$$
= -\frac{\eta_q\beta}{n_0 L}\left[ H_{w^{(1)}}\left(\gamma_1\cdot\left(G\Lambda_c\tilde{g}_s + \underbrace{V\Lambda_c\tilde{v}_s}_{L-2 \text{ ind. copies}}\right) + \theta\cdot\gamma_2\cdot G\Lambda_h c\right) \right.
$$

$$
\left. + \left(\theta\cdot\gamma_1\cdot c^\top\Lambda_h g_s + \theta^2\cdot\gamma_2\cdot\sum_\mu c_\mu h_\mu^2\right)\xi \right].
$$

Since $\frac{1}{n_0}c^\top\Lambda_h g_s \overset{P}{\to} 0$ as $n_0 \to \infty$, we have

$$
q^{(2)} \asymp -\frac{\eta_q\beta}{L}\left[ \underbrace{\frac{1}{n_0}H_{w^{(1)}}\left(\gamma_1\cdot(G\Lambda_c\tilde{g}_s + \underbrace{V\Lambda_c\tilde{v}_s}_{L-2 \text{ ind. copies}}) + \theta\cdot\gamma_2\cdot G\Lambda_h c\right)}_{M} + \underbrace{\theta^2\cdot\gamma_2\cdot(R - R^2/L)\cdot E_2}_{N}\cdot\xi \right]
$$

$$
= -\frac{\eta_q\beta}{L}(M + N\cdot\xi)
$$

Therefore ,

$$\langle \xi, q^{(2)} \rangle \asymp -\frac{\eta_q \beta}{L} \cdot (\xi^\top M + N)$$

and

$$\|q^{(2)}\| \asymp \frac{\eta_q \beta}{L} \cdot \sqrt{M^\top M + 2N\xi^\top M + N^2}.$$

By rotational invariance of the isotropic Gaussian, we may take $\xi$ to be the first standard basis vector in the following derivations. We then have,

$$\xi^\top M \overset{(d)}{=} \frac{1}{n_0} \frac{w^{(1)\top}}{\|w^{(1)}\|} \left( \gamma_1 \cdot \underbrace{G\Lambda_c \tilde{g}_s}_{L-1 \text{ ind. copies}} + \theta \cdot \gamma_2 \cdot G\Lambda_h c \right)$$

$$= \frac{1}{n_0} \cdot w^{(1)\top} \underbrace{G\Lambda_c \tilde{g}_s}_{L-1 \text{ ind. copies}} + \theta \cdot \frac{\gamma_2}{\gamma_1} \cdot \underbrace{\frac{w^{(1)\top} G\Lambda_h c}{n_0}}_{\asymp 0}$$

$$\asymp \frac{1}{n_0} \cdot w_1^{(1)} \tilde{g}_s^\top \Lambda_c \tilde{g}_s$$

$$\asymp (L-1) \cdot \gamma_2 \cdot E_1.$$

This gives us the alignment

$$\langle \xi, q^{(2)} \rangle \xrightarrow[d \to \infty]{P} -\frac{\eta_q \beta \gamma_2}{L} \left[ (L-1)E_1 + \theta^2 (R - R^2/L)E_2 \right] = \gamma$$

as claimed.

Finally, to compute the magnitude of $q^{(2)}$, all that remains is to determine $M^\top M$. We have,

$$M^\top M \overset{(d)}{=} \frac{1}{n_0^2} \cdot \gamma_1^2 \cdot \sum_{1 \le i,j \le L-1} \tilde{v}_{i_s}^\top \Lambda_c V_i^\top V_j \Lambda_c \tilde{v}_{j_s} + \frac{2}{n_0^2} \cdot \theta \gamma_1 \gamma_2 \cdot c^\top \Lambda_h G^\top G\Lambda_c \tilde{g}_s$$

$$+ \frac{2}{n_0^2} \cdot \theta \gamma_1 \gamma_2 \cdot c^\top \Lambda_h G^\top \underbrace{V\Lambda_c \tilde{v}_s}_{L-2 \text{ ind. copies}} + \frac{1}{n_0^2} \cdot \theta^2 \gamma_2^2 \cdot c^\top \Lambda_h G^\top G\Lambda_h c.$$

Examining each term separately, note that by repeated application of the Law of Large Numbers we obtain the following:

$$\frac{1}{n_0^2} \cdot \gamma_1^2 \cdot \sum_{1 \le i,j \le L-1} \tilde{v}_{i_s}^\top \Lambda_c V_i^\top V_j \Lambda_c \tilde{v}_{j_s} = \frac{1}{n_0^2} \cdot \gamma_1^2 \cdot \sum_{i=1}^{L-1} \tilde{v}_{i_s}^\top \Lambda_c V_i^\top V_i \Lambda_c \tilde{v}_{i_s} + \frac{1}{n_0^2} \cdot \gamma_1^2 \cdot \sum_{i \ne j} \tilde{v}_{i_s}^\top \Lambda_c V_i^\top V_j \Lambda_c \tilde{v}_{j_s}$$

$$\asymp (L-1)\gamma_1^2 \cdot \left( \frac{1}{n_0^2} (\tilde{v}_s^\top \Lambda_c \tilde{v}_s)^2 + \frac{1}{n_0^2} \tilde{v}_s^\top \Lambda_c \tilde{V}_u \tilde{V}_u^\top \Lambda_c \tilde{v}_s + \frac{1}{n_0^2}(L-2)((\tilde{v}_s^\top \Lambda_c \tilde{v}_s)^2) \right)$$

$$\asymp (L-1)\gamma_1^2 \cdot \left( (L-1) \cdot E_1^2 + \frac{E_3}{\alpha} \right),$$

$$\frac{2}{n_0^2} \cdot \theta \gamma_1 \gamma_2 \cdot c^\top \Lambda_h G^\top G\Lambda_c \tilde{g}_s + \frac{2}{n_0^2} \cdot \theta \gamma_1 \gamma_2 \cdot c^\top \Lambda_h G^\top \underbrace{V\Lambda_c \tilde{v}_s}_{L-2 \text{ ind. copies}} \asymp 0,$$

and

$$\frac{1}{n_0^2} \cdot \theta^2 \gamma_2^2 \cdot c^\top \Lambda_h G^\top G\Lambda_h c \overset{(d)}{=} \theta^2 \gamma_2^2 \cdot \frac{\|\Lambda_h c\|^2}{n_0} \cdot \frac{g_1^\top g_1}{n_0}$$

$$\asymp \theta^2 \gamma_2^2 \cdot (R - R^2/L) \cdot \frac{E_4}{\alpha}.$$

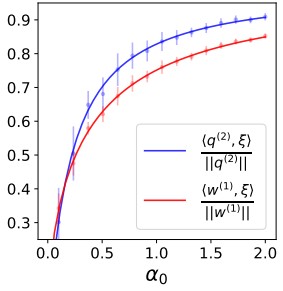

Figure 3: Cosine similarity between the signal $\xi$ and the query weights $q^{(2)}$ (blue) and readout weights $w^{(1)}$ (red) after step 3 of the training 3, for $L = 10, R = 3, \pi = 0.2, \theta = 6, \eta = 0.5$, and logistic loss $\ell$, as a function of the normalized number of samples $\alpha_0$. Solid lines: theoretical prediction of Theorem 3. Dots: numerical experiments in dimension $d = 1000$. Error bars represent one standard deviation over 10 trials.

Putting all the terms together, we obtain

$$M^\top M \asymp (L-1)\gamma_1^2 \cdot \left((L-1) \cdot E_1^2 + \frac{E_3}{\alpha}\right) + \theta^2 \gamma_2^2 \cdot (R - R^2/L) \cdot \frac{E_4}{\alpha}$$

and so

$$\|q^{(2)}\| = \frac{\eta_q \beta}{L} \cdot \left[(L-1)\gamma_1^2 \cdot \left((L-1) \cdot E_1^2 + \frac{E_3}{\alpha}\right) + \theta^2 \gamma_2^2 \cdot (R - R^2/L) \cdot \frac{E_4}{\alpha}\right.$$
$$\left. + 2N(L-1) \cdot \gamma_2 \cdot E_1 + N^2\right]^{1/2},$$

where we recall that

$$N = \theta^2 \cdot \gamma_2 \cdot (R - R^2/L) \cdot E_2.$$

This completes the precise characterization of the magnitude and $\xi$-alignment of the query vector $q^{(2)}$ where the definitions for the relevant constants $E_1, \ldots, E_4$ are found in (75).

### E.1 LARGE $\alpha_0$ BEHAVIOR

To conclude this appendix, we discuss the asymptotic behavior of the cosine similarities $\langle w^{(1)}, \xi \rangle / \|w^{(1)}\|$, $\langle q^{(2)}, \xi \rangle / \|q^{(2)}\|$ of the attention weights $w, q$ after one or two gradient step with the signal vector $\xi$, in the limit of large sample complexity $\alpha_0 \gg 1$. As we summarized in Corollary 1 in the main text, the cosine similarities rapidly approach 1 in absolute value as the sample complexity $\alpha_0$ is increased. We give here the full technical statement.

**Corollary 1** (Large $\alpha_0$ asymptotics). *Let $w^{(1)}, q^{(2)}$ be the readout weights and query weights of the attention model $A$ (5) at the end of step 3 of the training procedure detailed in subsection 3. In the asymptotic limit of Assumption 1, the cosine similarities $\langle w^{(1)}, \xi \rangle / \|w^{(1)}\|$, $\langle q^{(2)}, \xi \rangle / \|q^{(2)}\|$ converge in probability to deterministic limits $s_w, s_q$ from Theorem 3. We further assume that $\langle \xi, q^{(2)} \rangle \neq 0$ (72). When then further taking the limit $\alpha_0 \to \infty$, these limits admit the following asymptotic expansions*

$$s_w = 1 - \frac{L^2}{2\alpha_0(\pi\theta R)^2} + o\left(\frac{1}{\alpha_0}\right) \tag{79}$$

$$|s_q| = 1 - \frac{1}{2\alpha_0} \tag{80}$$

$$\cdot \frac{\frac{\eta_w^2 C(\ell)^2 (L-1)^2}{L}(\pi G_+^\infty + (1-\pi)G_-^\infty)^2 + (L-1)(\pi(G_+^\infty)^2 + (1-\pi)(G_-^\infty)^2) + \theta^2(R - \frac{R^2}{L})(G_+^\infty)^2}{\left((L-1)\pi G_+^\infty + (1-\pi)G_-^\infty + \theta^2(R - \frac{R^2}{L})G_+^\infty\right)^2} \tag{81}$$

$$+ o\left(\frac{1}{\alpha_0}\right) \tag{82}$$

*We denoted*

$$G_+^\infty = \ell' \left( C(\ell)\eta_b(2\pi - 1) + \frac{\eta_w \pi R^2 \theta^2}{2L^2}, 1 \right), \qquad G_-^\infty = \ell' \left( C(\ell)\eta_b(2\pi - 1), -1 \right). \qquad (83)$$

*The sign of $s_q$ is on the other hand given by that of*

$$- \left[ (L-1) + \theta^2 R(1 - {}^R/_L) \right] \pi G_+^\infty - (1-\pi) G_-^\infty. \qquad (84)$$

*Proof.* The proof of Corollary 1 follows straightforwardly from a $\alpha_0 \to \infty$ expansion of the expressions of Theorem 3. $\qquad \square$

Corollary 1 establishes how the weights of the attention model recover the signal vector $\xi$ when provided with sufficient data, at a rate of $1/\alpha_0$. The sign is given by an intricate but explicit condition (84) on all the parameters in the problem $\ell, \pi, \theta, R, L, \eta_b, \eta_w$, and can in certain cases be negative – signaling that the query vector $q$ detrimentally anti-aligns with the signal $\xi$. In order to avoid such a scenario, the condition (84) can offer some guideline for choosing the hyperparameters $\eta_b, \eta_w, \ell$. For example, for the logistic loss $\ell(y, z) = \log(1 + \exp(-yz))$, when $\pi < 1/2$ (resp. $\pi > 1/2$), choosing $\eta_b$ sufficiently large (resp. negative) ensures $s_q > 0$, namely that the query weights $q^{(2)}$ properly align with $\xi$ when $\alpha_0$ grows.

## F PROOF OF THEOREM 4

In this Appendix, we detail the proof of Theorem 4, which we summarized in the main text. We now present the full technical statement.

**Theorem 4** (Test errors after step 4). *Let $q$ denote the query weights after step 3 of the training procedure 3, and $\hat{w}, \hat{b}$ be the minimizers of the empirical risk (14) at step 4. We denote $\gamma = \langle q, \xi \rangle$. In the asymptotic limit of Assumption 1, the associated test error $\mathcal{E}_{\text{test}}$ (15) converges in probability to*

$$\mathcal{E}_{\text{test}}[\mathsf{A}] = (1-\pi)\mathbb{E}_{g,s_+,s_-} \left[ \Phi\left( \frac{\hat{b} + \langle g, s_-\rangle \mu_1}{\mu_3 \|s_-\|} \right) \right] + \pi \mathbb{E}_{g,s_+,s_-} \left[ \Phi\left( \frac{-\hat{b} - \langle \theta v, s_+\rangle \mu_2 - \langle g, s_+\rangle \mu_1}{\mu_3 \|s_+\|} \right) \right], \qquad (85)$$

*with $\mu_3 = \left[ \nu^2 + 1/1 - \gamma^2 \left( \mu_1^2 + \mu_2^2 - 2\gamma \mu_1 \mu_2 \right) - \mu_1^2 \right]^{\frac{1}{2}}$ .The description of the joint law of the finite-dimensional random variables $g, s_+, s_- \in \mathbb{R}^L$ is given in Lemma 1. The scalar statistics $\hat{b}, \mu_1, \mu_2, \nu$ are defined as the unique solutions of the following variational problem:*

$$\mu_1, \mu_2, \hat{b} = \underset{\mu_q, \mu_\xi, b}{\operatorname{argmin}} \, \phi(\mu_q, \mu_\xi, b) + \frac{\lambda}{2} \begin{bmatrix} \mu_q & \mu_\xi \end{bmatrix} \begin{bmatrix} 1 & \gamma \\ \gamma & 1 \end{bmatrix}^{-1} \begin{bmatrix} \mu_q \\ \mu_\xi \end{bmatrix}. \qquad (86)$$

*In the above display,*

$$\phi(\mu_q, \mu_\xi, b) := \mathbb{E}_{c_z, c_q, c_\xi, z, y} \left[ \ell(z^* + c_q \mu_q + c_\xi \mu_\xi + b, y) \right] + \frac{\lambda}{2}\nu^2, \qquad (87)$$

*where $c_z, c_q, c_\xi$ are scalar random variables whose joint law is detailed in Lemma 1, and $z \sim \mathcal{N}(0, 1)$. Finally,*

$$\nu^2 = \frac{1}{\lambda\chi}\mathbb{E}_{c_z, c_q, c_\xi, z, y} \left[ \frac{z^*(z^* - c_z \nu z)}{c_z^2} \right], \qquad \frac{1}{\alpha_1 \chi} = \mathbb{E}_{c_z, c_q, c_\xi, z, y} \left[ \frac{\ell_i''(z^*) c_z^2}{1 + \ell_i''(z^*) c_z^2 \chi} \right] + \lambda. \qquad (88)$$

*We used the shorthand $z^* := \operatorname{prox}_{c_z^2 \chi \ell(\cdot + c_q \mu_q + c_\xi \mu_\xi + b, y)}(c_z \nu z)$. Finally, the training loss $\mathcal{E}_{\text{train}}$ converges in probability to the minimizer of the right hand side of (86).*

Leveraging the equivalence between the attention model with zero query weights $\mathsf{A}_{0_d, w, b}$ (or, equivalently, vanishing softmax inverse temperature $\beta = 0$) with the pooled classifier $\mathsf{L}_{w,b}^{\text{pool}}$, a similar characterization for the latter can directly be deduced, as summarized in Theorem 5.

**Corollary 3** (Test error and training loss of $\mathsf{L}_{w,b}^{\text{pool}}$). *The training loss and test error if the pooled linear classifier $\mathsf{L}_{w,b}^{\text{pool}}$ (4) trained on the empirical minimization (16) converge i probability to limits $\mathcal{E}_{\text{train}}[\mathsf{L}^{\text{pool}}], \mathcal{E}_{\text{test}}[\mathsf{L}^{\text{pool}}]$, whose expressions fan be read from Theorem 4, if one sets $\beta = 0$.*

**Remark 1** (Length generalization). *Note that the characterization of the test error in Theorem 4 readily generalizes to the case where there exists a distribution shift between the training data and the testing data, when the model is tested on samples with a different length $L_{\text{test}} \neq L$ and sparsity $R_{\text{test}} \neq R$. The characterization (85) can be adapted to this case by using $L_{\text{test}}, R_{\text{test}}$ in the definition of the joint law of $g, s_+, s_-$ in Lemma 1, with the definitions of $\hat{b}, \mu_{1,2,3}$ otherwise unchanged. Fig. 5 (right) shows the $\alpha \to \infty$ error achieved by the attention model trained on $L = 4$ sequences with the square loss, and tested on different $L_{\text{test}}, R_{\text{test}}$, and shows that the model is capable of length generalization.*

## F.1 NOTATIONS AND ASSUMPTIONS

We take the following definition from Karoui (2018).

**Definition 1.** *Let*

$$X = (X_n(u) : n \in \mathbb{N}, u \in U_n), \qquad Y = (Y_n(u) : n \in \mathbb{N}, u \in U_n) \tag{89}$$

*be two families of nonnegative random variables, where $U_n$ is a possibly $n$-dependent parameter set. We write $X_n = O_{L_k}(Y_n)$ if*

$$\sup_{u \in U_n} \mathbb{E}[|X_n(u)|^k] = O(\sup_{u \in U_n} \mathbb{E}[|Y_n(u)|^k])$$

*where "O" refers to the classical big O-notation. That is, for two deterministic sequences $(a_n)$, $(b_n)$, we say $a_n = O(b_n)$ if there exists some $C > 0$ such that $a_n \leq C b_n$ for all $n$ sufficiently large.*

We make the following assumptions on the loss function $\ell$ (with the first argument denoted $z$):

(A1) $\ell$ is non-negative.

(A2) $\ell$ is convex in its first argument.

(A3) $\ell \in C^4$ in its first argument.

(A4) $\ell$ has bounded second–fourth derivatives.

(A5) $\ell$ is coercive, i.e.,

$$\lim_{|z| \to \infty} \ell(z; -1) + \ell(z; 1) = \infty.$$

**Remark 2.** *The above assumptions are satisfied for many natural choices of loss functions such as the quadratic loss, Huber loss, and logistic loss.*

**Remark 3.** *Having a bounded second derivative immediately implies the existence of a quadratic majorant of $\ell$ since for any $z \in \mathbb{R}$, a second-order Taylor expansion yields*

$$\ell(z) = \ell(0) + \ell'(0)z + \int_0^1 (1-t)\ell''(tz)z^2 \, dt \leq \ell(0) + \ell'(0)z + \frac{\|\ell''\|_\infty}{2} z^2.$$

## F.2 EMPIRICAL RISK MINIMIZATION

In what follows, we study the following learning problem:

$$\min_{w,b} \frac{1}{n_1} \sum_{i \in [n_1]} \ell(\langle f_i, w \rangle + b; y_i) + \frac{\lambda}{2} \|w\|^2, \tag{90}$$

where $\ell(z; y)$ is a loss function that is convex with respect to $z$. Let $w^*$ be the optimal weight vector and $b^*$ be the optimal bias for (90). Our goal is to characterize the following quantities:

$$\mu_1 = \langle q, w^* \rangle, \qquad \mu_2 = \langle \xi, w^* \rangle, \qquad \nu = \|P_{q,\xi}^\perp w^*\|, \tag{91}$$

and $b^*$, where $P_{q,\xi}^\perp$ denotes the projection onto the space orthogonal to $q$ and $\xi$. Having $b^*, \mu_1, \mu_2,$ and $\nu$ will provide for a full characterization of the test error due to Lemma 2.

**Remark 4.** *Recall we have assumed that $\|\xi\| = 1$ and at no loss of generality we also take $\|q\| = 1$, absorbing $\|q\|$ into $\beta$. Moreover, as a reminder, $\gamma = \langle \xi, q \rangle$. It is easy to check that for $\gamma \neq \pm 1$,*

$$P_{q,\xi}^\perp w = [q \quad \xi] \begin{bmatrix} 1 & \gamma \\ \gamma & 1 \end{bmatrix}^{-1} \begin{bmatrix} \mu_q \\ \mu_\xi \end{bmatrix}$$

*and that*

$$\|w\|^2 = [\mu_q \quad \mu_\xi] \begin{bmatrix} 1 & \gamma \\ \gamma & 1 \end{bmatrix}^{-1} \begin{bmatrix} \mu_q \\ \mu_\xi \end{bmatrix} + \|P_{q,\xi}^\perp w\|^2.$$

*for a weight vector $w$ with*

$$\mu_q = \langle q, w \rangle, \qquad \mu_\xi = \langle \xi, w \rangle.$$

From Lemma 1, we can rewrite the feature vectors $\{f_i\}$ as

$$f_i = c_{q,i} q + c_{\xi,i} \xi + c_{z,i} P_{q,\xi}^\perp z_i,$$

where $\{c_{q,i}, c_{\xi,i}, c_{z,i}\}_{i \leq n_1}$ are scalar random variables that are independent of the isotropic Gaussian vectors $\{z_i\}_{i \leq n_1}$. We write the joint law of $c_{q,i}, c_{\xi,i}, c_{z,i}$ as

$$c_{q,i}, c_{\xi,i}, c_{z,i} \sim \begin{cases} \mathcal{P}_+(c_q, c_\xi, c_z), & \text{if } y_1 = 1 \\ \mathcal{P}_-(c_q, c_\xi, c_z), & \text{if } y_1 = -1 \end{cases}.$$

The exact specification of the joint distributions are given in Lemma 1. Specifically,

$$\mathcal{P}_+(c_q, c_\xi, c_z): \qquad c_q = \langle g, s_+ \rangle - \frac{\gamma \|s_+\| z_0}{\sqrt{1 - \gamma^2}}, \qquad c_\xi = \langle \theta v, s_+ \rangle + \frac{\|s_+\| z_0}{\sqrt{1 - \gamma^2}}, \qquad c_z = \|s_+\| \tag{92}$$

$$\mathcal{P}_-(c_q, c_\xi, c_z): \qquad c_q = \langle g, s_- \rangle - \frac{\gamma \|s_-\| z_0}{\sqrt{1 - \gamma^2}}, \qquad c_\xi = \frac{\|s_-\| z_0}{\sqrt{1 - \gamma^2}}, \qquad c_z = \|s_-\|. \tag{93}$$

With this new decomposition of the feature vectors, the empirical risk minimization of (90) splits into (i) a three scalar variable problem of $\mu_q, \mu_\xi$ and $b$, governing the $q, \xi$ plane and a bias, and (ii) a $(d-2)$-dimensional sub-problem determining the orthogonal component to $\text{span}\{q, \xi\}$. The next display formalizes this sequential optimization problem:

$$\min_{\mu_q, \mu_\xi, b} \phi_d(\mu_q, \mu_\xi, b) + \frac{\lambda}{2} [\mu_q \quad \mu_\xi] \begin{bmatrix} 1 & \gamma \\ \gamma & 1 \end{bmatrix}^{-1} \begin{bmatrix} \mu_q \\ \mu_\xi \end{bmatrix},$$

where

$$\phi_d(\mu_q, \mu_\xi, b) := \min_{x \in \mathbb{R}^{d-2}} \frac{1}{n_1} \sum_{i \in [n_1]} \ell(\langle c_{z,i} z_i, x \rangle + c_{q,i} \mu_q + c_{\xi,i} \mu_\xi + b; y_i) + \frac{\lambda}{2} \|x\|^2, \tag{94}$$

and $\{z_i\}_{i \leq n_1}$ is a collection of $(d-2)$-dimensional, isotropic, normal random vectors.

Henceforth, our goal is to characterize the asymptotic limit of $\phi(\mu_q, \mu_\xi, b)$ and $\nu^2 = \|x^*\|^2$, where $x^*$ denotes the optimal solution to (94). Since $x^*$ is a stationary point, we must have

$$x^* = -\frac{1}{n_1 \lambda} \sum_{i \in [n_1]} \ell'(\langle c_{z,i} z_i, x^* \rangle + c_{q,i} \mu_q + c_{\xi,i} \mu_\xi + b; y_i)(c_{z,i} z_i).$$

Thus,

$$\nu^2 = \|x^*\|^2 = -\frac{1}{n_1 \lambda} \sum_{i \in [n_1]} \ell'(\langle c_{z,i} z_i, x^* \rangle + c_{q,i} \mu_q + c_{\xi,i} \mu_\xi + b; y_i)\langle c_{z,i} z_i, x^* \rangle. \tag{95}$$

In the following, we will denote

$$\epsilon_i := c_{q,i} \mu_q + c_{\xi,i} \mu_\xi + b, \qquad \qquad \ell_i(u + \epsilon_i) := \ell(u + \epsilon_i; y_i). \tag{96}$$

to elicit parallels between our derivations and those present in Karoui (2018). For simplicity of notation, we further write

$$\tilde{f}_i = c_{z,i} z_i$$

## F.3 LEAVE-ONE-OUT: DETERMINISTIC ANALYSIS

The key probabilistic structure in our problem is that different feature vectors are independent. This naturally prompts us to consider a leave-one-out analysis. We first need to introduce some notation. From this point forward $x$ is in $\mathbb{R}^{d-2}$. Let

$$\Phi_d^* := \min_x F_d^*(x) := \min_x \frac{1}{n_1} \sum_{i \in [n_1]} \ell_i(\langle \tilde{f}_i, x \rangle + \epsilon_i) + \frac{\lambda}{2}\|x\|^2 \qquad x_d^* = \arg\min_x F_d^*(x)$$

$$\Phi_{d,\backslash i}^* := \min_x F_{d,\backslash i}^*(x) := \min_x \frac{1}{n_1} \sum_{j \neq i} \ell_j(\langle \tilde{f}_j, x \rangle + \epsilon_j) + \frac{\lambda}{2}\|x\|^2 \quad x_{d,\backslash i}^* = \arg\min_x F_{d,\backslash i}^*(x)$$

denote the optimal values and the optimizing solutions of the original optimization problem and its leave-one-out version, respectively. Going forward, we will often omit the $d$-dependence of these quantities to alleviate the notation.

### F.3.1 LEAVE-ONE-OUT ANALYSIS

A key step in the following consists in constructing a close approximation $\tilde{x}_i$ of $x^*$, with simpler distributional properties. To that end, we introduce the surrogate optimization problem:

$$\widetilde{\Phi}_{d,i} := \Phi_{d,\backslash i}^* + \min_x \widetilde{F}_{d,i}(x), \qquad \tilde{x}_{i,d} := \arg\min_x \widetilde{F}_{d,i}(x)$$

where

$$\widetilde{F}_{d,i}(x) := \left\{ \frac{1}{n_1}\ell_i(\langle \tilde{f}_i, x \rangle) + \frac{1}{2}(x - x_{\backslash i}^*)^\top H_{\backslash i}(x - x_{\backslash i}^*) \right\} \tag{97}$$

and

$$H_{\backslash i} := \frac{1}{n_1} \sum_{j \neq i} \ell_j''(\langle \tilde{f}_j, x_{\backslash i}^* \rangle + \epsilon_j)\tilde{f}_j \tilde{f}_j^\top + \lambda I$$

is the (leave-one-out) Hessian matrix. Heuristically, this surrogate problem may be viewed as a quadratic approximation of $\Phi^*$ in the vicinity of $x_{\backslash i}^*$. It is straightforward to verify the following lemma.

**Lemma 3.** *Let $\mathcal{M}_i(x;\gamma)$ denote the Moreau envelope of $\ell_i(x)$, i.e.,*

$$\mathcal{M}_i(x;\gamma) := \min_z \ell_i(z) + \frac{(x-z)^2}{2\gamma}$$

*and let*

$$\mathrm{Prox}_i(x;\gamma) := \arg\min_z \ell_i(z) + \frac{(x-z)^2}{2\gamma}$$

*be the corresponding proximal operator. Then it holds that*

$$\tilde{r}_i := \langle \tilde{f}_i, \tilde{x}_i \rangle + \epsilon_i = \mathrm{Prox}_i(\tilde{r}_{i,\backslash i}; \gamma_i), \tag{98}$$

*where $\tilde{r}_{i,\backslash i} := \langle \tilde{f}_i, x_{\backslash i}^* \rangle + \epsilon_i$ and*

$$\gamma_i := \frac{1}{n_1}\tilde{f}_i^\top H_{\backslash i}^{-1}\tilde{f}_i. \tag{99}$$

*Moreover,*

$$\tilde{x}_i = x_{\backslash i}^* - \frac{1}{n_1}\ell_i'(\tilde{r}_i)H_{\backslash i}^{-1}\tilde{f}_i \tag{100}$$

*and*

$$\widetilde{\Phi}_i = \Phi_{\backslash i}^* + \frac{1}{n_1}\mathcal{M}_i(\langle \tilde{f}_i, x_{\backslash i}^* \rangle).$$

**Remark 5.** *Let $x = \mathrm{Prox}(c;\gamma)$. It is often convenient to recall the following identity:*

$$\ell'(x) + \frac{x-c}{\gamma} = 0. \tag{101}$$

### F.3.2 ON THE BOUNDEDNESS OF $\ell'$

A key technical difference with the closely related analysis of Karoui (2018) lies in the assumption made therein that $\ell'$ is bounded. We would like the present results to hold for the quadratic loss in particular, which does not satisfy this assumption. The following lemma bridges this gap by showing how the optimizer of the inner problem using loss $\ell$ coincides, with large probability, with that of a modified loss with bounded first derivative.

**Definition 2.** *Given $I > 0$, we define the clipped loss $\ell_{\text{clip}}(\cdot, y) : \mathbb{R} \mapsto \mathbb{R}$ as follows:*

1. *$\ell_{\text{clip}} \in C_b^{4}$[1] and convex*

2. *$\ell_{\text{clip}}(z) = \ell(r)$ for $z \in [-I, I]$*

3. *Letting $M = \sup_{z \in [-I, I]} |\ell'(z)|$, we require that $\|\ell'_{\text{clip}}\|_\infty \leq 2M$*

4. *we further require that $\ell_{\text{clip}} \leq \ell$.*

The construction given for $\ell_{\text{clip}}$ in definition 2 can be achieved in the following manner. Consider the "bump function"

$$\psi(t) = \begin{cases} \exp\left(\frac{1}{t(t-1)}\right), & \text{if } t \in (0, 1) \\ 0, & \text{else} \end{cases}$$

and, fixing a $\iota > 0$, define $\eta : \mathbb{R} \to [0, 1]$ by

$$\eta(z) = \begin{cases} 0, & z \leq I \\ \frac{\int_0^{(z-I)/\iota} \psi(t)\, dt}{\int_0^1 \psi(t)\, dt}, & z \in (I, I + \iota), \\ 1, & z \geq I + \iota \end{cases}$$

Note that $\eta \in C^\infty$ with bounded derivatives of all orders. Now, consider the left and right linear extensions of $\ell$,

$$L_-(z) = \ell(-I) + \ell'(-I)(z + I), \qquad L_+(z) = \ell(I) + \ell'(I)(z - I),$$

which allow us to define $\ell_{\text{clip}}$ as the piecewise function

$$\ell_{\text{clip}}(z) = \begin{cases} L_-(z), & z \leq -I - \iota \\ (1 - \eta(-z))\ell(z) + \eta(-z)L_-(z), & z \in (-I - \iota, -I) \\ \ell(z), & z \in [-I, I] \\ (1 - \eta(z))\ell(z) + \eta(z)L_+(z), & z \in (I, I + \iota) \\ L_+(z), & z \geq I + \iota \end{cases}$$

The prescribed properties of definition 2 are then easily verified from basic calculus.

**Lemma 4** (Clipped loss derivative). *Recall that*

$$x^* = \arg\min_x \frac{1}{n_1} \sum_{i \in [n_1]} \ell_i(\langle \tilde{f}_i, x \rangle + \epsilon_i) + \frac{\lambda}{2} \|x\|. \tag{102}$$

*For a given $\delta \in (0, 1)$, let*

$$\mathfrak{R}^2 := \frac{2}{\lambda} \mathbb{E}\left[\ell(\epsilon; y)\right], \quad I := (1 + \mathfrak{R})\sqrt{2 \log \frac{2n_1}{\delta}} + 1 + \sqrt{\mu_q^2 + \frac{(\mu_q \gamma + \mu_\xi)^2}{1 - \gamma^2}} + |b| + \sqrt{L\theta^2} \tag{103}$$

*where $\epsilon \sim c_q \mu_q + c_\xi \mu_\xi + b$. Define*

$$x^*_{\text{clip}} = \arg\min_x \frac{1}{n_1} \sum_{i \in [n_1]} \ell_{\text{clip},i}(\langle \tilde{f}_i, x \rangle + \epsilon_i) + \frac{\lambda}{2} \|x\|. \tag{104}$$

*Then, with probability at least $1 - \delta$,*

$$x^* = x^*_{\text{clip}}. \tag{105}$$

---

[1] Four times differentiable with continuous and bounded derivatives.

*Proof.* The strategy consists in controlling the supremum $\sup_{i \in [n_1]} |r_i^{\text{clip}}|$ of the residuals $r_i^{\text{clip}} = z^* [z^* - c_z \boldsymbol{\nu} Z]$. Since by construction $\|\ell'_{\text{clip}}\|_\infty < C\text{polyLog}(n_1)$, one can apply the results of Karoui (2018) to the clipped problem, showing that

$$|r_i^{\text{clip}}| \leq |\text{Prox}_{\text{clip}}(\langle x_{\text{clip},\backslash i}^*, \tilde{f}_i \rangle + \epsilon_i, \gamma_i)| + \delta^{(1)} \leq |\mathfrak{g}_i| \, \|x_{\text{clip},\backslash i}^*\| + \delta^{(1)} + |\epsilon_i| \tag{106}$$

using the contractivity of the proximal operator. We used the shorthand $\mathfrak{g}_i = \left\langle x_{\text{clip},\backslash i}^* / \|x_{\text{clip},\backslash i}^*\|, \tilde{f}_i \right\rangle$. Note that from Karoui (2018), $\delta^{(1)} := \sup_i |r_i^{\text{clip}} - \text{Prox}_{\text{clip}}(\langle x_{\text{clip},\backslash i}^*, \tilde{f}_i \rangle + \epsilon_i, \gamma_i)| = O_{L_k}\left(\frac{\text{polyLog}(n_1)}{\sqrt{n_1}}\right)$. From the identity $F_{\text{clip},\backslash i}(x_{\text{clip},\backslash i}^*) \leq F_{\text{clip},\backslash i}(0)$, one can bound

$$\|x_{\text{clip},\backslash i}^*\|^2 \leq \frac{2}{\lambda n_1} \sum_{j \neq i} \ell_{\text{clip}}(\epsilon_j; y_j) \leq \mathfrak{R}^2 + \delta^{(2)}, \tag{107}$$

with $\delta^{(2)} = O_{L_2}\left(\frac{\text{polyLog}(n_1)}{\sqrt{n_1}}\right)$. Using the identity $|\sqrt{1+x} - 1| \leq |x|$, we have $\|x_{\text{clip},\backslash i}^*\| \leq \mathfrak{R} + |\delta^{(2)}|$. Summarizing,

$$\sup_i |r_i^{\text{clip}}| \leq (\sup_i |\mathfrak{g}_i|)(\mathfrak{R} + |\delta^{(2)}|) + \delta^{(1)} + \sup_i |\epsilon_i|. \tag{108}$$

For $n_1$ large enough, from Markov's inequality,

$$\mathbb{P}[\delta^{(1)} > 1] \leq \frac{\delta}{6}, \qquad\qquad \mathbb{P}[|\delta^{(2)}| > 1] \leq \frac{\delta}{6}. \tag{109}$$

We now need to control the term $\sup_i |\epsilon_i|$. From (92), for any given and *fixed* $\mu_\xi, \mu_q, b$ and remembering $\|s_\pm\| \leq 1$, one can bound

$$|\epsilon_i| \leq a_\epsilon \|g_i\|_1 + b_\epsilon |z_{0,i}| + c_\epsilon \tag{110}$$

with

$$a_\epsilon = |\mu_q|, \qquad b_\epsilon = \left| \mu_q \frac{\gamma}{\sqrt{1-\gamma^2}} + \mu_\xi \frac{1}{\sqrt{1-\gamma^2}} \right|, \qquad c_\epsilon = |b| + \sqrt{L\theta^2}. \tag{111}$$

We remind that all entries of $g \in \mathbb{R}^L$, alongside with $z_0$, are normal-distributed. Then, for any $h$, using an union bound

$$\mathbb{P}\left[ \sup_i |\epsilon_i| \geq \sqrt{a_\epsilon^2 + b_\epsilon^2} h + c_\epsilon \right] \leq \sum_{i \in [n_1]} \mathbb{P}\left[ a_\epsilon \|g_i\|_1 + b_\epsilon |z_{0,i}| \geq \sqrt{a_\epsilon^2 + b_\epsilon^2} h \right]. \tag{112}$$

Examining more closely the summand $\mathbb{P}[a_\epsilon \|g_i\|_1 + b_\epsilon |z_{0,i}| \geq h]$, one has

$$\mathbb{P}\left[ a_\epsilon \|g_i\|_1 + b_\epsilon |z_{0,i}| \geq \sqrt{a_\epsilon^2 + b_\epsilon^2} h \right] \leq \sum_{s \in \{-1,+1\}^{L+1}} \mathbb{P}\left[ a_\epsilon s_1 g_{i,1} + \cdots + b_\epsilon s_{L+1} z_{0,i} \geq \sqrt{a_\epsilon^2 + b_\epsilon^2} h \right] \tag{113}$$

from a coarse union bound, remarking that the left hand side appears in the right hand side sum. Now that one has ridden of the absolute value, observe that each term in the summand is distributed as $\mathcal{N}(0, \sqrt{a_\epsilon^2 + b_\epsilon^2})$. Thus,

$$\Pr\left[ \sup_i |\epsilon_i| \geq \sqrt{a_\epsilon^2 + b_\epsilon^2} h + c_\epsilon \right] \leq 2^{L+1} n_1 \frac{e^{-\frac{1}{2}h^2}}{h}. \tag{114}$$

In particular, for $h = \sqrt{2 \log n_1}$,

$$\Pr\left[ \sup_i |\epsilon_i| \geq \sqrt{a_\epsilon^2 + b_\epsilon^2} \sqrt{2 \log n_1} + c_\epsilon \right] \leq 2^{L+1} \frac{1}{\sqrt{2 \log n_1}}. \tag{115}$$

Let us again suppose $n_1$ is large enough so that this probability is smaller than $\delta/6$.

Thus, for $n_1$ large enough, the probability of the complementary event of $\Delta = \{\delta^{(1)} < 1\} \cap \{\delta^{(2)} < 1\} \cap \{\sup_i |\epsilon_i| < \sqrt{a_\epsilon^2 + b_\epsilon^2}\sqrt{2\log n_1} + c_\epsilon\}$ is bounded as $\mathbb{P}[\overline{\Delta}] \leq \delta/2$. Now, for any $t \geq 2 + \sqrt{a_\epsilon^2 + b_\epsilon^2}\sqrt{2\log n_1} + c_\epsilon$

$$\mathbb{P}[\sup_i |r_i^{\text{clip}}| > t] \leq \mathbb{P}\left[\sup_i |g_i| > \frac{t - \delta^{(1)}}{\mathfrak{R} + |\delta^{(2)}|}\right] \tag{116}$$

$$\leq \mathbb{P}\left[\left\{\sup_i |g_i| > \frac{t - \delta^{(1)} - \sup_i |\epsilon_i|}{\mathfrak{R} + |\delta^{(2)}|}\right\} \cap \Delta\right] + \frac{\delta}{2} \tag{117}$$

$$\leq \mathbb{P}\left[\sup_i |g_i| > \frac{t - 1 - (\sqrt{a_\epsilon^2 + b_\epsilon^2}\sqrt{2\log n_1} + c_\epsilon)}{\mathfrak{R} + 1}\right] + \frac{\delta}{2} \tag{118}$$

$$\leq \sum_{i \in [n_1]} \mathbb{P}\left[|g_i| > \frac{t - 1 - (\sqrt{a_\epsilon^2 + b_\epsilon^2}\sqrt{2\log n_1} + c_\epsilon)}{\mathfrak{R} + 1}\right] + \frac{\delta}{2} \tag{119}$$

$$\leq n_1 e^{-\frac{1}{2}\left(\frac{t - 1 - (\sqrt{a_\epsilon^2 + b_\epsilon^2}\sqrt{2\log n_1} + c_\epsilon)}{(1 + \mathfrak{R})}\right)^2} + \frac{\delta}{2} \tag{120}$$

where the last line follows by Mill's inequality. In particular,

$$\mathbb{P}\left[\sup_i |r_i^{\text{clip}}| > (1 + \mathfrak{R})\sqrt{2\log\frac{2n_1}{\delta}} + 1 + \sqrt{a_\epsilon^2 + b_\epsilon^2}\sqrt{2\log n_1} + c_\epsilon\right] \leq \delta. \tag{121}$$

The last step of the proof comes from the simple observation that with probability at least $1 - \delta$, for all $i \in [n_1]$, $\ell_i(r_i) = \ell_{\text{clip},i}(r_i)$, and so under this event we have

$$-\lambda x^* = \frac{1}{n_1}\sum_{i \in [n_1]} \ell_i(r_i) = \frac{1}{n_1}\sum_{i \in [n_1]} \ell_{\text{clip},i}(r_i). \tag{122}$$

Therefore, $x^*$ satisfies the stationarity condition for the clipped problem. By uniqueness of the minimizer $x_{\text{clip}}^*$, we have

$$x^* = x_{\text{clip}}^* \tag{123}$$

in this event. $\qquad\square$

A consequence of Lemma 4 is that one can assume, without loss of generality up to an event of probability $\delta$, that the first derivative $\ell'$ is bounded. More precisely,

$$\|\ell'\|_\infty = O\left(\text{polyLog}(n_1)\right). \tag{124}$$

This enables in particular the borrowing of a number of results from Karoui (2018), where such an assumption is leveraged. Henceforth, we worth under the $(1 - \delta)$-probability event where $x^* = x_{\text{clip}}^*$ and work strictly with the clipped loss $\ell_{\text{clip}}$, however we omit notation and simply write $\ell$ for simplicity.

### F.3.3 CONCENTRATION RESULTS

We first introduce and recall several quantities of importance in this section. For $i, j \in [n_1]$, we write

$$r_i = \langle \tilde{f}_i, x_i^* \rangle + \epsilon_i, \quad \tilde{r}_{j,i} = \langle \tilde{f}_j, \tilde{x}_i \rangle + \epsilon_j, \quad \tilde{r}_{j,\backslash i} = \langle \tilde{f}_j, x_{\backslash i}^* \rangle + \epsilon_j.$$

The following lemma establishes that the introduced surrogate estimator $\tilde{x}_i$ constitutes a good approximation of the full minimizer $x^*$ as well as further concentration results.

**Lemma 5** (Approximation by surrogate estimator). *We have, for any $k$,*

$$\sup_{i \in [n_1]} \|x^* - \tilde{x}_i\| = O_{L_k}\left(\frac{\text{polyLog}(n_1)}{n_1}\right) \quad \text{and} \quad \sup_{i \in [n_1]} \|x_{\backslash i}^* - \tilde{x}_i\| = O_{L_k}\left(\frac{1}{\sqrt{n_1}}\right), \tag{125}$$

*Moreover,*

$$\text{Var}(\|x^*\|^2) = O\left(\frac{\text{polyLog}(n_1)}{n_1}\right). \tag{126}$$

*Furthermore, at the level of the residuals, one has the bounds*

$$\sup_{i \in [n_1]} |r_i - \tilde{r}_i| = O_{L_k}\left(\frac{\text{polyLog}(n_1)}{\sqrt{n_1}}\right) \tag{127}$$

*Proof.* The proof follows directly from Lemmas C.2, Theorem C.6 and Proposition C.7 of Karoui (2018). The statement on the residuals corresponds to Theorem 2.2 of the same work. □

The lemma thus shows that the squared norm $\|x^*\|^2$ concentrates. We denote in the following by $\nu^2 := \mathbb{E}\left[\|x^*\|^2\right]$ its limiting value. The statement on the residual can be further complemented by the following lemma, which covers the off-diagonal terms.

**Lemma 6.** *We further have*

$$\sum_{j \neq i} (r_j - \tilde{r}_{j,i})^2 = O_{L_k}\left(\frac{\text{polyLog}(n_1)}{n_1}\right), \tag{128}$$

*where we write $\tilde{r}_{j,i} = \langle \tilde{f}_j, \tilde{x}_i \rangle + \epsilon_j$.*

*Proof.* Denoting $F_{\backslash i} \in \mathbb{R}^{(n-1) \times d}$ the matrix with rows $\{\tilde{f}_j\}_{j \neq i}$, we need to control

$$\sum_{j \neq i} (r_j - \tilde{r}_{j,i})^2 = \|F_{\backslash i}(x^* - \tilde{x}_i)\|^2 \leq \|F_{\backslash i}\|^2 O_{L_k}\left(\frac{\text{polyLog}(n_1)}{n_1^2}\right) = O_{L_k}\left(\frac{\text{polyLog}(n_1)}{n_1}\right), \tag{129}$$

using Lemma 5 to control $\|x^* - \tilde{x}_i\|$. □

**Lemma 7** (On $\gamma_i$). *We have*

$$\sup_{i \in [n_1]} |\gamma_i - c_{z,i}^2 \chi| = O_{L_k}\left(\frac{1}{\sqrt{n_1}}\right) \quad \text{where} \quad \chi = \frac{1}{n_1} \text{tr}[H^{-1}]. \tag{130}$$

*We called*

$$H := \frac{1}{n_1} \sum_{j \in [n_1]} \ell_j''(r_j) \tilde{f}_j \tilde{f}_j^\top + \lambda I \tag{131}$$

*the full Hessian.*

*Proof.* This follows from Corollary D.7 and Lemma E.4 in Karoui (2018). □

**Lemma 8** ($\Phi^*$ concentrates). *We have*

$$\text{Var}[\Phi^*] = O\left(\frac{\text{polyLog}(n_1)}{n_1}\right) \tag{132}$$

*Proof.* Appealing to the Efron-Stein lemma, we have

$$\text{Var}[\Phi^*] \leq \sum_{i \in [n_1]} \mathbb{E}\left[(F^*(x^*) - F_{\backslash i}(x_{\backslash i}^*))^2\right] \tag{133}$$

The summand can be controlled as

$$\mathbb{E}\left[(F^*(x^*) - F_{\backslash i}(x_{\backslash i}^*))^2\right] \leq 2\mathbb{E}\left[\left(F_{\backslash i}^*(x^*) - F_{\backslash i}(x_{\backslash i}^*)\right)^2\right] + \frac{2}{n_1^2}\mathbb{E}\left[\ell_i(r_i)^2\right] \tag{134}$$

We first control the second term.

$$\frac{1}{n_1^2}\mathbb{E}\left[\ell_i(r_i)^2\right] \leq \frac{1}{n_1^2}2\left(\ell_i(0)^2 + \|\ell'\|_\infty^2\mathbb{E}\left[r_i^2\right]\right). \tag{135}$$

As we will later show in Remark 6, the moments of $r_i$ are indeed bounded, making the right-hand-side $O(\mathrm{polyLog}(n_1)/n_1^2)$. Note that the current result is not used to reach Remark 6, so there is no circular argument. We finally examine the first term. By the mean value theorem,

$$F^*(x^*) - F_{\setminus i}(x_{\setminus i}^*) = \left\langle \frac{1}{n_1}\sum_{j\neq i}\ell_i'(\check{r}_j)\tilde{f}_i + \lambda\frac{x^* + x_{\setminus i}^*}{2}, x^* - x_{\setminus i}^* \right\rangle, \tag{136}$$

where $\check{r}_j$ belongs to the (unordered) interval $(r_j, \tilde{r}_{j,\setminus i})$. We now show that both terms in the scalar product are small. First, we will use the fact that the first term is close to $F_{\setminus i}(x_{\setminus i}^*)$, which is by definition of $x_{\setminus i}^*$ vanishing. More precisely,

$$\|\frac{1}{n_1}\sum_{j\neq i}\ell_i'(\check{r}_j)\tilde{f}_i + \lambda\frac{x^* + x_{\setminus i}^*}{2}\| = \|\frac{1}{n_1}\sum_{j\neq i}(\ell_i'(\check{r}_j) - \ell_i'(\tilde{r}_{j,\setminus i}))\tilde{f}_i + \lambda\frac{x^* - x_{\setminus i}^*}{2}\| \tag{137}$$

$$\leq \frac{1}{n_1}\sum_{j\neq i}\|\ell_i''\|_\infty|\check{r}_j - \tilde{r}_{j,\setminus i}| + \frac{\lambda}{2}\|x^* - x_{\setminus i}^*\| \tag{138}$$

$$= O_{L_k}\left(\frac{\mathrm{polyLog}(n_1)}{\sqrt{n_1}}\right). \tag{139}$$

Since $\check{r}_j \in (r_j, \tilde{r}_{j,\setminus i})$,

$$|\check{r}_j - \tilde{r}_{j,\setminus i}| \leq |r_j - \tilde{r}_{j,\setminus i}| = O_{L_k}\left(\frac{\mathrm{polyLog}(n_1)}{\sqrt{n_1}}\right) \tag{140}$$

from Theorem 2.2 of Karoui (2018). From Theorem 2.2. and Lemma C.2 of Karoui (2018), we further have $\|x^* - x_{\setminus i}^*\| = O_{L_k}\left(\mathrm{polyLog}(n_1)/\sqrt{n_1}\right)$. Therefore, from Cauchy-Schwartz,

$$|F^*(x^*) - F_{\setminus i}(x_{\setminus i}^*)| = O_{L_k}\left(\frac{\mathrm{polyLog}(n_1)}{n_1}\right). \tag{141}$$

Putting everything together,

$$\mathbb{E}\left[(F^*(x^*) - F_{\setminus i}(x_{\setminus i}^*))^2\right] = O\left(\frac{\mathrm{polyLog}(n_1)}{n_1^2}\right) \tag{142}$$

and

$$\mathrm{Var}[\Phi^*] = O\left(\frac{\mathrm{polyLog}(n_1)}{n_1}\right) \tag{143}$$

from the Efron-Stein inequality, concluding the proof. $\qquad\square$

**Lemma 9** ($\chi$ concentrates). *Recall $\chi = 1/n_1 \mathrm{tr}[H^{-1}]$. The following concentration result holds:*

$$\mathrm{Var}[\chi] = O\left(\frac{\mathrm{polyLog}(n_1)}{n_1}\right). \tag{144}$$

*Proof.* From the Efron-Stein lemma,

$$\mathrm{Var}[\chi] \leq \sum_{i\in[n_1]}\mathbb{E}\left[(\chi - \chi_{\setminus i})^2\right], \tag{145}$$

where $\chi_{\setminus i} = 1/n_1 \mathrm{tr}[H_{\setminus i}^{-1}]$. We recall

$$H_{\setminus i} = \frac{1}{n_1}\sum_{j\neq i}\ell''(\tilde{r}_{j\setminus i})\tilde{f}_j\tilde{f}_j^\top + \lambda I_{n_1}. \tag{146}$$

Let us further decompose

$$\mathbb{E}\left[(\chi - \chi_{\setminus i})^2\right] \leq 2\mathbb{E}\left[(\chi - \tilde{\chi}_i)^2\right] + 2\mathbb{E}\left[(\tilde{\chi}_i - \chi_{\setminus i})^2\right], \tag{147}$$

defining

$$\tilde{\chi}_i = \frac{1}{n_1}\operatorname{tr}[H_i^{-1}], \qquad\qquad H_i = \frac{1}{n_1}\sum_{j \neq i}\ell''(\tilde{r}_{j,i})\tilde{f}_j\tilde{f}_j^\top + \lambda I_{n_1}. \tag{148}$$

We first focus on $\mathbb{E}\left[(\chi - \tilde{\chi}_i)^2\right]$.

$$\chi - \tilde{\chi}_i = \frac{1}{n_1}\operatorname{tr}[H^{-1}(H_i - H)H_i^{-1}] \tag{149}$$

$$= \frac{1}{n_1}\sum_{j \neq i}[\ell''(r_j) - \ell''(\tilde{r}_{j,i})]\frac{\tilde{f}_j^\top H^{-1}H_i^{-1}\tilde{f}_j}{n_1} + \frac{1}{n_1}\ell''(r_i)\frac{\tilde{f}_i^\top H^{-1}H_i^{-1}\tilde{f}_i}{n_1} \tag{150}$$

$$= \frac{1}{n_1}\sum_{j \neq i}\ell^{(3)}(\check{r}_j)(r_j - \tilde{r}_{j,i})\frac{\tilde{f}_j^\top H^{-1}H_i^{-1}\tilde{f}_j}{n_1} + \frac{1}{n_1}\ell''(r_i)\frac{\tilde{f}_i^\top H^{-1}H_i^{-1}\tilde{f}_i}{n_1} \tag{151}$$

where we used the mean value theorem and $\check{r}_j \in (r_j, \tilde{r}_{j,i})$. Thus,

$$|\chi - \tilde{\chi}_i| \leq \frac{1}{n_1}|\langle\delta, \varrho\rangle| + O_{L_k}\left(\frac{\|\ell''\|_\infty \operatorname{polyLog}(n_1)}{\lambda^2 n_1}\right). \tag{152}$$

we introduce the vectors $\delta, \varrho \in \mathbb{R}^{n_1 - 1}$ with elements

$$\delta_j = (r_j - \tilde{r}_{j,i}) \tag{153}$$

$$\varrho_j = \ell^{(3)}(\check{r}_j)\frac{\tilde{f}_j^\top H^{-1}H_i^{-1}\tilde{f}_j}{n_1}. \tag{154}$$

The latter can be controlled as $\|\varrho\| = O_{L_k}(\sqrt{n_1}\operatorname{polyLog}(n_1))$ while from Lemma 7, $\|\delta\| = O_{L_k}(\operatorname{polyLog}(n_1)/\sqrt{n_1})$. Thus, the Cauchy-Schwarz inequality implies

$$|\chi - \tilde{\chi}_i| = O_{L_k}\left(\frac{\operatorname{polyLog}(n_1)}{n_1}\right). \tag{155}$$

We now examine $\mathbb{E}\left[(\tilde{\chi}_i - \chi_{\setminus i})^2\right]$. From a Taylor expansion,

$$\tilde{\chi}_i - \chi_{\setminus i} = \frac{1}{n_1}\sum_{j \neq i}\ell^{(3)}(\tilde{r}_{j\setminus i})(\tilde{r}_{j,i} - \tilde{r}_{j\setminus i})\frac{\tilde{f}_j^\top H_{\setminus i}^{-1}H_i^{-1}\tilde{f}_j}{n_1} \tag{156}$$

$$+ \frac{1}{2n_1}\sum_{j \neq i}\ell^{(4)}(\check{s}_j)(\tilde{r}_{j,i} - \tilde{r}_{j\setminus i})^2\frac{\tilde{f}_j^\top H_{\setminus i}^{-1}H_i^{-1}\tilde{f}_j}{n_1} \tag{157}$$

for some $\hat{s}_j \in (\tilde{r}_{j,i}, \tilde{r}_{j\setminus i})$. From Lemma C.4 of Karoui (2018), $|\tilde{r}_{j,i} - \tilde{r}_{j\setminus i}| = O_{L_k}(\operatorname{polyLog}(n_1)/\sqrt{n_1})$, from which it follows that the second term is $O_{L_k}(\operatorname{polyLog}(n_1)/n_1)$. The objective is now to approximate $H_i$ in the first term by the $\tilde{f}_i-$ independent Hessian $H_{\setminus i}$, to unravel all statistical dependencies on $\tilde{f}_i$. The correction is

$$\left|\frac{1}{n_1}\sum_{j \neq i}\ell^{(3)}(\tilde{r}_{j\setminus i})(\tilde{r}_{j,i} - \tilde{r}_{j\setminus i})\frac{\tilde{f}_j^\top H_{\setminus i}^{-1}(H_i^{-1} - H_{\setminus i}^{-1})\tilde{f}_j}{n_1}\right| \tag{158}$$

$$\leq \|\ell^{(3)}\|_\infty \sup_{j \neq i}|\tilde{r}_{j,i} - \tilde{r}_{j\setminus i}|\frac{1}{n_1}\sum_{j \neq i}\frac{\|\tilde{f}_j\|^2}{\lambda n_1}\|H_i^{-1} - H_{\setminus i}^{-1}\| \tag{159}$$

$$\leq \|\ell^{(3)}\|_\infty \sup_{j \neq i}|\tilde{r}_{j,i} - \tilde{r}_{j\setminus i}|\frac{1}{n_1}\sum_{j \neq i}\frac{\|\tilde{f}_j\|^2}{\lambda^3 n_1}\|H_i - H_{\setminus i}\|. \tag{160}$$

But

$$\|H_i - H_{\setminus i}\| = \|\frac{1}{n_1}\sum_{j\neq i}\ell^{(3)}(\hat{t}_j)(\tilde{r}_{j,i} - \tilde{r}_{j\setminus i})\tilde{f}_j\tilde{f}_j^\top\| \leq \sup_{j\neq i}\left|\ell^{(3)}(\hat{t}_j)(\tilde{r}_{j,i} - \tilde{r}_{j\setminus i})\right|\|\hat{\Sigma}_{\setminus i}\| \quad (161)$$

where $\hat{\Sigma}_{\setminus i}$ is the empirical covariance of the features, excluding the $i-$th. Putting everything together yields

$$\left|\frac{1}{n_1}\sum_{j\neq i}\ell^{(3)}(\tilde{r}_{j\setminus i})(\tilde{r}_{j,i} - \tilde{r}_{j\setminus i})\frac{\tilde{f}_j^\top H_{\setminus i}^{-1}(H_i^{-1} - H_{\setminus i}^{-1})\tilde{f}_j}{n_1}\right| = O_{L_k}\left(\frac{\mathrm{polyLog}(n_1)}{n_1}\right). \quad (162)$$

Thus, going back to the original objective,

$$\mathbb{E}\left[(\tilde{\chi}_i - \chi_{\setminus i})^2\right] = \mathbb{E}\left[\left(\frac{1}{n_1}\sum_{j\neq i}\ell^{(3)}(\tilde{r}_{j\setminus i})(\tilde{r}_{j,i} - \tilde{r}_{j\setminus i})\frac{\tilde{f}_j^\top H_{\setminus i}^{-2}\tilde{f}_j}{n_1}\right)^2\right] + O\left(\frac{\mathrm{polyLog}(n_1)}{n_1^2}\right)$$

$$(163)$$

Leveraging the closed-form expression of $\tilde{r}_{j,i} - \tilde{r}_{j\setminus i}$, the first term can be written as

$$\mathbb{E}\left[\left(\ell'(\tilde{r}_{i,i})\frac{1}{n_1^2}\sum_{j\neq i}\ell^{(3)}(\tilde{r}_{j\setminus i})\tilde{f}_j^\top H_{\setminus i}^{-1}\tilde{f}_i\frac{\tilde{f}_j^\top H_{\setminus i}^{-2}\tilde{f}_j}{n_1}\right)^2\right]$$

$$\leq \mathbb{E}\left[\ell'(\tilde{r}_{i,i})^4\right]^{\frac{1}{2}}\mathbb{E}_{\{\tilde{f}_j\}_{j\neq i}}\left[\|\frac{1}{n_1^2}\sum_{j\neq i}\ell^{(3)}(\tilde{r}_{j\setminus i})\frac{\tilde{f}_j^\top H_{\setminus i}^{-2}\tilde{f}_j}{n_1}H_{\setminus i}^{-1}\tilde{f}_j\|^4\mathbb{E}_g\left[g^4\right]\right]^{\frac{1}{2}}, \quad (164)$$

using Minkovski's inequality; $g \sim \mathcal{N}(0,1)$ in the expression above. Note that, introducing the vector $h \in \mathbb{R}^{n_1-1}$ with elements $h_j = \ell^{(3)}(\tilde{r}_{j\setminus i})\tilde{f}_j^\top H_{\setminus i}^{-2}\tilde{f}_j/n_1$

$$\|\frac{1}{n_1^2}\sum_{j\neq i}\ell^{(3)}(\tilde{r}_{j\setminus i})\frac{\tilde{f}_j^\top H_{\setminus i}^{-2}\tilde{f}_j}{n_1}H_{\setminus i}^{-1}\tilde{f}_j\| = \|\frac{1}{n_1^2}h^\top F_{\setminus i}H_{\setminus i}^{-1}\| \leq \frac{1}{\lambda n_1^2}\|h\|\|F_{\setminus i}\|. \quad (165)$$

But $\|F_{\setminus i}\| = O_{L_k}(\sqrt{n_1}\mathrm{polyLog}(n_1))$, and

$$\|h\| \leq \sqrt{n_1}\|\ell^{(3)}\|_\infty\sup_{j\neq i}\left|\frac{\tilde{f}_j^\top H_{\setminus i}^{-2}\tilde{f}_j}{n_1}\right| \leq \|\ell^{(3)}\|_\infty\frac{1}{\lambda\sqrt{n_1}}\sup_{j\neq i}\|\tilde{f}_j\|^2 = O_{L_k}(\mathrm{polyLog}(n_1)\sqrt{n_1}).$$

$$(166)$$

Thus,

$$\mathbb{E}\left[\left(\ell'(\tilde{r}_{i,i})\frac{1}{n_1^2}\sum_{j\neq i}\ell^{(3)}(\tilde{r}_{j\setminus i})\tilde{f}_j^\top H_{\setminus i}^{-1}\tilde{f}_i\frac{\tilde{f}_j^\top H_{\setminus i}^{-2}\tilde{f}_j}{n_1}\right)^2\right] = O\left(\mathbb{E}\left[\ell'(\tilde{r}_{i,i})^4\right]^{\frac{1}{2}}\frac{\mathrm{polyLog}(n_1)}{n_1^2}\right).$$

$$(167)$$

To complete the proof, we need control of $\mathbb{E}\left[\ell'(\tilde{r}_{i,i})^4\right]^{\frac{1}{2}}$, which is provided by the proof of Lemma (7), where we established that $\ell'(\tilde{r}_{i,i}) = O_{L_k}(1)$. $\square$

### F.3.4 LIMITING RESIDUAL DISTRIBUTIONS

It now remains to ascertain the law of $\tilde{r}$, which we describe in the following lemma.

**Lemma 10** (Limiting distribution of $\tilde{r}_{i\setminus i}$). *The leave-one-out residual admit the simple representation*

$$\tilde{r}_{i,\setminus i} = \epsilon_i + c_{z,i}\nu Z + O_{L_2}\left(\frac{\mathrm{polyLog}(n_1)}{\sqrt{n_1}}\right) \quad (168)$$

*with $Z \sim \mathcal{N}(0,1)$ independently from $\epsilon_i, c_{z,i}$.*

*Proof.* We have

$$\tilde{r}_{i\backslash i} - \epsilon_i = \left\langle \tilde{f}_i, x_{\backslash i}/\|x_{\backslash i}\| \right\rangle \|x_{\backslash i}\| \tag{169}$$

and $Z := \frac{1}{c_{z,i}} \left\langle \tilde{f}_i, x_{\backslash i}/\|x_{\backslash i}\| \right\rangle \sim \mathcal{N}(0,1)$. Furthermore, from the proof of proposition C.7 of Karoui (2018),

$$\|x_{\backslash i}\|^2 = \|x^*\|^2 + O_{L_2}\left(\frac{\mathrm{polyLog}(n_1)}{n_1}\right) \tag{170}$$

$$= \nu^2 + O_{L_2}\left(\frac{\mathrm{polyLog}(n_1)}{\sqrt{n_1}}\right) + O_{L_2}\left(\frac{\mathrm{polyLog}(n_1)}{n_1}\right) \tag{171}$$

$$= \nu^2 + O_{L_2}\left(\frac{\mathrm{polyLog}(n_1)}{\sqrt{n_1}}\right). \tag{172}$$

Therefore,

$$\|x_{\backslash i}\| = \nu\sqrt{1 + O_{L_2}\left(\frac{\mathrm{polyLog}(n_1)}{\sqrt{n_1}}\right)} = \nu + O_{L_2}\left(\frac{\mathrm{polyLog}(n_1)}{\sqrt{n_1}}\right), \tag{173}$$

using the inequality $|\sqrt{1+x} - 1| \leq |x|$ in the last step. Finally,

$$\mathbb{E}\left[Z^2\left(\|x_{\backslash i}\| - \nu\right)^2\right] = \mathbb{E}\left[\left(\|x_{\backslash i}\| - \nu\right)^2\right] = O\left(\frac{\mathrm{polyLog}(n_1)}{n_1}\right), \tag{174}$$

in other words

$$Z\left(\|x_{\backslash i}\| - \nu\right) = O_{L_2}\left(\frac{\mathrm{polyLog}(n_1)}{\sqrt{n_1}}\right). \tag{175}$$

$\square$

**Lemma 11** (Limiting distribution of $\tilde{r}_{i,i}$). *Setting $\chi_E := \mathbb{E}[\chi]$, we have*

$$\tilde{r}_{i,i} = \mathrm{Prox}(\epsilon_i + c_{z,i}\nu Z; c_{z,i}^2 \chi_E) + O_{L_2}\left(\frac{\mathrm{polyLog}(n_1)}{\sqrt{n_1}}\right). \tag{176}$$

*Proof.* Let us introduce the shorthands $\delta_r = \tilde{r}_{i\backslash i} - \epsilon_i - c_{z,i}\nu Z$ and $\delta_\chi = \gamma_i - c_{z,i}^2 \mathbb{E}[\chi]$. From Lemma 3,

$$\left|\tilde{r}_{i,i} - \mathrm{Prox}(\epsilon_i + c_{z,i}\nu Z; c_{z,i}^2 \mathbb{E}[\chi])\right| \tag{177}$$

$$= \left|\mathrm{Prox}(\epsilon_i + c_{z,i}\nu Z + \delta_r; c_{z,i}^2 \mathbb{E}[\chi] + \delta_\chi) - \mathrm{Prox}(\epsilon_i + c_{z,i}\nu Z; c_{z,i}^2 \mathbb{E}[\chi])\right| \tag{178}$$

$$= \frac{1}{1 + c_{z,i}^2 \check{\chi}\ell''(\check{r})}\delta_r + \frac{\ell'(\check{r})}{1 + c_{z,i}^2 \check{\chi}\ell''(\check{r})}\delta_\chi, \tag{179}$$

using the two-variable mean value theorem, and eliciting the derivatives of the proximal function. $\check{r}, \check{\chi}$ are on the line between the points $(\tilde{r}_{i\backslash i} - \epsilon_i + c_{z,i}\nu Z + \delta_r, c_{z,i}^2 \mathbb{E}[\chi] + \delta_\chi)$ and $(\tilde{r}_{i\backslash i} - \epsilon_i + c_{z,i}\nu Z, c_{z,i}^2 \mathbb{E}[\chi])$. From Lemma 11, $\delta_r = O_{L_2}(\mathrm{polyLog}(n_1)/\sqrt{n_1})$. For the second term

$$\mathbb{E}\left[\left|\frac{\ell'(\check{r})}{1 + c_{z,i}^2 \check{\chi}\ell''(\check{r})}\delta_\chi\right|\right] \leq \|\ell'\|_\infty|\delta_\chi| \tag{180}$$

But

$$|\delta_\chi| \leq |\gamma_i - c_{z,i}\chi| + c_{z,i}|\chi - \mathbb{E}[\chi]| = O_{L_2}\left(\frac{\mathrm{polyLog}(n_1)}{\sqrt{n_1}}\right) \tag{181}$$

One thus reaches

$$\frac{\ell'(\check{r})}{1 + c_{z,i}^2 \check{\chi}\ell''(\check{r})}\delta_\chi = O_{L_2}\left(\frac{\mathrm{polyLog}(n_1)}{\sqrt{n_1}}\right). \tag{182}$$

Putting everything together, one thus reaches that

$$\tilde{r}_{i,i} - \mathrm{Prox}(\epsilon_i + c_{z,i}\nu Z; c_{z,i}^2 \mathbb{E}[\chi]) = O_{L_2}\left(\frac{\mathrm{polyLog}(n_1)}{\sqrt{n_1}}\right). \tag{183}$$

$\square$

**Remark 6** (Second moment of $r_i$). *The second moment $\mathbb{E}\left[r_i^2\right]$ of the responses is $O(1)$, for any $i \in [n_1]$.*

*Proof.* Fix any $i \in [n_1]$. The moment $\mathbb{E}\left[r_i^2\right]$ can be controlled as

$$\mathbb{E}\left[r_i^2\right] \le 2\mathbb{E}\left[(r_i - \tilde{r}_i)^2\right] + 2\mathbb{E}\left[\tilde{r}_i^2\right] \tag{184}$$

$$\le 2\mathbb{E}\left[\mathrm{Prox}(\epsilon_i + c_{z,i}\nu Z; c_{z,i}^2 \mathbb{E}\left[\chi\right])^2\right] + O\left(\frac{\mathrm{polyLog}(n_1)}{n_1}\right) \tag{185}$$

$$\le 4\mathbb{E}\left[\epsilon_i^2 + c_{z,i}^2 \nu^2 Z^2\right] + O\left(\frac{\mathrm{polyLog}(n_1)}{n_1}\right) = O(1). \tag{186}$$

$\square$

### F.3.5 COMPUTING THE EXPECTATIONS

**Self-consistent equation on $\nu$ —**

**Lemma 12.** *The expected squared norm $\nu_E^2 := \mathbb{E}\left[\|x^*\|^2\right]$ satisfies*

$$\nu_E^2 = -\frac{1}{\lambda}\mathbb{E}_{Z,y,\epsilon,c_z}\left[\ell'\left(\mathrm{Prox}(\epsilon_i + c_z\nu_E Z; c_z^2\chi_E) + \epsilon, y\right)\mathrm{Prox}(\epsilon_i + c_z\nu_E Z; c_z^2\chi_E)\right] + O\left(\frac{1}{\sqrt{n_1}}\right), \tag{187}$$

*where $\tilde{r}$ is a random variable distributed as $\tilde{r}_i$, given $y = y_i$.*

*Proof.* Using the stationarity condition,

$$-\lambda x^* = \frac{1}{n_1}\sum_{i \in [n_1]}\ell_i'(r_i)\tilde{f}_i. \tag{188}$$

Thus,

$$-\lambda\nu^2 = \frac{1}{n_1}\sum_{i \in [n_1]}\mathbb{E}\left[\ell_i'(r_i)(r_i - \epsilon_i)\right] \tag{189}$$

Since

$$|\ell_i'(r_i)(r_i - \epsilon_i) - \ell_i'(\tilde{r}_i)(\tilde{r}_i - \epsilon_i)| \le \left[\|\ell'\|_\infty + \|\ell_i''\|_\infty(|\epsilon_i| + |r_i|)\right]|r_i - \tilde{r}_i| \tag{190}$$

From Cauchy-Schwartz's inequality and Lemma 5,

$$\mathbb{E}\left[\left[\|\ell'\|_\infty + \|\ell_i''\|_\infty(|\epsilon_i| + |r_i|)\right]|r_i - \tilde{r}_i|\right] \le \mathbb{E}\left[(\|\ell'\|_\infty + \|\ell_i''\|_\infty(|\epsilon_i| + |r_i|))^2\right]^{1/2}O\left(\frac{\mathrm{polyLog}(n_1)}{\sqrt{n_1}}\right). \tag{191}$$

The boundedness of the first expectation follows from Remark 6, and the existence of the second moment of $\epsilon_i$ follows from the proof of Lemma 4. Thus

$$\frac{1}{n_1}\sum_{i \in [n_1]}\ell_i'(r_i)(r_i - \epsilon_i) = \frac{1}{n_1}\sum_{i \in [n_1]}\ell_i'(\tilde{r}_i)(\tilde{r}_i - \epsilon_i) + O_{L_1}\left(\frac{\mathrm{polyLog}(n_1)}{\sqrt{n_1}}\right). \tag{192}$$

We now appeal to Lemma 11 to elicit the second term. Let $\tilde{\delta}_i = \tilde{r}_{i,i} - p_i$, using the shorthand $p_i = \mathrm{Prox}(\epsilon_i + c_{z,i}\nu Z; c_{z,i}^2\mathbb{E}\left[\chi\right])$. Then,

$$|\ell_i'(\tilde{r}_i)(\tilde{r}_i - \epsilon_i) - \ell'(p_i)(p_i - \epsilon_i)| = \left|\ell''(\check{p}_i)\tilde{\delta}_i(p_i + \tilde{\delta}_i - \epsilon_i) + \ell'(p_i)\tilde{\delta}_i\right| \tag{193}$$

$$\le \|\ell''\|_\infty\left[\tilde{\delta}_i^2 + 2|\tilde{\delta}_i|(|\epsilon_i| + c_{z,i}|Z|)\right] + \|\ell'\|_\infty|\tilde{\delta}_i| \tag{194}$$

Using Cauchy-Schwartz's inequality, and the fact that $\tilde{\delta}_i = O_{L_2}(\mathrm{polyLog}(n_1)/\sqrt{n_1})$ from Lemma 11, the term in square brackets is $O_{L_1}(\mathrm{polyLog}(n_1)/\sqrt{n_1})$. Thus,

$$|\ell_i'(\tilde{r}_i)(\tilde{r}_i - \epsilon_i) - \ell'(p_i)(p_i - \epsilon_i)| = O_{L_1}\left(\frac{\mathrm{polyLog}(n_1)}{\sqrt{n_1}}\right) \tag{195}$$

and

$$\frac{1}{n_1} \sum_{i \in [n_1]} \ell_i'(r_i)(r_i - \epsilon_i) \tag{196}$$

$$= \frac{1}{n_1} \sum_{i \in [n_1]} \ell_i'(\text{Prox}(\epsilon_i + c_{z,i}\nu Z_i; c_{z,i}^2 \mathbb{E}[\chi]))(\text{Prox}(\epsilon_i + c_{z,i}\nu Z_i; c_{z,i}^2 \mathbb{E}[\chi]) - \epsilon_i) \tag{197}$$

$$+ O_{L_1}\left(\frac{\text{polyLog}(n_1)}{\sqrt{n_1}}\right). \tag{198}$$

Taking expectation,

$$\nu^2 = -\frac{1}{\lambda}\mathbb{E}_{Z,y,\epsilon}\left[\ell'\left(\text{Prox}(\epsilon + c_z\nu Z; c_z^2\mathbb{E}[\chi]) + \epsilon, y\right) \text{Prox}(\epsilon + c_z\nu Z; c_z^2\mathbb{E}[\chi])\right] + O\left(\frac{1}{\sqrt{n_1}}\right), \tag{199}$$

which completes the proof. $\square$

**Remark 7.** *Note that alternatively, $\nu_E^2$ may be expressed as*

$$\nu_E^2 = \frac{1}{\lambda\chi_E}\mathbb{E}\left[\frac{\text{Prox}\left(\epsilon + c_z\nu Z; c_z^2\chi_E\right)\left[\text{Prox}\left(\epsilon + c_z\nu_E Z; c_z^2\chi_E\right) - c_z\nu_E Z\right]}{c_z^2}\right] + O\left(\frac{1}{\sqrt{n_1}}\right). \tag{200}$$

*by applying (101) to Lemma 12.*

**Self-consistent equation for $\chi$ —**

**Lemma 13.** *Recall $\chi = 1/n_1 \text{tr}[H^{-1}]$ and $\chi_E = \mathbb{E}[\chi]$. We have*

$$\lambda\chi_E + \mathbb{E}\left[\frac{\ell''(\text{Prox}\left(\epsilon + c_z\nu_E Z; c_z^2\chi_E\right); y)c_z^2\chi_E}{1 + \ell''(\text{Prox}\left(\epsilon + c_z\nu_E Z; c_z^2\chi_E\right); y)c_z^2\chi_E}\right] = \frac{1}{\alpha} + O\left(\frac{\text{polyLog}(n_1)}{\sqrt{n_1}}\right) \tag{201}$$

*Proof.* By the construction of the Hessian matrix $H$, we have

$$\frac{1}{n_1} \sum_i H^{-1}\ell_i''(r_i)\tilde{f}_i\tilde{f}_i^\top + \lambda H^{-1} = I.$$

It follows that

$$\frac{1}{n_1^2} \sum_i \ell_i''(r_i)\tilde{f}_i^\top H^{-1}\tilde{f}_i + \lambda\chi = \frac{1}{\alpha}.$$

Applying the matrix inversion lemma then gives us

$$\frac{1}{n_1} \sum_i \frac{\ell_i''(r_i)c_{z,i}^2\widehat{\chi}_i}{1 + \ell_i''(r_i)c_{z,i}^2\widehat{\chi}_i} + \lambda\chi = \frac{1}{\alpha},$$

where

$$\widehat{\chi}_i = \frac{1}{n_1}z_i^\top \hat{H}_i^{-1}z_i$$

and

$$\hat{H}_i = \frac{1}{n_1}\sum_{j \neq i}\ell_j''(r_j)\tilde{f}_j\tilde{f}_j^\top + \lambda I.$$

We note that $\hat{\chi}$ is close to $1/n_1 \text{tr} \hat{H}_i^{-1}$. To formalize this intuition, introduce

$$\hat{\chi}_{\backslash i} = \frac{1}{n_1}z_i^\top H_{\backslash i}^{-1}z_i = 1/n_1 \text{tr}[H_{\backslash i}^{-1}] + O_{L_k}\left(\frac{\text{polyLog}(n_1)}{\sqrt{n_1}}\right), \tag{202}$$

the last equality following from Lemma G.3 of Karoui (2018). But

$$|\hat{\chi}_i - \hat{\chi}_{\backslash i}| = \left| \frac{1}{n_1} z_i^\top \hat{H}_i^{-1} (H_{\backslash i} - \hat{H}_i) H_{\backslash i}^{-1} \right| \tag{203}$$

$$\leq \frac{1}{\lambda^2} O_{L_k(1)} \left| \frac{1}{n_1} \sum_{j \neq i} \ell^{(3)}(\hat{r}_j)(r_j - r_{j,\backslash i}) \tilde{f}_j \tilde{f}_j^\top \right| \tag{204}$$

$$\leq \frac{1}{\lambda^2} O_{L_k(1)} O_{L_k}(\text{polyLog}(n_1)) \sup_{j \neq i} |r_j - r_{j,\backslash i}| = O_{L_k} \left( \frac{\text{polyLog}(n_1)}{\sqrt{n_1}} \right). \tag{205}$$

The derivation mirrors the steps of Lemma 9, and the last bound follows from Theorem 2.2 of Karoui (2018). Thus,

$$\hat{\chi}_i = \frac{1}{n_1} \text{tr}[H_{\backslash i}^{-1}] + O_{L_k} \left( \frac{\text{polyLog}(n_1)}{\sqrt{n_1}} \right). \tag{206}$$

Now in trace form, we approximate $1/n_1 \, \text{tr}[H_{\backslash i}^{-1}]$ back by $1/n_1 \, \text{tr}[\hat{H}_i^{-1}]$. This can be done along the exact same lines as the previous approximation, finally yielding

$$\hat{\chi}_i = \frac{1}{n_1} \text{tr}[\hat{H}_i^{-1}] + O_{L_k} \left( \frac{\text{polyLog}(n_1)}{\sqrt{n_1}} \right). \tag{207}$$

We now show that $\hat{\chi}_i$ is close to $\chi$:

$$|\hat{\chi}_i - \chi| = \frac{1}{n_1} |\text{tr}[\hat{H}_i^{-1}(H - \hat{H}_i)H^{-1}]| \tag{208}$$

$$= \frac{1}{n_1^2} |\ell_i''(r_i)| |\text{tr}[\hat{H}_i^{-1} \tilde{f}_i \tilde{f}_i^\top H^{-1}]| \tag{209}$$

$$\leq \frac{1}{n_1^2} \|\ell''\|_\infty \|\hat{H}_i^{-1} H^{-1}\| \|\tilde{f}_i\|^2 = O_{L_k} \left( \|\ell''\|_\infty \frac{\text{polyLog}(n_1)}{n_1 \lambda^2} \right) \tag{210}$$

Furthermore, we can also approximate $\ell''(r_i) \approx \ell''(\tilde{r}_i)$. More precisely,

$$|\ell''(r_i) - \ell''(\tilde{r}_i)| = O_{L_k} \left( \|\ell^{(3)}\|_\infty \frac{\text{polyLog}(n_1)}{n_1} \right) \tag{211}$$

Thus,

$$\frac{\ell_i''(r_i) c_{z,i}^2 \hat{\chi}_i}{1 + \ell_i''(r_i) c_{z,i}^2 \hat{\chi}_i} = \frac{\ell_i''(\tilde{r}_i) c_{z,i}^2 \chi}{1 + \ell_i''(\tilde{r}_i) c_{z,i}^2 \chi} + O_{L_k} \left( \frac{\text{polyLog}(n_1)}{\sqrt{n_1}} \right) \tag{212}$$

Observe that further

$$\left| \frac{\ell_i''(\tilde{r}_i) c_{z,i}^2 \chi}{1 + \ell_i''(\tilde{r}_i) c_{z,i}^2 \chi} - \frac{\ell_i''(\tilde{r}_i) c_{z,i}^2 \mathbb{E}[\chi]}{1 + \ell_i''(\tilde{r}_i) c_{z,i}^2 \mathbb{E}[\chi]} \right| = \left| \frac{\ell_i''(\tilde{r}_i) c_{z,i}^2 (\chi - \mathbb{E}[\chi])}{(1 + \ell_i''(\tilde{r}_i) c_{z,i}^2 \mathbb{E}[\chi])(1 + \ell_i''(\tilde{r}_i) c_{z,i}^2 \chi)} \right| \tag{213}$$

$$\leq \|\ell''\|_\infty O_{L_2} \left( \frac{\text{polyLog}(n_1)}{\sqrt{n_1}} \right), \tag{214}$$

using the concentration of $\chi$, see Lemma 9, and that $0 \leq c_{z,i} \leq 1$. Summarizing,

$$\frac{\ell_i''(r_i) c_{z,i}^2 \hat{\chi}_i}{1 + \ell_i''(r_i) c_{z,i}^2 \hat{\chi}_i} = \frac{\ell_i''(\tilde{r}_i) c_{z,i}^2 \mathbb{E}[\chi]}{1 + \ell_i''(\tilde{r}_i) c_{z,i}^2 \mathbb{E}[\chi]} + O_{L_2} \left( \frac{\text{polyLog}(n_1)}{\sqrt{n_1}} \right). \tag{215}$$

Finally, let $\tilde{\delta}_i = \tilde{r}_{i,i} - p_i$, using the shorthand $p_i = \text{Prox}(\epsilon_i + c_{z,i} \nu Z; c_{z,i}^2 \mathbb{E}[\chi])$. One can control

$$\left| \frac{\ell_i''(\tilde{r}_i) c_{z,i}^2 \mathbb{E}[\chi]}{1 + \ell_i''(\tilde{r}_i) c_{z,i}^2 \mathbb{E}[\chi]} - \frac{\ell_i''(p_i) c_{z,i}^2 \mathbb{E}[\chi]}{1 + \ell_i''(p_i) c_{z,i}^2 \mathbb{E}[\chi]} \right| \leq \frac{1}{\lambda} \|\ell^{(3)}\|_\infty |\tilde{\delta}_i|. \tag{216}$$

using $\chi \leq 1/\lambda$. From Lemma 11, $\tilde{\delta}_i = O_{L_2}(\text{polyLog}(n_1)/\sqrt{n_1})$. Putting all intermediary results together, and taking the expectation, it holds that

$$\mathbb{E} \left[ \frac{\ell''(\text{Prox}_i(c_z \nu_E Z; c_z^2 \chi_E); y) c_z^2 \chi_E}{1 + \ell''(\text{Prox}_i(c_z \nu_E Z; c_z^2 \chi_E); y) c_z^2 \chi_E} \right] + \lambda \chi_E = \frac{1}{\alpha} + O \left( \frac{\text{polyLog}(n_1)}{\sqrt{n_1}} \right), \tag{217}$$

proving the lemma. □

### F.3.6 LAST STEPS

We begin by defining the constants $\boldsymbol{\nu}$ and $\boldsymbol{\chi}$ as solutions of the following self-consistent equations:

$$\boldsymbol{\nu}^2 = \frac{1}{\lambda \boldsymbol{\chi}} \mathbb{E}\left[\frac{z^*\left[z^* - c_z \boldsymbol{\nu} Z\right]}{c_z^2}\right], \tag{218}$$

$$\mathbb{E}\left[\frac{\ell''(z^*; y) c_z^2 \boldsymbol{\chi}}{1 + \ell''(z^*; y) c_z^2 \boldsymbol{\chi}}\right] + \lambda \boldsymbol{\chi} = \frac{1}{\alpha} \tag{219}$$

where

$$z^* = \mathrm{Prox}\left(\epsilon + c_z \boldsymbol{\nu} Z; c_z^2 \boldsymbol{\chi}\right).$$

and take for granted that $\boldsymbol{\nu}$ and $\boldsymbol{\chi}$ exist uniquely. We further assume the regularity conditions for the map $(\mu_q, \mu_\xi, b) \mapsto (\boldsymbol{\nu}, \boldsymbol{\chi})$.

**Assumption 3.** *The map $(\mu_q, \mu_\xi, b) \mapsto (\boldsymbol{\nu}, \boldsymbol{\chi})$ is continuous and*

$$(\nu_E, \chi_E) \to (\boldsymbol{\nu}, \boldsymbol{\chi})$$

*as $n_1 \to \infty$, where the convergence holds uniformly over $(\mu_q, \mu_\xi, b)$ in any compact set.*

Define the asymptotic inner objective function by

$$\phi_A(\mu_q, \mu_\xi, b) := \mathbb{E}_{c_z, c_q, c_\xi, z, y}[\ell(z^* + \epsilon; y)] + \frac{\lambda}{2} \boldsymbol{\nu}^2$$

where we recall that $\epsilon = \mu_q c_q + \mu_\xi c_\xi + c_z z + b$, $z \sim \mathcal{N}(0, 1)$ independent of $c_q$, $c_\xi$, and $c_z$. Let

$$G := \min_{\mu_q, \mu_\xi, b} g_d(\mu_q, \mu_\xi, b), \qquad g_d(\mu_q, \mu_\xi, b) := \phi_d(\mu_q, \mu_\xi, b) + \frac{\lambda}{2} \begin{bmatrix} \mu_q & \mu_\xi \end{bmatrix} \begin{bmatrix} 1 & \gamma \\ \gamma & 1 \end{bmatrix}^{-1} \begin{bmatrix} \mu_q \\ \mu_\xi \end{bmatrix} \tag{220}$$

and

$$G_A := \min_{\mu_q, \mu_\xi, b} g_A(\mu_q, \mu_\xi, b), \qquad g_A(\mu_q, \mu_\xi, b) := \phi_A(\mu_q, \mu_\xi, b) + \frac{\lambda}{2} \begin{bmatrix} \mu_q & \mu_\xi \end{bmatrix} \begin{bmatrix} 1 & \gamma \\ \gamma & 1 \end{bmatrix}^{-1} \begin{bmatrix} \mu_q \\ \mu_\xi \end{bmatrix} \tag{221}$$

so that $G$ denotes our original optimization problem and $G_A$ is the surrogate problem where the random $n_1$-dependent function $\phi$ has been replaced by $\phi_A$. In establishing the result of Theorem 4 it remains to establish the asymptotic equivalence between $G$ and $G_A$. We begin with the following brief result which establishes the sufficiency in considering minimization of $g_d$ and $g_A$ over a compact set in $\mathbb{R}^3$.

**Lemma 14.** *Let $v = (\mu_q, \mu_\xi, b)$ and set*

$$v_d^* = \arg\min_{v \in \mathbb{R}^3} g_d(v), \qquad v_A^* = \arg\min_{v \in \mathbb{R}^3} g_A(v).$$

*For $\delta \in (0, 1)$, there exists a compact set $\mathcal{V} := \mathcal{V}(\delta) \subset \mathbb{R}^3$, not depending on $d$ (equivalently on $n_1$), such that*

$$v_d^*, v_A^* \in \mathcal{V}$$

*for all $d \in \mathbb{N}$, with probability exceeding $1 - \delta$.*

*Proof.* If a function $h : \mathbb{R}^p \to \mathbb{R}$ is coercive, in the sense that

$$\lim_{\|x\| \to \infty} h(x) = +\infty,$$

then $h$ has bounded level sets

$$\mathrm{lev}_h(c) := \{x \in \mathbb{R}^p : h(x) \leq c\} \quad \text{for } c \in \mathbb{R}.$$

To show that $g_A$ is coercive note that if $\|v\| \to \infty$, but $\|(\mu_q, \mu_\xi)\|$ remains bounded, then necessarily $|b| \to \infty$ and (A5) implies $g_A \to \infty$. If indeed $\|(\mu_q, \mu_\xi)\| \to \infty$, then due to the quadratic regularization term

$$Q(\mu_q, \mu_\xi) = \frac{\lambda}{2} \begin{bmatrix} \mu_q & \mu_\xi \end{bmatrix} \begin{bmatrix} 1 & \gamma \\ \gamma & 1 \end{bmatrix}^{-1} \begin{bmatrix} \mu_q \\ \mu_\xi \end{bmatrix}$$

we have

$$\lim_{\|(\mu_q, \mu_\xi)\| \to \infty} g_A(\mu_q, \mu_\xi, b) = \infty.$$

since $\ell \geq 0$, and so $g_A$ is indeed coercive. Moreover, the map $v \mapsto (\boldsymbol{\nu}, \boldsymbol{\chi})$ is continuous by Assumption 3, and so $\phi_A$ is continuous in $v$ by continuity of $\ell$ and the proximal operator. It then follows that $g_A$ is continuous in $v$ and so, having established coercivity, its level sets are closed and bounded, hence compact. From similar observations and reasoning, we see that $g_d(v)$ is also continuous. Since $\{(\epsilon_i, y_i)\}_{i \geq 1}$ are sub-Gaussian and $\ell$ has a quadratic majorant by Remark 3, $\{\ell(\epsilon_i; y_i)\}_{i \geq 1}$ are sub-exponential and so by Bernstein's Inequality (Vershynin, 2018, Theorem 2.8.1), for any $\kappa > 0$ sufficently large,

$$\mathbb{P}\left( \left| \frac{1}{n_1} \sum_{i \in [n_1]} \ell(\epsilon_i; y_i) - \mathbb{E}[\ell(\epsilon; y)] \right| > \kappa \right) \leq 2 \exp\left( -C\kappa^2 n_1 \right)$$

where $C > 0$ is an absolute constant. Therefore, taking $\kappa > 0$ large so that $\sum_{n_1 \geq 1} 2 \exp\left( -C\kappa^2 n_1 \right) \leq \delta$, by a union bound, one can ensure that for

$$\Omega_\delta := \bigcap_{n_1 \geq 1} \left\{ \left| \frac{1}{n_1} \sum_{i \in [n_1]} \ell(\epsilon_i; y_i) - \mathbb{E}[\ell(\epsilon; y)] \right| \leq \kappa \right\},$$

$\mathbb{P}(\Omega_\delta) \geq 1 - \delta$. In the remainder of the proof, we work on the event $\Omega_\delta$. Letting $\lambda_{\min} > 0$ denote the smallest eigenvalue of the positive definite matrix in $Q(\mu_q, \mu_\xi)$, we have

$$\frac{\lambda \lambda_{\min}}{2} \|(\mu_{q,d}^*, \mu_{\xi,d}^*)\|^2 \leq g_d(v_d^*) \leq \mathbb{E}[\ell(\epsilon, y)] + \kappa =: \beta_0$$

where we write the optimal solution by $v_d^* = (\mu_{q,d}^*, \mu_{\xi,d}^*, b_d^*)$. The first inequality above is due to $g_d(v) \geq Q(\mu_q, \mu_\xi)$ whereas the second follows from $g_d(v_d^*) \leq g_d(0)$. Thus, we find that the first two components of $v_d^*$ are bounded uniformly (independent of $d$) — in particular

$$\|(\mu_{q,d}^*, \mu_{\xi,d}^*)\| \leq \sqrt{\frac{2\beta_0}{\lambda \lambda_{\min}}} =: B_0$$

Moreover, as

$$g_d(v) \geq \mathbb{E}[\ell(\epsilon, y)] - \kappa \to \infty$$

as $|b| \to \infty$ for fixed $(\mu_q, \mu_\xi)$ by (A5), there exists $B_1 > 0$ — independent of $d$ — such that

$$\inf_{\|(\mu_q, \mu_\xi)\| \leq B_0, \, |b| > B_1} g_d(\mu_q, \mu_\xi, b) \geq \beta_0 + 1$$

Hence, on $\Omega_\delta$, the minimizer $v_d^*$ of $g_d$ lies in the set

$$\mathcal{U} := \{ (\mu_q, \mu_\xi, b) : \|(\mu_q, \mu_\xi)\| \leq B_0, \, |b| \leq B_1 \}$$

which is compact by continuity of $g_d$ and importantly does not depend on $d$. Therefore, taking $\beta = \max(g_A(0), \beta_0 + 1)$, having previously established the level-compactness of $g_A$, we have that

$$\mathcal{V} := \mathcal{U} \cup \mathrm{lev}_{g_A}(\beta)$$

is compact and contains $v_A^*$ and $v_d^*$ for all $d \in \mathbb{N}$. $\qquad \square$

The compactness yielded by the above lemma is an important fact that will be carried in the subsequent results. Notably, we remark that all preliminaries that have been established hereunto involving $O(b_n)$ errors terms for some sequence $(b_n)$ hold uniformly over the above defined set $\mathcal{V}$. To see why, simply recall the meaning of writing $a_n = O(b_n)$ is to infer the existence of an $n$-independent constant $C > 0$ such that

$$a_n \leq C \cdot b_n$$

for $n$ sufficiently large. Revisiting our previous results, one can check that, given a sequence $a_n(v)$ parameterized by $v \in \mathcal{V}$, the map $v \to C(v)$, namely the map from the parameter to the order-defining constant, is continuous. This turns out to be a simple consequences of the continuous of the loss $\ell$. Therefore, $\sup_{v \in \mathcal{V}} C(v) < \infty$, and, as stated, all previous results hold uniformly over $v \in \mathcal{V}$.

**Lemma 15** (Uniform convergence to $\phi_A$). *We have*

$$\sup_{v \in \mathcal{V}} |\mathbb{E}\phi_d(v) - \phi_A(v)| \longrightarrow 0$$

*as $d \to \infty$*

*Proof.* Let

$$z_{n_1}^* := \epsilon + \mathrm{Prox}(\epsilon + c_z \nu_E Z; c_z^2 \chi_E),$$

noting that the dependence on $n_1$ in $z_{n_1}^*$ comes through the deterministic $n_1$-dependent quantities $\chi_E$ and $\nu_E$. Recall that by Assumption 3, $(\nu_E, \chi_E) \longrightarrow (\boldsymbol{\nu}, \boldsymbol{\chi})$ uniformly over $\mathcal{V}$. By continuity of the proximal operator, applying the continuous mapping theorem together with Slutsky's theorem yields convergence of

$$z_{n_1}^* \xrightarrow{P} \epsilon + z^*.$$

Note that this convergence holds uniformly over $\mathcal{V}$ as the proximal operator is non-expansive (i.e. Lipschitz). For some $\hat{r}_i$ lying between $\tilde{r}_i$ and $r_i$, and $\check{r}_i$ between $\tilde{r}_i$ and $z_{n_1}^*$, a Taylor expansion yields

$$\mathbb{E}[\phi_d(v)] = \frac{1}{n_1} \sum_{i \in [n_1]} \left( \mathbb{E}[\ell(z_{n_1}^*; y_i)] + \mathbb{E}[\ell'(\hat{r}_i; y_i)(r_i - \tilde{r}_i)] + \mathbb{E}[\ell'(\check{r}_i; y_i)(\tilde{r}_i - z_{n_1}^*)] \right)$$

$$+ \frac{\lambda}{2} \nu_E^2 + O\left( \frac{1}{\sqrt{n_1}} \right)$$

$$= \mathbb{E}[\ell(z_{n_1}^*; y)] + O\left( \frac{\mathrm{polyLog}(n_1)}{\sqrt{n_1}} \right) + \frac{\lambda}{2} \nu_E^2$$

where in the second equality we used $\|\ell'\|_\infty = O(\mathrm{polyLog}(n_1))$ and applied the upper bound on $|r_i - \tilde{r}_i|$ from Lemma 5, and Lemma 11 to bound $|\tilde{r}_i - z_{n_1}^*|$. Now, for $M > 0$, decomposing

$$\mathbb{E}[\ell(z_{n_1}^*; y)] = \mathbb{E}\left[ \ell(z_{n_1}^*; y) 1_{\{\ell(z_{n_1}^*; y) \leq M\}} \right] + \mathbb{E}\left[ \ell(z_{n_1}^*; y) 1_{\{\ell(z_{n_1}^*; y) > M\}} \right],$$

we have that

$$\mathbb{E}\left[ \ell(z_{n_1}^*; y) 1_{\{\ell(z_{n_1}^*; y) \leq M\}} \right] \to \mathbb{E}\left[ \ell(z^* + \epsilon; y) 1_{\{\ell(z^*; y) \leq M\}} \right]$$

uniformly over $\mathcal{V}$ by the Dominated Convergence Theorem. Uniform convergence of $(\nu_E, \chi_E) \to (\boldsymbol{\nu}, \boldsymbol{\chi})$ yields uniform boundedness in $L^2$ of $(\ell(z_{n_1}^*; y))_{n_1 \geq 1}$ since $\ell$ has bounded second derivative. Namely,

$$\sup_{v \in \mathcal{V}} \sup_{n_1 \in \mathbb{N}} \mathbb{E}[\ell(z_{n_1}^*; y)^2] < \infty$$

which provides uniform integrability of $(\ell(z_{n_1}^*; y))_{n_1 \geq 1}$. That is for arbitrary $\varepsilon > 0$, there exists $M > 0$ for which

$$\sup_{v \in \mathcal{V}} \mathbb{E}\left[ \ell(z_{n_1}^*; y) 1_{\{\ell(z_{n_1}^*; y) > M\}} \right] < \varepsilon$$

as $n_1 \to \infty$ and so, uniformly over $\mathcal{V}$, one has

$$\mathbb{E}\left[ \ell(z_{n_1}^*; y) \right] \to \mathbb{E}\left[ \ell(z^* + \epsilon; y) \right]$$

Lastly, by Assumption 3, $\lambda \nu_E^2 / 2 \longrightarrow \lambda \boldsymbol{\nu}^2 / 2$ uniformly over $\mathcal{V}$, which yields the result.

$\square$

**Lemma 16** (Uniform convergence to $\mathbb{E}\phi(v)$). *We have*

$$\sup_{v \in \mathcal{V}} |\phi(v) - \mathbb{E}[\phi(v)]| \xrightarrow{P} 0$$

*as $d \to \infty$*

*Proof.* We include the parametrization of $v$ in $x_d^*(v)$, $F_d^*(v)$, and other quantities where the parameters $v = (\mu_q, \mu_\xi, b)$ were previously fixed and hence omitted in the notation. Note that continuous differentiability of the map $v = (\mu_q, \mu_\xi, b) \mapsto \ell(\langle c_{z,i} z_i, x \rangle + c_{q,i} \mu_q + c_{\xi,i} \mu_\xi + b; y_i)$ carries to the map $v \mapsto x_d^*(v)$ because strong convexity from the regularizer $\lambda/2\|x\|^2$ ensures a unique minimizer and the Implicit Function Theorem provides that the minimizer depends smoothly on $v$. Thus, the map $v \mapsto x_d^*(v)$ is uniformly bounded over $\mathcal{V}$ as the set is compact. Then, observe that

$$\sup_{v \in \mathcal{V}} \frac{\lambda}{2} \|x_d^*(v)\|^2 \leq F_d^*(0; v) = \frac{1}{n_1} \sum_{i \in [n_1]} \ell(\epsilon_i(v); y_i) = O(\text{polyLog}(n_1))$$

by compactness of $\mathcal{V}$ and since $\sup_{i \leq n_1} |\epsilon_i| = O(\text{polyLog}(n_1))$ by the proof of Lemma 4. Again, invoking compactness of $\mathcal{V}$ and Lemma 4, we have that

$$\sup_{v \in \mathcal{V}} \left\| \nabla_v \left[ \frac{1}{n_1} \sum_{i \in [n_1]} \ell_i(\langle \tilde{f}_i, x_d^*(v) \rangle) \right] \right\| = O(\text{polyLog}(n_1))$$

since by the Implicit Function theorem, $\partial_v x_d^*(v) = O(\text{polyLog}(n_1))$, and we have that $\|\ell'\|_\infty = O(\text{polyLog}(n_1))$. Putting these results together, we have that $\phi_d$ is Lipschitz on $\mathcal{V}$ with a polylogarithmic constant which we denote by $L_d$. Namely,

$$|\phi(v) - \phi(w)| \leq L_d \|v - w\| = \|v - w\| \cdot O(\text{polyLog}(n_1))$$

for $v, w \in \mathcal{V}$. Lipschitzness of $\mathbb{E}\phi$ follows by linearity of the expectation and thus the centered process $Z := \phi - \mathbb{E}\phi$ is $2L_d$-Lipschitz. We finish the proof with a covering-net argument. Fix $\varepsilon > 0$, set

$$\delta_d = \frac{\varepsilon}{4L_d}$$

By compactness of $\mathcal{V}$, let $v^{(1)}, \ldots, v^{(N_d)}$ be points in $\mathcal{V}$ such that

$$\mathcal{V} \subset \cup_{m=1}^{N_d} B_{\delta_d}(v^{(m)})$$

where $B_{\delta_d}(v^{(m)})$ denotes a ball of radius $\delta_d$ centered at $v^{(m)}$. A standard volume argument shows that we may take $N_d = O(\text{polyLog}(n_1))$ as $L_d = O(\text{polyLog}(n_1))^2$. Using the variance bound of Lemma 8, Chebyshev's inequality yields

$$\mathbb{P}\left( |Z(v^{(m)})| > \frac{\varepsilon}{2} \right) = O(\text{polyLog}(n_1)/n_1).$$

for $m \in [N_d]$. A union bound then provides

$$\mathbb{P}\left( \max_{m \leq N_d} |Z(v^{(m)})| > \frac{\varepsilon}{2} \right) = O(\text{polyLog}(n_1)/n_1)$$

as $N_d = O(\text{polyLog}(n_1))$. By construction of the cover $\{v^{(1)}, \ldots, v^{(N_d)}\}$, for any $v \in \mathcal{V}$, there exists $v^{(m)}$ such that

$$|Z(v)| \leq |Z(v^{(m)})| + \frac{\varepsilon}{2}.$$

Hence,

$$\mathbb{P}\left( \sup_{v \in \mathcal{V}} |\phi(v) - \mathbb{E}[\phi(v)]| > \varepsilon \right) \leq \mathbb{P}\left( \max_{m \leq N_d} |Z(v^{(m)})| > \frac{\varepsilon}{2} \right) \to 0$$

as $d \to \infty$ which concludes the proof. □

The following result marks the grand conclusion of the section and completes the proof of Theorem 4.

**Lemma 17.** *We have*

$$|G - G_A| \xrightarrow{P} 0 \tag{222}$$

*as $d \to \infty$.*

---

[2]Without loss of generality we may assume $\mathcal{V}$ is a closed sphere of radius $r > 0$ and by (Vershynin, 2018, Corollary 4.2.13), $N_d \leq (2r\delta_d^{-1} + 1)^3$.

*Proof.* Let $v^*$ and $v_A^*$ be the respective minimizers of $g$ and $g_A$, hiding the $d$-dependence for notational ease. Setting

$$\Delta = \sup_{v \in \mathcal{V}} |\phi(v) - \phi_A(v)|,$$

we have

$$G - G_A = g(v^*) - g_A(v_A^*) \leq g(v_A^*) - g_A(v_A^*) \leq \Delta.$$

By symmetry, we obtain

$$|G - G_A| \leq \Delta$$

and so the result follows by the triangle inequality in applying Lemma 15 and Lemma 16. $\qquad\square$

## G    PROOF OF THEOREM 5

Appendix F details the asymptotic characterization of the learning of the attention model 6, in the asymptotic limit of Assumption 1. We now expound the related characterization for the linear classifier baselines $\mathsf{L}_{w,b}^{\text{pool}}$ (4) and $\mathsf{L}_{w,b}^{\text{vec}}$ (3), summarized in the main text in Theorem 5. The first part of the latter for the pooled classifier $\mathsf{L}_{w,b}^{\text{pool}}$ (4) was already covered in Corollary 3 in Appendix F, as it coincides with a special case of Theorem 4 for the attention model.

We consequently turn to analyzing the learning of the linear classifier acting on the vectorized inputs $\mathsf{L}_{w,b}^{\text{vec}}$ (3), described in subsection 1.1. Formally, let us consider the empirical risk minimization problem

$$w^*, b^* = \underset{w \in \mathbb{R}^{Ld}, b \in \mathbb{R}}{\text{argmin}} \frac{1}{n} \sum_{i \in [n]} \ell(\langle f_i, w \rangle + \langle \mu(v_i), w \rangle + b, y_i) + \frac{\lambda}{2} \|w\|^2 := \underset{b \in \mathbb{R}}{\text{argmin}} \phi(b) \qquad (223)$$

where we denote $f_i := \text{vec}(Z_i)$ the flattened background term of the inputs, and $\mu(v_i) = \theta \text{vec}(v_i \xi^\top)$. We denote by $p_v$ the law of $v$ over $\{0, 1\}^L$, and recall that $p_v(v = 0_L) := 1 - \pi$ by definition. Note also that in these notations, $y = 1 - \delta_{v, 0_L}$ is a function of $v$.

**Proposition 3** (Test error and train loss of the linear classifier on vectorized inputs). *The test error and train loss of the linear classifier acting on the vectorized inputs, described in subsection 1.1, trained with the empirical risk minimization* (223)*, concentrate in the asymptotic limit of Assumption 1 to*

$$\mathcal{E}_{\text{train}} = \min_b \mathbb{E}_{y, v', z} \left[ \ell(z^* + b, y) \right] + \frac{\lambda}{2} \nu^2$$

$$\mathcal{E}_{\text{test}} = (1 - \pi) \Phi \left( \frac{b^*}{\nu} \right) + \mathbb{E}_{v \neq 0_L} \left[ \Phi \left( \frac{-b^* - \theta m(v)}{\nu} \right) \right].$$

*For any $v \in \mathbb{R}^L$ we noted $m(v) = v_1 m_1 + \cdots + v_L m_L$, where the summary statistics $\nu, \chi, \{m_k\}_{k \in [L]}, b^*$ are given by the set of self-consistent equations:*

$$m_k = -\frac{1}{\lambda} \mathbb{E}_{y, v', z} \left[ \ell'(z^* + b^*, y) \left( \theta^2 v_k' + \frac{m_k}{\nu} z \right) \right]$$

$$\nu^2 = -\frac{1}{\lambda} \mathbb{E}_{y, v', z} \left[ \ell'(z^* + b^*, y) z^* \right]$$

$$\frac{L}{\alpha \chi} = \mathbb{E}_{y, v', z} \left[ \frac{\ell''(z^* + b^*, y)}{1 + \ell''(z^* + b^*, y) \chi} \right] + \lambda$$

*where $v' \sim p_v$, $y = 1 - \delta_{v, 0_L}$, $z \sim \mathcal{N}(0, 1)$, and*

$$b^* = \arg\min_b \mathbb{E} \left[ \ell(z^* + b^*, y) \right] + \frac{\lambda}{2} \nu^2.$$

*We employed the shorthand $z^* = \text{prox}_{\chi \ell(\cdot + b^*, y)}(\nu z + m(v'))$.*

Note that the data distribution formally coincides with a Gaussian mixture with $2^L + 1$ isotropic clusters, and the analysis of logistic regression on such data is covered in Loureiro et al. (2021). In

this appendix, we rather give a more concise derivation in the specific setting considered, leveraging once more the leave-one-out approach. The following derivation closely follows the steps of the proof of Theorem 4, detailed in Appendix F. For the sake of conciseness, we only provide an informal sketch of the derivation. Before doing so, let us observe that the equations (224) are amenable to being massaged into a form closer to that of Loureiro et al. (2021); Mignacco et al. (2020a).

**Remark 8.** *The system of self-consistent equations 224 can also be written as*

$$\hat{\chi} = \frac{1}{\chi}\mathbb{E}\left[1 - \text{prox}'_{\chi\ell(\cdot+b^*,y)}(\nu z + m(v'))\right], \qquad \chi = \frac{L}{\alpha}\frac{1}{\lambda + \hat{\chi}} \tag{224}$$

$$\hat{m}_k = \frac{\theta^2}{\chi}\mathbb{E}\left[(z^* - \nu z - m(v')v'_k\right], \qquad m_k = \frac{\hat{m}_k}{\lambda + \hat{\chi}} \tag{225}$$

$$\hat{\nu}^2 = \frac{1}{\chi^2}\mathbb{E}\left[(z^* - m(v') - \nu z)^2\right], \qquad \nu^2 = \frac{\frac{L}{\alpha}\hat{\nu}^2 + \frac{1}{\theta^2}\sum_{k=1}^{L}\hat{m}_k^2}{(\lambda + \hat{\chi})^2}. \tag{226}$$

*Proof.* We begin by noting that the derivative of the proximal operator reads

$$\frac{\partial \text{prox}_{\gamma\ell(\cdot)}(\omega)}{\partial_\omega} = \frac{1}{1 + \gamma\ell''(\text{prox}_{\gamma\ell(\cdot)}(\omega))}. \tag{227}$$

Therefore,

$$\hat{\chi} = \mathbb{E}\left[\frac{\ell''(z^* + b^*, y)}{1 + \ell''(z^* + b^*, y)\chi}\right] \tag{228}$$

and the last equation of (224) can thus be written as

$$\chi = \frac{L}{\alpha}\frac{1}{\lambda + \hat{\chi}}. \tag{229}$$

Let us now focus on the first equation of (224). We have

$$0 = \lambda m_k + \theta^2\mathbb{E}\left[\ell'(z^* + b^*, y)v'_k\right] + \frac{m_k}{\nu}\mathbb{E}\left[\ell'(z^* + b^*, y)z\right] \tag{230}$$

$$= \lambda m_k - \frac{\theta^2}{\chi}\mathbb{E}\left[(z^* - \nu z - m(v')v'_k\right] + m_k\mathbb{E}\left[\frac{\ell''(z^* + b^*, y)}{1 + \chi\ell''(z^* + b^*, y)}\right]. \tag{231}$$

Thus,

$$m_k = \frac{\hat{m}_k}{\lambda + \hat{\chi}}. \tag{232}$$

Finally, starting from the second equation of (224),

$$0 = \lambda\nu^2 - \frac{1}{\chi}\mathbb{E}\left[(z^* - \nu z - m(v'))^2\right] - \frac{1}{\chi}\mathbb{E}\left[(z^* - \nu z - m(v'))(\nu z + m(v'))\right] \tag{233}$$

$$= \lambda\nu^2 - \frac{1}{\chi}\mathbb{E}\left[(z^* - \nu z - m(v'))^2\right] - \frac{1}{\theta^2}\sum_{k=1}^{L}\hat{m}_k m_k + \nu^2\mathbb{E}\left[\frac{\ell''(z^* + b^*, y)}{1 + \chi\ell''(z^* + b^*, y)}\right] \tag{234}$$

$$= \lambda\nu^2 - \frac{1}{\chi^2}\frac{L}{\alpha}\frac{1}{\lambda + \hat{\chi}}\mathbb{E}\left[(z^* - \nu z - m(v'))^2\right] \tag{235}$$

$$- \frac{1}{\theta^2(\lambda + \hat{\chi})}\sum_{k=1}^{L}\hat{m}_k^2 + \nu^2\mathbb{E}\left[\frac{\ell''(z^* + b^*, y)}{1 + \chi\ell''(z^* + b^*, y)}\right] \tag{236}$$

Thus

$$\nu^2 = \frac{\frac{L}{\alpha}\hat{\nu}^2 + \frac{1}{\theta^2}\sum_{k=1}^{L}\hat{m}_k^2}{(\lambda + \hat{\chi})^2} \tag{237}$$

$\square$

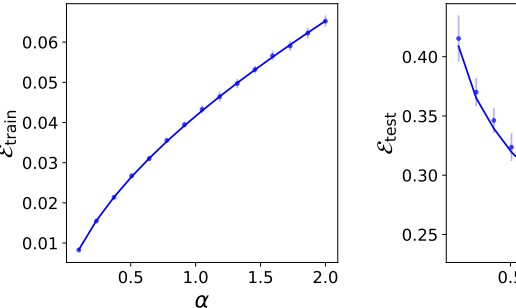

Figure 4: Train loss (**left**) and test error (**right**) of the linear classifier acting on the vectorized outputs, as discussed in subsection 1.1 of the main text, for $L = 3, R = 2, \theta = 2, \pi = 0.5, \lambda = 0.01$. Solid lines: theoretical characterization of Proposition 3. Dots : numerical simulations in dimension $d = 1000$. Error bars indicate one standard deviation over 8 trials.

**Sketch of the derivation —** For a given $b$, let us introduce

$$\Phi = \operatorname*{argmin}_{w} \frac{1}{n} \sum_{j \in [n]} \ell(\langle f_j, w \rangle + \langle \mu(v_j), w \rangle + b, y_j) + \frac{\lambda}{2}\|w\|^2$$

$$\Phi_{\backslash i} = \operatorname*{argmin}_{w} \frac{1}{n} \sum_{j \neq i} \ell(\langle f_j, w \rangle + \langle \mu(v_j), w \rangle + b, y_j) + \frac{\lambda}{2}\|w\|^2$$

$$\widetilde{\Phi} = \Phi_{\backslash i} + \min_{w} \left[ \frac{1}{n}\ell(\langle f_i, w \rangle + \langle \mu(v_i), w \rangle + b, y_i) + \frac{1}{2}(w - w_{\backslash i}^*)^\top H_{\backslash i}(w - w_{\backslash i}^*) \right],$$

where the Hessian is defined as

$$H_{\backslash i} = \frac{1}{n} \sum_{j \neq i} \ell''(\langle f_j, w \rangle + \langle \mu(v_j), w \rangle + b, y_j)(f_j + \mu(v_j))(f_j + \mu(v_j))^\top + \lambda I_{Ld}$$

Then it holds that

$$\langle f_i + \mu(v_i), w^* \rangle = \operatorname{prox}_{\chi\ell(\cdot, y_i)}(\langle f_i + \mu(v_i), w_{\backslash i}^* \rangle)$$

where

$$\chi = \frac{1}{n} \left[ f_i^\top H_{\backslash i}^{-1} f_i + \mu(v_i)^\top H_{\backslash i}^{-1} \mu(v_i) + 2f_i^\top H_{\backslash i}^{-1} \mu(v_i) \right] \approx \frac{1}{n} \operatorname{tr}[H^{-1}].$$

We used that $\|\mu(v_i)\mu(v_i)^\top\|, \|\mu(v_i)f_i^\top\| \ll \|f_i f_i^\top\|$.

**Probabilistic analysis —** In similar fashion to the proof of Theorem 4, one can show that the parameter $\chi$ satisfies self-consistently

$$\frac{L}{\alpha\chi} = \mathbb{E}_{z,y,v}\left[ \frac{\ell''(z^*, y)}{1 + \ell''(z^*, y)\chi} \right] + \lambda.$$

where $z^* = \operatorname{prox}_{\chi\ell(\cdot, y)}(\nu z + m(v))$, with $m(v) := \langle \mu(v), w^* \rangle$. Using the stationarity condition

$$w^* = -\frac{1}{\lambda} \left[ \frac{1}{n} \sum_{j \in [n]} \ell'(\langle f_j, w^* \rangle + \langle \mu(v_j), w^* \rangle + b, y_j)(f_j + \mu(v_j)) \right]$$

allows to reach

$$\nu^2 = -\frac{1}{\lambda}\mathbb{E}_{y,z}\left[ \ell'(z^* + b, y)z^* \right]$$

and

$$m(v) = -\frac{1}{\lambda}\mathbb{E}_{y,v'}\left[\ell'(z^* + b, y)(\langle\mu(v)\mu(v')\rangle + \frac{m(v)}{\nu}z + \sqrt{\|\mu(v)\|^2 - m(v)^2/\nu^2}\omega)\right]$$
$$= -\frac{1}{\lambda}\mathbb{E}_{y,v'}\left[\ell'(z^* + b, y)(\langle\mu(v)\mu(v')\rangle + \frac{m(v)}{\nu}z)\right]$$

where $z \sim \mathcal{N}(0, 1)$. Finally

$$\phi(b) = \mathbb{E}\left[\ell(z^*, y)\right] + \frac{\lambda}{2}\nu^2.$$

This completes the derivation, but there is one further simplification. Let us introduce the unit vectors $\{e_k\}_{k\in L} \in \mathbb{R}^{dL}$, where the $kd + 1$ to $(k + 1)d$-th elements of $e_k$ correspond to $\theta\xi$, with all components otherwise zero. Note that all these vectors are orthogonal to each other. Then one can write

$$\mu(v) = \sum_{k\in[L]} v_k e_k.$$

Then, simply, one has

$$\langle\mu(v), \mu(v')\rangle = \theta^2\langle v, v'\rangle,$$

and

$$m(v) = \sum_{k\in[L]} v_k m_k$$

with $m_k := \langle w, e_k \rangle$. This simplifies the equation for $m(v)$, and yields the characterization of Proposition 3.

The theoretical predictions for the test and train errors of Proposition 3 are displayed in Fig. 4, and show a good agreement with numerical experiments performed in dimension $d = 1000$.

## H    PROOF OF COROLLARY 2

The full technical statement of Theorem 4, presented in Appendix F, and of Theorem 5, presented in Appendix G, provide a tight asymptotic characterization of learning errors in terms of a small set of summary statistics, characterized in turn as the solutions of a set of self-consistent equations. For the case of the square loss $\ell(y, z) = 1/2(y - z)^2$, in the limit of vanishing regularization $\lambda = 0^+$, these equations considerably simplify, making it possible to reach closed-form expressions for the test error in particular. These expressions are summarized in Corollary 2 in the main text. In this appendix, we provide the full technical statement. For ease of presentation, we break the statement into three proposition which we derive in succession, respectively for the attention model $\mathsf{A}_{q,w,b}$ (6), the pooled linear classifier $\mathsf{L}_{w,b}^{\text{pool}}$ (4) and the vectorized linear classifier $\mathsf{L}_{w,b}^{\text{vec}}$ (3).

### H.1    ATTENTION MODEL

**Proposition 4.** *From Theorem 4, in the asymptotic limit of Assumption 1, the test error of the attention model converges in probability to a limit $\mathcal{E}_{\text{test}}[\mathsf{A}]$. For the quadratic loss function $\ell(y, z) = 1/2(y - z)^2$, this quantity admits a well-defined limit in the limit $\lambda \to 0$. This limit admits the*

*expansion:*

$$\mathcal{E}_{\text{test}}[\mathsf{A}] = \mathcal{E}_{\text{test}}^{\infty}[\mathsf{A}] \tag{238}$$

$$+ \frac{1}{\alpha_1}(1-\pi)\mathbb{E}\left[\frac{e^{-\frac{1}{2}\left(\frac{\hat{b}^{\infty} + \langle g, s_- \rangle \mu_1^{\infty}}{\mu_3^{\infty}\|s_-\|}\right)^2}}{\sqrt{2\pi}} \frac{\delta\hat{b} + \langle g, s_- \rangle\delta\mu_1 - (\hat{b}^{\infty} + \langle g, s_- \rangle\mu_1^{\infty})\delta\mu_3/\mu_3^{\infty}}{\mu_3^{\infty}\|s_-\|}\right] \tag{239}$$

$$+ \frac{\pi}{\alpha_1}\mathbb{E}\left[\frac{e^{-\frac{1}{2}\left(\frac{-\hat{b}^{\infty} - \langle \theta v, s_+ \rangle \mu_2^{\infty} - \langle g, s_+ \rangle \mu_1^{\infty}}{\mu_3^{\infty}\|s_+\|}\right)^2}}{\sqrt{2\pi}}\right. \tag{240}$$

$$\left. \cdot \frac{\left(-\delta\hat{b} - \langle \theta v, s_+ \rangle\delta\mu_2 - \langle g, s_+ \rangle\delta\mu_1 + (\hat{b}^{\infty} + \langle \theta v, s_+ \rangle\mu_2^{\infty} + \langle g, s_+ \rangle\mu_1^{\infty})\delta\mu_3/\mu_3^{\infty}\right)}{\mu_3^{\infty}\|s_+\|}\right] \tag{241}$$

$$+ o\left(\frac{1}{\alpha_1}\right). \tag{242}$$

*The limiting error is*

$$\mathcal{E}_{\text{test}}^{\infty}[\mathsf{A}] = (1-\pi)\mathbb{E}_{g,s_+,s_-}\left[\Phi\left(\frac{\hat{b}^{\infty} + \langle g, s_- \rangle\mu_1^{\infty}}{\mu_3^{\infty}\|s_-\|}\right)\right] \tag{243}$$

$$+ \pi\mathbb{E}_{g,s_+,s_-}\left[\Phi\left(\frac{-\hat{b}^{\infty} - \langle \theta v, s_+ \rangle\mu_2^{\infty} - \langle g, s_+ \rangle\mu_1^{\infty}}{\mu_3^{\infty}\|s_+\|}\right)\right]. \tag{244}$$

*We introduced*

$$\begin{pmatrix} \mu_1^{\infty} \\ \mu_2^{\infty} \\ \hat{b}^{\infty} \end{pmatrix} = (I^{\infty})^{-1}J^{\infty}, \qquad \begin{pmatrix} \delta\mu_1 \\ \delta\mu_2 \\ \delta\hat{b} \end{pmatrix} = (I^{\infty})^{-1}\left(\delta J + \delta I \begin{pmatrix} \mu_1^{\infty} \\ \mu_2^{\infty} \\ \hat{b}^{\infty} \end{pmatrix}\right), \tag{245}$$

*where*

$$I^{\infty} = \begin{pmatrix} \mathbb{E}\left[c_q^2\right] & \mathbb{E}\left[c_q c_\xi\right] & \mathbb{E}\left[c_q\right] \\ \mathbb{E}\left[c_q c_\xi\right] & \mathbb{E}\left[c_\xi^2\right] & \mathbb{E}\left[c_\xi\right] \\ \mathbb{E}\left[c_q\right] & \mathbb{E}\left[c_\xi\right] & 1 \end{pmatrix}, \qquad J^{\infty} = \begin{pmatrix} \mathbb{E}\left[yc_q\right] \\ \mathbb{E}\left[yc_\xi\right] \\ 2\pi - 1. \end{pmatrix} \tag{246}$$

$$, \delta I = \frac{1}{\mathbb{E}\left[c_z^2\right]}\begin{pmatrix} \mathbb{E}\left[c_q^2 c_z^2\right] & \mathbb{E}\left[c_q c_\xi c_z^2\right] & \mathbb{E}\left[c_q c_z^2\right] \\ \mathbb{E}\left[c_q c_\xi c_z^2\right] & \mathbb{E}\left[c_\xi^2 c_z^2\right] & \mathbb{E}\left[c_\xi c_z^2\right] \\ \mathbb{E}\left[c_q c_z^2\right] & \mathbb{E}\left[c_\xi c_z^2\right] & \mathbb{E}\left[c_z^2\right] \end{pmatrix}, \quad \delta J = -\frac{1}{\mathbb{E}\left[c_z^2\right]}\begin{pmatrix} \mathbb{E}\left[yc_q c_z^2\right] \\ \mathbb{E}\left[yc_\xi c_z^2\right] \\ \mathbb{E}\left[yc_z^2\right]. \end{pmatrix} \tag{247}$$

*Finally, we denoted* $\delta\mu_3 = {}^1/\mu_3^{\infty}(1/2\nu^2 + \mu_1^{\infty}\delta\mu_1 + \mu_2^{\infty}\delta\mu_2 - \gamma\mu_1^{\infty}\delta\mu_2 - \gamma\mu_2^{\infty}\delta\mu_1) - {}^{\mu_1^{\infty}\delta\mu_1}/\mu_3^{\infty}$. *We remind that the joint law of* $c_z, c_\xi, c_q$ *is given in Lemma 1, and* $\gamma$ *is defined in Theorem 3.*

**Sketch of the derivation—** In what follows, we consider the case of quadratic loss

$$\ell(x; y) = \frac{1}{2}(yx - 1)^2.$$

In our problem, $\ell_i(x) = \ell(x_i + \epsilon_i; y_i)$. We have

$$\ell_i'(x) = x_i + \epsilon_i - y_i \qquad \text{and} \qquad \ell_i''(x) = 1.$$

Moreover, for this case, the proximal operator assumes a compact, closed-form expression

$$\text{Prox}_i(x; \gamma) = \frac{x}{1+\gamma} + \frac{\gamma}{1+\gamma}(y_i - \epsilon_i).$$

These closed-form expressions allow us to greatly simplify the self-consistent equations appearing in Theorem 4. Specifically, we can rewrite (88) as

$$\frac{1}{\alpha_1} = \mathbb{E}\left[\frac{c_z^2\chi}{1 + c_z^2\chi}\right] + \lambda\chi. \tag{248}$$

and

$$\nu^2 = \frac{\mathbb{E}\left[\frac{c_z^2 \chi (y-\epsilon)^2}{(1+c_z^2\chi)^2}\right]}{\lambda + \mathbb{E}\left[\frac{c_z^2}{(1+c_z^2\chi)^2}\right]}.$$

Let $\chi$ be the unique solution to (248). In the ridgeless limit (with $\lambda \to 0^+$), it is straightforward to check that

$$\lim_{\lambda \to 0^+} \lambda\chi = \frac{1}{\alpha_1} - 1, \qquad \text{for } \alpha_1 < 1.$$

and

$$\lim_{\lambda \to 0^+} \chi = \chi^*_{\text{ridgeless}}, \qquad \text{for } \alpha_1 > 1,$$

where $\chi^*_{\text{ridgeless}}$ is the unique solution to

$$\frac{1}{\alpha_1} = \mathbb{E}\left[\frac{c_{b,i}^2 \chi}{1 + c_{b,i}^2 \chi}\right].$$

We focus on the latter $\alpha_1 > 1$ case in the following. In the ridgeless limit, the fixed point equation for $\nu$ further simplifies to

$$\nu^2 = \frac{\mathbb{E}\left[\frac{c_z^2 \chi (y-\epsilon)^2}{(1+c_z^2\chi)^2}\right]}{\mathbb{E}\left[\frac{c_z^2}{(1+c_z^2\chi)^2}\right]}.$$

Then, the function $\phi(\mu_q, \mu_\xi, b)$ assumes the simple form

$$\phi(\mu_q, \mu_\xi, b) = \frac{1}{2}\mathbb{E}\left[\frac{(y-\epsilon)^2}{1 + c_z^2\chi}\right]. \tag{249}$$

Requiring that the gradients with respect to $\mu_q, \mu_\xi, b$ leads to the following characterization for the minimizers $\mu_1, \mu_2, \hat{b}$

$$I(\alpha_1)\begin{pmatrix}\mu_1 \\ \mu_2 \\ \hat{b}\end{pmatrix} = J(\alpha_1) \tag{250}$$

with

$$I(\alpha_1) = \begin{pmatrix} \mathbb{E}\left[\frac{c_q^2}{1+c_z^2\chi}\right] & \mathbb{E}\left[\frac{c_q c_\xi}{1+c_z^2\chi}\right] & \mathbb{E}\left[\frac{c_q}{1+c_z^2\chi}\right] \\ \mathbb{E}\left[\frac{c_q c_\xi}{1+c_z^2\chi}\right] & \mathbb{E}\left[\frac{c_\xi^2}{1+c_z^2\chi}\right] & \mathbb{E}\left[\frac{c_\xi}{1+c_z^2\chi}\right] \\ \mathbb{E}\left[\frac{c_q}{1+c_z^2\chi}\right] & \mathbb{E}\left[\frac{c_\xi}{1+c_z^2\chi}\right] & \mathbb{E}\left[\frac{1}{1+c_z^2\chi}\right] \end{pmatrix}, \qquad J(\alpha_1) = \begin{pmatrix} \mathbb{E}\left[\frac{y c_q}{1+c_z^2\chi}\right] \\ \mathbb{E}\left[\frac{y c_\xi}{1+c_z^2\chi}\right] \\ \mathbb{E}\left[\frac{y}{1+c_z^2\chi}\right] \end{pmatrix}. \tag{251}$$

Note that $I(\alpha_1)$ is the Gram matrix of the random variables $(c_q, c_\xi, 1)$ for the inner product $\langle a, b \rangle = \mathbb{E}\left[ab/1 + c_z^2\chi\right]$, and is thus invertible since the random variables are linearly independent

**Large $\alpha_1$ behavior** We now study in further detail the regime of large sample complexity $\alpha_1 \gg 1$. In this limit,

$$\chi = \frac{1}{\alpha_1 \mathbb{E}\left[c_z^2\right]} + o\left(\frac{1}{\alpha_1}\right) \tag{252}$$

while

$$\nu^2 = \frac{1}{\alpha_1}\frac{\mathbb{E}\left[c_z^2(y-\epsilon^\infty)^2\right]}{\mathbb{E}\left[c_z^2\right]^2} + o\left(\frac{1}{\alpha_1}\right). \tag{253}$$

Note that the limit $\nu^2 \xrightarrow{\alpha_1 \to \infty} 0$ implies that for large sample complexity, the readout weights lie in the span of $\xi, q$. We denote $\epsilon^\infty = c_q \mu_1^\infty + c_\xi \mu_2^\infty + \hat{b}^\infty$, with

$$\begin{pmatrix}\mu_1^\infty \\ \mu_2^\infty \\ \hat{b}^\infty\end{pmatrix} = (I^\infty)^{-1}J^\infty \tag{254}$$

where

$$I^\infty = \begin{pmatrix} \mathbb{E}\left[c_q^2\right] & \mathbb{E}\left[c_q c_\xi\right] & \mathbb{E}\left[c_q\right] \\ \mathbb{E}\left[c_q c_\xi\right] & \mathbb{E}\left[c_\xi^2\right] & \mathbb{E}\left[c_\xi\right] \\ \mathbb{E}\left[c_q\right] & \mathbb{E}\left[c_\xi\right] & 1 \end{pmatrix}, \qquad J^\infty = \begin{pmatrix} \mathbb{E}\left[yc_q\right] \\ \mathbb{E}\left[yc_\xi\right] \\ 2\pi - 1 \end{pmatrix}. \tag{255}$$

The corresponding residual test error is then simply given by adapting (85) to obtain

$$\mathcal{E}_{\text{test}} \xrightarrow{\alpha_1 \to \infty} \mathcal{E}_{\text{test}}^\infty = (1-\pi)\mathbb{E}_{g,s_+,s_-}\left[\Phi\left(\frac{\hat{b}^\infty + \langle g, s_-\rangle \mu_1^\infty}{\mu_3^\infty \|s_-\|}\right)\right] \tag{256}$$

$$+ \pi\mathbb{E}_{g,s_+,s_-}\left[\Phi\left(\frac{-\hat{b}^\infty - \langle \theta v, s_+\rangle \mu_2^\infty - \langle g, s_+\rangle \mu_1^\infty}{\mu_3^\infty \|s_+\|}\right)\right], \tag{257}$$

with $\mu_3^\infty = \left[1/1 - \gamma^2\left((\mu_1^\infty)^2 + (\mu_2^\infty)^2 - 2\gamma\mu_1^\infty\mu_2^\infty\right) - (\mu_1^\infty)^2\right]^{\frac{1}{2}}$. We now turn to ascertaining the leading correction. We introduce

$$\begin{pmatrix} \mu_1 \\ \mu_2 \\ \hat{b} \end{pmatrix} = \begin{pmatrix} \mu_1^\infty \\ \mu_2^\infty \\ \hat{b}^\infty \end{pmatrix} + \frac{1}{\alpha_1}\begin{pmatrix} \delta\mu_1 \\ \delta\mu_2 \\ \delta\hat{b} \end{pmatrix} + o\left(\frac{1}{\alpha_1}\right), \tag{258}$$

with

$$\begin{pmatrix} \delta\mu_1 \\ \delta\mu_2 \\ \delta\hat{b} \end{pmatrix} = (I^\infty)^{-1}\left(\delta J + \delta I \begin{pmatrix} \mu_1^\infty \\ \mu_2^\infty \\ \hat{b}^\infty \end{pmatrix}\right), \tag{259}$$

where we denote

$$\delta I = \frac{1}{\mathbb{E}\left[c_z^2\right]}\begin{pmatrix} \mathbb{E}\left[c_q^2 c_z^2\right] & \mathbb{E}\left[c_q c_\xi c_z^2\right] & \mathbb{E}\left[c_q c_z^2\right] \\ \mathbb{E}\left[c_q c_\xi c_z^2\right] & \mathbb{E}\left[c_\xi^2 c_z^2\right] & \mathbb{E}\left[c_\xi c_z^2\right] \\ \mathbb{E}\left[c_q c_z^2\right] & \mathbb{E}\left[c_\xi c_z^2\right] & \mathbb{E}\left[c_z^2\right] \end{pmatrix}, \quad \delta J = -\frac{1}{\mathbb{E}\left[c_z^2\right]}\begin{pmatrix} \mathbb{E}\left[yc_q c_z^2\right] \\ \mathbb{E}\left[yc_\xi c_z^2\right] \\ \mathbb{E}\left[yc_z^2\right]. \end{pmatrix} \tag{260}$$

Finally, let us denote $\delta\mu_3 = 1/\mu_3^\infty(1/2\nu^2 + \mu_1^\infty\delta\mu_1 + \mu_2^\infty\delta\mu_2 - \gamma\mu_1^\infty\delta\mu_2 - \gamma\mu_2^\infty\delta\mu_1) - \mu_1^\infty\delta\mu_1/\mu_3^\infty$. Then, the following asymptotic correction holds:

$$\mathcal{E}_{\text{test}} = \mathcal{E}_{\text{test}}^\infty \tag{261}$$

$$+ \frac{1}{\alpha_1}(1-\pi)\mathbb{E}\left[\frac{e^{-\frac{1}{2}\left(\frac{\hat{b}^\infty + \langle g, s_-\rangle \mu_1^\infty}{\mu_3^\infty \|s_-\|}\right)^2}}{\sqrt{2\pi}}\frac{\left(\delta\hat{b} + \langle g, s_-\rangle\delta\mu_1 - (\hat{b}^\infty + \langle g, s_-\rangle\mu_1^\infty)\delta\mu_3/\mu_3^\infty\right)}{\mu_3^\infty \|s_-\|}\right] \tag{262}$$

$$+ \frac{\pi}{\alpha_1}\mathbb{E}\left[\frac{e^{-\frac{1}{2}\left(\frac{-\hat{b}^\infty - \langle \theta v, s_+\rangle \mu_2^\infty - \langle g, s_+\rangle \mu_1^\infty}{\mu_3^\infty \|s_+\|}\right)^2}}{\sqrt{2\pi}}\frac{(-\delta\hat{b} - \langle \theta v, s_+\rangle\delta\mu_2 - \langle g, s_+\rangle\delta\mu_1 + (\hat{b}^\infty + \langle \theta v, s_+\rangle\mu_2^\infty + \langle g, s_+\rangle\mu_1^\infty)\delta\mu_3/\mu_3^\infty)}{\mu_3^\infty \|s_+\|}\right]$$

$$\tag{263}$$

$$+ o\left(\frac{1}{\alpha_1}\right). \tag{264}$$

## H.2 POOLED CLASSIFIER

**Proposition 5.** *From Theorem 4, in the asymptotic limit of Assumption 1, the test error of the pooled classifier model converges in probability to a limit $\mathcal{E}_{\text{test}}[\mathsf{L}^{\text{pool}}]$. For the quadratic loss function $\ell(y, z) = 1/2(y-z)^2$, this quantity admits a well-defined limit in the limit $\lambda \to 0$. This limit admits the expansion:*

$$\mathcal{E}_{\text{test}}[\mathsf{L}^{\text{pool}}] = \mathcal{E}_{\text{test}}^\infty[\mathsf{L}^{\text{pool}}] - (1-\pi)\frac{e^{-\frac{1}{2}\left(\frac{2\pi-1-\pi\mathcal{X}^2(1-\pi)}{2\pi\mathcal{X}(1-\pi)}\right)^2}}{2\sqrt{2\pi}}\left(\frac{2\pi-1-\pi\mathcal{X}^2(1-\pi)}{2\pi\mathcal{X}(1-\pi)}\right)\frac{\nu^2}{(\mu_2^\infty)^2} \tag{265}$$

$$+ \pi\frac{e^{-\frac{1}{2}\left(-\frac{2\pi-1+\pi\mathcal{X}^2(1-\pi)}{2\pi\mathcal{X}(1-\pi)}\right)^2}}{2\sqrt{2\pi}}\left(\frac{2\pi-1+\pi\mathcal{X}^2(1-\pi)}{2\pi\mathcal{X}(1-\pi)}\right)\frac{\nu^2}{(\mu_2^\infty)^2} + o\left(\frac{1}{\alpha_1}\right) \tag{266}$$

*The limiting error is*

$$\mathcal{E}_{\text{test}}^{\infty}[\mathsf{L}^{\text{pool}}] = (1 - \pi)\Phi\left(\frac{2\pi - 1 - \pi\mathcal{X}^2(1 - \pi)}{2\pi\mathcal{X}(1 - \pi)}\right) + \pi\Phi\left(-\frac{2\pi - 1 + \pi\mathcal{X}^2(1 - \pi)}{2\pi\mathcal{X}(1 - \pi)}\right). \quad (267)$$

*We denoted the signal-to-noise ratio* $\mathcal{X} = \theta R/\sqrt{L}$.

**Sketch of derivation —** We remind that the pooled classifier corresponds to setting the softmax inverse temperature in the attention model to zero, namely $\beta = 0$. In this limit, the joint distribution of the parameters $s_+, s_-, c_z, c_\xi, c_q$ detailed in (92) simplify to

$$s_+ = s_- = \frac{1_L}{L}, \quad \begin{pmatrix} c_q \\ c_\xi - \delta_{y,1}\frac{\theta R}{L} \end{pmatrix} \sim \mathcal{N}\left(0_2, \frac{1}{L(1 - \gamma^2)}\begin{bmatrix} 1 & -\gamma \\ -\gamma & 1 \end{bmatrix}\right), \quad c_z = \frac{1}{\sqrt{L}}. \quad (268)$$

Then, the limiting summary statistics $\mu_1^{\infty}, \mu_2^{\infty}, \hat{b}^{\infty}$ are given by $\mu_1^{\infty} = \gamma\mu_2^{\infty}$ and

$$\begin{pmatrix} \pi\mathcal{X}^2 + 1 & \pi\mathcal{X} \\ \pi\mathcal{X} & 1 \end{pmatrix}\begin{pmatrix} \mu_2^{\infty}/\sqrt{L} \\ \hat{b}^{\infty} \end{pmatrix} = \begin{pmatrix} \pi\mathcal{X} \\ 2\pi - 1 \end{pmatrix} \quad (269)$$

i.e.

$$\frac{\mu_2^{\infty}}{\sqrt{L}} = \frac{2\pi\mathcal{X}(1 - \pi)}{1 + \pi\mathcal{X}^2(1 - \pi)} \quad (270)$$

$$\hat{b}^{\infty} = \frac{2\pi - 1 - \pi\mathcal{X}^2(1 - \pi)}{1 + \pi\mathcal{X}^2(1 - \pi)} \quad (271)$$

The residual error then reads

$$\mathcal{E}_{\text{test}}^{\infty} = (1 - \pi)\Phi\left(\frac{\hat{b}}{\mu_2^{\infty}/\sqrt{L}}\right) + (1 - \pi)\Phi\left(\frac{-\hat{b} - \mathcal{X}\mu_2^{\infty}/\sqrt{L}}{\mu_2^{\infty}/\sqrt{L}}\right) \quad (272)$$

$$= (1 - \pi)\Phi\left(\frac{2\pi - 1 - \pi\mathcal{X}^2(1 - \pi)}{2\pi\mathcal{X}(1 - \pi)}\right) + \pi\Phi\left(-\frac{2\pi - 1 + \pi\mathcal{X}^2(1 - \pi)}{2\pi\mathcal{X}(1 - \pi)}\right). \quad (273)$$

We used the identity

$$\mathbb{E}_g\left[\Phi\left(\frac{a + bg}{c}\right)\right] = \mathbb{E}_{g,g'}\left[\mathbf{1}_{-a-bg+cg'\geq 0}\right] = \Phi\left(\frac{a}{\sqrt{b^2 + c^2}}\right). \quad (274)$$

Finally observe that $I = \alpha/1 + \alpha I^{\infty}, J = \alpha/1 + \alpha J^{\infty}$. As a consequence,

$$\begin{pmatrix} \mu_1 \\ \mu_2 \\ \hat{b} \end{pmatrix} = \begin{pmatrix} \mu_1^{\infty} \\ \mu_2^{\infty} \\ \hat{b}^{\infty} \end{pmatrix} \quad (275)$$

and

$$\nu^2 = \frac{L}{\alpha_1}\left(1 + \hat{b}^2 - 2\hat{b}(2\pi - 1) + \frac{(\mu_2^{\infty})^2}{L}(1 + \pi\mathcal{X}^2) + 2\mathcal{X}\pi(\hat{b} - 1)\frac{\mu_2^{\infty}}{\sqrt{L}}\right) \quad (276)$$

$$\mu_3 = \sqrt{1 - \gamma^2}\mu_2^{\infty} + \frac{L}{2\alpha_1}\frac{1 + \hat{b}^2 - 2\hat{b}(2\pi - 1) + \frac{(\mu_2^{\infty})^2}{L}(1 + \pi\mathcal{X}^2) + 2\mathcal{X}\pi(\hat{b} - 1)\frac{\mu_2^{\infty}}{\sqrt{L}}}{\sqrt{1 - \gamma^2}\mu_2^{\infty}} \quad (277)$$

$$+ o\left(\frac{1}{\alpha_1}\right). \quad (278)$$

It follows that the leading order correction to the test error reads

$$\mathcal{E}_{\text{test}} = \mathcal{E}_{\text{test}}^{\infty} \quad (279)$$

$$- (1 - \pi)\frac{e^{-\frac{1}{2}\left(\frac{2\pi - 1 - \pi\mathcal{X}^2(1 - \pi)}{2\pi\mathcal{X}(1 - \pi)}\right)^2}}{2\sqrt{2\pi}}\left(\frac{2\pi - 1 - \pi\mathcal{X}^2(1 - \pi)}{2\pi\mathcal{X}(1 - \pi)}\right)\frac{\nu^2}{(\mu_2^{\infty})^2} \quad (280)$$

$$+ \pi\frac{e^{-\frac{1}{2}\left(-\frac{2\pi - 1 + \pi\mathcal{X}^2(1 - \pi)}{2\pi\mathcal{X}(1 - \pi)}\right)^2}}{2\sqrt{2\pi}}\left(\frac{2\pi - 1 + \pi\mathcal{X}^2(1 - \pi)}{2\pi\mathcal{X}(1 - \pi)}\right)\frac{\nu^2}{(\mu_2^{\infty})^2} + o\left(\frac{1}{\alpha_1}\right). \quad (281)$$

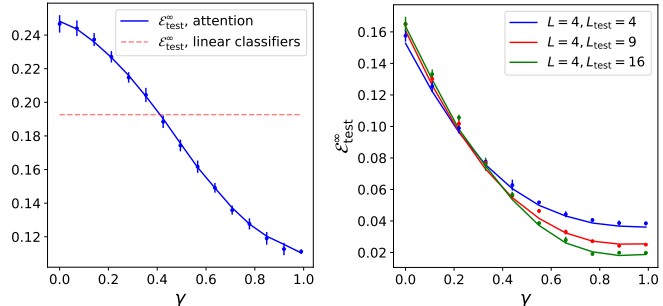

Figure 5: (**left**) Residual error $\mathcal{E}_{\text{test}}^{\infty}$ in the $\alpha_1 \to \infty$ limit as a function of the alignment $\gamma = \langle q, \xi \rangle$ between the attention query weights and the signal vector, for the attention model (blue) and the linear classifiers (dashed red), trained with the quadratic loss at vanishing regularization. $L = 5, R = 1, \theta = 3, \pi = 0.75$. Solid lines correspond to the theoretical characterizations (256) and (272). Dots correspond to numerical simulations in dimension $d = 100$, and large number of samples $n = 10^5$, averaged over 10 trials, with error bars representing one standard deviation. (**right**) Residual error of the attention model for different training lengths $L, R = L^{1/2}$ and test-time lengths $L_{\text{test}}, R_{\text{test}} = L_{\text{test}}^{1/2}$, and $\theta = 3$

.

**Comparison with the attention model** We contrast in Fig. 5 the residual errors $\mathcal{E}_{\text{test}}^{\infty}$ achieved in the limit of large sample complexity $\alpha_1 \gg 1$ by the attention-based and linear classifiers. As we detail in Appendix G, the vectorized and pooled linear classifiers share identical residual test errors. Interestingly, for a small alignment $\gamma$ between the attention query weights $q$ and the signal vector $\xi$, the attention model performs worse than the linear classifiers, as the discrepancy between $q, \xi$ can cause the model to spuriously privilege tokens devoid of signal.

## H.3 VECTORIZED CLASSIFIER

**Proposition 6.** *From Theorem 5, in the asymptotic limit of Assumption 1, the test error of the vectorized classifier model converges in probability to a limit $\mathcal{E}_{\text{test}}[\mathsf{L}^{\text{vec}}]$. For the quadratic loss function $\ell(y, z) = 1/2(y - z)^2$, this quantity admits a well-defined limit in the limit $\lambda \to 0$. This limit admits the expansion:*

$$\mathcal{E}_{\text{test}}[\mathsf{L}^{\text{vec}}] = \mathcal{E}_{\text{test}}^{\infty}[\mathsf{L}^{\text{vec}}] + \left\{ \pi \frac{e^{-\frac{1}{2}\left(-\mathcal{X} - \frac{b^{\infty}}{\nu^{\infty}}\right)^2}}{\sqrt{2\pi}} \frac{b^{\infty} + \mathcal{X}\nu^{\infty}}{2(\nu^{\infty})^3} - (1-\pi) \frac{e^{-\frac{1}{2}\left(\frac{b^{\infty}}{\nu^{\infty}}\right)^2}}{\sqrt{2\pi}} \frac{b^{\infty}}{2(\nu^{\infty})^3} \right\} \tag{282}$$

$$\cdot \left( \frac{1 + \pi\mathcal{X}^2 - \pi^2\mathcal{X}^2(1 - b^*)}{1 + \pi\mathcal{X}^2} + (b^{\infty})^2 - 2(2\pi - 1)b^{\infty} \right) \frac{L}{\alpha_1} + o\left(\frac{1}{\alpha_1}\right), \tag{283}$$

*with*

$$(\nu^{\infty})^2 = \left( \frac{2\pi\mathcal{X}(\pi - 1)}{1 + \pi(1 - \pi)\mathcal{X}^2} \right)^2, \qquad b^{\infty} = \frac{2\pi - 1 - \pi(1 - \pi)\mathcal{X}^2}{1 + \pi(1 - \pi)\mathcal{X}^2}. \tag{284}$$

*The limiting error is*

$$\mathcal{E}_{\text{test}}^{\infty}[\mathsf{L}^{\text{vec}}] = (1 - \pi)\Phi\left( \frac{2\pi - 1 - \pi\mathcal{X}^2(1 - \pi)}{2\pi\mathcal{X}(1 - \pi)} \right) + \pi\Phi\left( -\frac{2\pi - 1 + \pi\mathcal{X}^2(1 - \pi)}{2\pi\mathcal{X}(1 - \pi)} \right). \tag{285}$$

*We denoted the signal-to-noise ratio $\mathcal{X} = \theta R/\sqrt{L}$.*

**Sketch of derivation —** For the quadratic loss and vanishing regularization, the fixed point equations of Proposition 3 simplify to

$$m = \frac{\theta^2 p(1 - b^*)}{1 + \theta^2 p(1 + (L - 1)\rho)} \tag{286}$$

$$\nu^2 = \chi \frac{1 + \theta^2 p + \theta^2 (L - 1)p\rho - \theta^2 Lp^2(1 - b^*)^2}{1 + \theta^2 p(1 + (L - 1)\rho)} + \frac{\theta^2 Lp^2(1 - b^*)^2}{(1 + \theta^2 p(1 + (L - 1)\rho))^2} \tag{287}$$

$$- 2\chi(2\pi - 1)b^* + (b^*)^2\chi \tag{288}$$

$$\chi = \frac{L}{\alpha - L} \tag{289}$$

where $p = \frac{\pi R}{L}$ and

$$\rho = \delta_{R \geq 2} \frac{R(R - 1)}{L(L - 1)} \frac{\pi}{p}, \qquad b^* = 1 + \frac{(2\pi - 2)A}{A - \theta^2 Lp^2}. \tag{290}$$

We used a shorthand $A := 1 + \theta^2 p(1 + (L - 1)\rho)$. These expression are amenable to being more compactly rewritten, introducing the $\mathcal{X}$ introduced in Theorem 1. We remind that in the current setting, $\mathcal{X}$ admits the compact expression

$$\mathcal{X} = \frac{\theta R}{\sqrt{L}}. \tag{291}$$

The self-consistent equations then simplify to

$$b^* = 1 + \frac{(2\pi - 2)(1 + \pi \mathcal{X}^2)}{1 + \pi(1 - \pi)\mathcal{X}^2} \tag{292}$$

$$m = \frac{1}{R} \frac{\pi \mathcal{X}^2(1 - b^*)}{1 + \pi \mathcal{X}^2} \tag{293}$$

$$\nu^2 = \chi \frac{1 + \pi \mathcal{X}^2 - \pi^2 \mathcal{X}^2(1 - b^*)}{1 + \pi \mathcal{X}^2} + \frac{\pi^2 \mathcal{X}^2(1 - b^*)^2}{(1 + \pi \mathcal{X}^2)^2} - 2\chi(2\pi - 1)b^* + (b^*)^2\chi. \tag{294}$$

$\alpha_1 \to \infty, \mathcal{X} = O(1), \alpha_1 \gg L$ **regime —** Following a similar derivation as the ones detailed in the previous subsections, the test error is found to admit the large $\alpha_1$ residual

$$\mathcal{E}_{\text{test}}^{\infty} = \pi \Phi\left(-\mathcal{X} - \frac{b^\infty}{\nu^\infty}\right) + (1 - \pi)\Phi\left(\frac{b^\infty}{\nu^\infty}\right) \tag{295}$$

with

$$(\nu^\infty)^2 = \frac{\pi^2 \mathcal{X}^2(1 - b^\infty)^2}{(1 + \pi \mathcal{X}^2)^2} = \left(\frac{2\pi \mathcal{X}(\pi - 1)}{1 + \pi(1 - \pi)\mathcal{X}^2}\right)^2, \tag{296}$$

$$b^\infty = 1 + \frac{(2\pi - 2)(1 + \pi \mathcal{X}^2)}{1 + \pi(1 - \pi)\mathcal{X}^2} = \frac{2\pi - 1 - \pi(1 - \pi)\mathcal{X}^2}{1 + \pi(1 - \pi)\mathcal{X}^2}, \tag{297}$$

and the asymptotic expansion

$$\mathcal{E}_{\text{test}} = \mathcal{E}_{\text{test}}^{\infty} + \left\{ \pi \frac{e^{-\frac{1}{2}\left(-\mathcal{X} - \frac{b^\infty}{\nu^\infty}\right)^2}}{\sqrt{2\pi}} \frac{b^\infty + \mathcal{X}\nu^\infty}{2(\nu^\infty)^3} - (1 - \pi)\frac{e^{-\frac{1}{2}\left(\frac{b^\infty}{\nu^\infty}\right)^2}}{\sqrt{2\pi}} \frac{b^\infty}{2(\nu^\infty)^3} \right\} \tag{298}$$

$$\cdot \left(\frac{1 + \pi \mathcal{X}^2 - \pi^2 \mathcal{X}^2(1 - b^*)}{1 + \pi \mathcal{X}^2} + (b^\infty)^2 - 2(2\pi - 1)b^\infty\right) \frac{L}{\alpha_1} + o\left(\frac{1}{\alpha_1}\right) \tag{299}$$

**Remark 9** (Comparison with the pooled model). *Note that the residual error $\mathcal{E}_{\text{test}}^{\infty}$ can be explicitly expressed as*

$$\mathcal{E}_{\text{test}}^{\infty} = (1 - \pi)\Phi\left(\frac{(2\pi - 2)(1 + \pi \mathcal{X}^2)}{2\pi \mathcal{X}(1 - \pi)}\right) + \pi \Phi\left(\frac{2\pi - 1 + \pi \mathcal{X}^2(1 - \pi)}{2\pi \mathcal{X}(1 - \pi)}\right). \tag{300}$$

*This incidentally corresponds to the residual error achieved by the pooled classifier trained with ridgeless quadratic loss* (272), *since for the considered data distribution* $\mathcal{X} = \mathcal{X} = \theta R/\sqrt{L}$. *We also furthermore have a similar correspondence at the level of the summary statistics, namely* $b^\infty = \hat{b}^\infty, \nu^\infty = \mu_2^\infty/\sqrt{L}$, *where* $\hat{b}^\infty, \mu_2^\infty/\sqrt{L}$ *are defined for the pooled model in* (270). *Furthermore, the leading order corrections are related by a simple factor L:*

$$\frac{\mathcal{E}_{\text{test,vector}} - \mathcal{E}_{\text{test}}^\infty}{\mathcal{E}_{\text{test,pool}} - \mathcal{E}_{\text{test}}^\infty} = L + o(1). \tag{301}$$

Note that a consequence of Remark 9 is that in the $\alpha \to \infty$ limit, for ridgeless regression with a quadractic loss, the pooled and vectorized models converge to the same solution, in the sense that the weights of the vectorized model correspond to that of the pooled model stacked $L$ times. Both models furthermore yield the same limiting test error. Let us also comment that Arnaboldi et al. (2025) also observe a similar speed up between related flattened and pooled models learning from sequential data, in a related task, in terms of weak recovery time. The result of Remark 9 instead bears on the coefficient of the leading asymptotic correction in terms of sample complexity.

**Remark 10.** *We note that the joint limit* $\alpha_1, L \to \infty, \mathfrak{b} = \alpha/L = O(1), \mathcal{X} = O(1)$ *can also be analyzed, and is simply given by equations* (292) *setting*

$$\chi = \frac{1}{\mathfrak{b} - 1} \tag{302}$$

**Study of the** $\alpha_1 \to \infty$ **residual error** We now examine the behaviour of the residual error $\mathcal{E}_{\text{test}}^\infty$ with the signal-to-noise ratio $\mathcal{X}$. We first examine the case $\mathcal{X} \to \infty$. In this limit,

$$b^\infty = -1 + o(1), \qquad (\nu^\infty)^2 = \frac{4}{\mathcal{X}^2} + o\left(\frac{1}{\mathcal{X}^2}\right) \tag{303}$$

The residual error then decays to zero as

$$\mathcal{E}_{\text{test}}^\infty \asymp \sqrt{\frac{2}{\pi}} \frac{e^{-\frac{\mathcal{X}^2}{8}}}{\mathcal{X}}. \tag{304}$$

In the opposite limit of small signal $\mathcal{X} \to 0$,

$$b^\infty = 2\pi - 1 + o(1), \qquad (\nu^\infty)^2 = 4\pi^2(1 - \pi)^2 \mathcal{X}^2 + o(\mathcal{X}^2). \tag{305}$$

Then

$$\mathcal{E}_{\text{test}}^\infty \xrightarrow{\mathcal{X} \to 0} \min(\pi, 1 - \pi). \tag{306}$$

These limiting errors stand in coherence with Theorem 1.

# I  DERIVATION OF CONJECTURE 1

As a corollary of Theorem 4, we derive in the appendix the *capacity* of the three considered models, namely the largest number of samples that can typically be perfectly classified, up to vanishing training error. The corresponding separability threshold was characterized in the seminal work of Cover (2006), and revisited in many later works, e.g. Gardner & Derrida (1988); Krauth & Mézard (1989); Candès & Sur (2020); Mignacco et al. (2020a). Note that at the level of the representations $f_{\text{vec}}(\cdot), f_q(\cdot), f_{\text{pool}}(\cdot)$, the capacity intuitively reflects how well the representations separate positive and negative samples in feature space, with a larger separability thresholds signaling more markedly separated classes.

**Definition 3** (Separability threshold). *Consider the empirical risk minimization problem* (14) *for the attention model, or the related problem for the linear classifier models, with logistic loss* $\ell(z; y) = \log(1 + e^{-yz})$ *and vanishing regularization* $\lambda = 0^+$. *As stated in Theorem 4, the training loss converges in probability in the considered asymptotic limit to a limit* $\mathcal{E}_{\text{train}}$. *We define the separability threshold* $\alpha^\star$ *of the model as*

$$\alpha^\star = \sup \{\alpha \geq 0 \mid \mathcal{E}_{\text{train}} = 0\}. \tag{307}$$

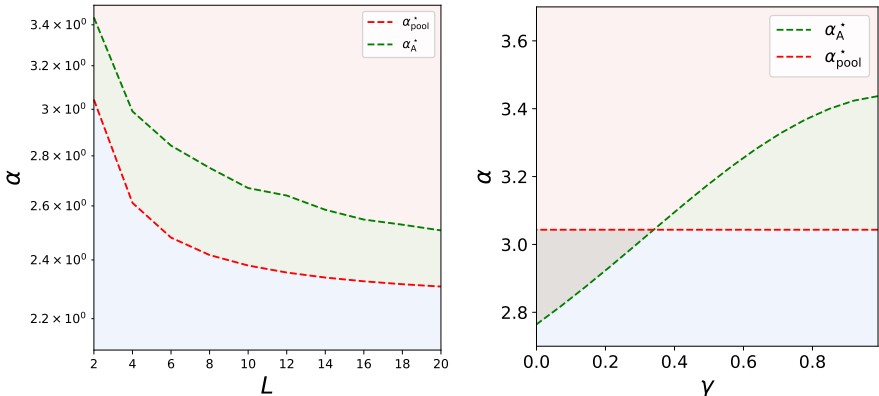

Figure 6: Separability thresholds for the attention model (green), and the pooled (red) and vectorized (blue) linear classifiers, as given in Conjectures 2, 3 and 2, as a function of the sequence length $L$ (left) and the attention query/signal cosine similarity $\gamma$ (right). Left: $\theta = 2, \pi = 0.3, \gamma = 1$, and $R = 1$ is kept fixed while $L$ is increased. Right: $\theta = 2, \pi = 0.3, L = 2, R = 1$ and $\gamma$ is varied.

A closed-form characterization of the separability threshold $\alpha^\star$ can be heuristically derived from Theorem 4 for each of the three models. We first provide the characterization for the vectorized classifier.

**Conjecture 2** (Separability threshold for the vectorized classifier). *The separability threshold for the vectorized classifier is equal to*

$$\alpha^\star_{\text{vec}} = \max_{s \in [0,1], \mathfrak{b}} \frac{L(1 - s^2)}{\int_0^\infty [\pi \Phi' (\mathfrak{b} + \mathcal{X}s + u) + (1 - \pi)\Phi' (u - \mathfrak{b})] u^2 du} \tag{308}$$

*We have used the shorthand $\mathcal{X} = {}^{\theta R}/\sqrt{L}$.*

*Proof.* First note that the following identity follows from Proposition 3, and most conveniently seen from the rewriting of Remark 8:

$$\nu^2 - \frac{L}{\theta^2}m^2 = \frac{\alpha}{L}\mathbb{E}\left[\ell'(z^* + b, y)^2\right]\chi^2 \tag{309}$$

for any given $b$. We assume that the loss function is of the form $\ell(z, y) = \tilde{\ell}(yz)$, and satisfies $\lim_{z \to \infty} \tilde{\ell}(z) = 0$, while being convex. We assume $\tilde{\ell}$ to be decreasing, with a monotonically increasing and negative derivative satisfying $\lim_{z \to \infty} \tilde{\ell}'(z) = 0^-$. We denote $\kappa = -\lim_{z \to -\infty} \tilde{\ell}'(z)$, which we assume to be finite. Note that all those assumptions are satisfied in particular by the logistic loss function. We again assumed all token locations are symmetric, leading to a solution $m_k = m$ for all $k \in [L]$. Introducing the cosine similarity $s = \sqrt{L}m/\theta\nu \in [0, 1]$, and the normalized quantities $\gamma = \chi/\nu, \mathfrak{b} = {}^b/\nu$, and introducing the random variable $u = \tilde{\ell}'((z^* + b)y)$

$$1 - s^2 = \frac{\alpha}{L}\gamma^2 \mathbb{E}\left[u^2\right]. \tag{310}$$

But $z^* - \delta_{y,1}Rm - \nu z + \chi yu = 0$ by definition of the proximal operator, and $z^* = y\tilde{\ell}^{-1}(u) - b$, while $z \sim \mathcal{N}(0, 1)$. Furthermore, $u \in (-\kappa, 0)$ Thus

$$\mathbb{E}\left[u^2\right] = -2\int_{-\kappa}^0 \left[\pi\Phi\left(\frac{\tilde{\ell}^{-1}(u) - b - Rm + \chi u}{\nu}\right) + (1 - \pi)\Phi\left(\frac{\tilde{\ell}^{-1}(u) + b + \chi u}{\nu}\right)\right] u\, du \tag{311}$$

$$= -2\int_{-\kappa}^0 \left[\pi\Phi\left(\frac{\tilde{\ell}^{-1}(u)}{\nu} - \mathfrak{b} - \mathcal{X}s + \gamma u\right) + (1 - \pi)\Phi\left(\frac{\tilde{\ell}^{-1}(u)}{\nu} + \mathfrak{b} + \gamma u\right)\right] u\, du. \tag{312}$$

Following Mignacco et al. (2020b) we aim to determine the necessary conditions on $\alpha$ such that there exists a solution satisfying $\nu = \infty, \gamma = \infty$ – which should hold for a solution achieving zero training loss. We conjecture the following limit

$$\lim_{\gamma,\nu\to\infty} \gamma^2 \mathbb{E}\left[u^2\right] = 2 \int_0^\infty \left[\pi\Phi\left(-\mathfrak{b} - \mathcal{X}s - u\right) + (1-\pi)\Phi\left(\mathfrak{b} - u\right)\right] u du \tag{313}$$

$$= \int_0^\infty \left[\pi\Phi'\left(\mathfrak{b} + \mathcal{X}s + u\right) + (1-\pi)\Phi'\left(u - \mathfrak{b}\right)\right] u^2 du \tag{314}$$

where we remind that $\Phi'$ is simply a standard Gaussian density. Then, a necessary condition for the existence of a solution with $\nu = \infty, \gamma = \infty$ is the existence of an $s \in [0,1]$ so that

$$\alpha = \frac{L(1 - s^2)}{\int_0^\infty \left[\pi\Phi'\left(\mathfrak{b} + \mathcal{X}s + u\right) + (1-\pi)\Phi'\left(u - \mathfrak{b}\right)\right] u^2 du} \tag{315}$$

$\square$

Note that the pooled classifier can be mapped to a special case of the vectorized classifier, formally evaluating the expression for the vectorized classifier for $L, R \to 1, \theta \to \theta R/\sqrt{L}$. Leveraging this connection yields the following conjecture.

**Conjecture 3** (Separability threshold for the pooled classifier). *The separability threshold for the pooled classifier is equal to*

$$\alpha_{\mathsf{pool}}^\star = \max_{s\in[0,1],\mathfrak{b}} \frac{(1 - s^2)}{\int_0^\infty \left[\pi\Phi'\left(\mathfrak{b} + \mathcal{X}s + u\right) + (1-\pi)\Phi'\left(u - \mathfrak{b}\right)\right] u^2 du} = \frac{\alpha_{\mathsf{vec}}^\star}{L}. \tag{316}$$

Finally, a similar characterization can be conjectured from Theorem 4 for the attention model.

**Conjecture 4** (Separability threshold for the attention model). *The separability threshold for the pooled classifier is equal to*

$$\alpha_{\mathsf{A}}^\star = \max_{m_q,m_\xi,\mathfrak{b}} \frac{1}{\mathbb{E}_{y,c_z,c_\xi,c_q}\left[c_z^3 \int_0^\infty \Phi'\left(\frac{c_z^2 u + y(b + c_q m_q + c_\xi m_\xi)}{c_z}\right) u^2 du\right]} \tag{317}$$

*Proof.* The derivation proceeds in close likeness to that for the vectorized classifiers. First observe that from Theorem 4, the following identity holds:

$$\nu^2 = \alpha\chi^2 \mathbb{E}\left[c_z^2 \ell'(z^* + c_q\mu_q + c_\xi\mu_\xi + b, y)^2\right] \tag{318}$$

Introducing the normalized quantities $\gamma = \chi/\nu, \mathfrak{b} = b/\nu, m_q = \mu_q/\nu, m_\xi = \mu_\xi/\nu$, and introducing the random variable $u = \tilde{\ell}'((z^* + c_q\mu_q + c_\xi\mu_\xi + b)y)$, this identity can be compactly rewritten as

$$1 = \alpha\gamma^2 \mathbb{E}\left[c_z^2 u^2\right]. \tag{319}$$

But $z^* - c_z\nu z + c_z^2\chi yu = 0$ by definition of the proximal operator, and $z^* = y\tilde{\ell}^{-1}(u) - b - c_q\mu_q - c_\xi\mu_\xi$, while $z \sim \mathcal{N}(0,1)$. Thus

$$\mathbb{E}\left[c_z^2 u^2\right] = -2 \int_{-\kappa}^0 \mathbb{E}\left[c_z^2 \Phi\left(\frac{\frac{\tilde{\ell}^{-1}(u)}{\nu} + c_z^2\gamma u - y(\mathfrak{b} - c_q m_q - c_\xi m_\xi)}{c_z}\right) u du\right]. \tag{320}$$

Again, following Mignacco et al. (2020b) we aim to determine the necessary conditions on $\alpha$ such that there exists a solution satisfying $\nu = \infty, \gamma = \infty$. This assumption is further motivated by numerical experiments, as illustrated in Fig. 7, where $\nu, \gamma$ are observed to diverge as $\lambda \to 0$. Then,

$$\lim_{\gamma,\nu\to\infty} \gamma^2 \mathbb{E}\left[c_z^2 u^2\right] = 2 \int_0^\infty \mathbb{E}\left[c_z\Phi\left(c_z^2 u - \frac{y(\mathfrak{b} - c_q m_q - c_\xi m_\xi)}{c_z}\right) u du\right]$$

$$= \int_0^\infty \mathbb{E}\left[c_z^3\Phi'\left(c_z u - \frac{y(\mathfrak{b} - c_q m_q - c_\xi m_\xi)}{c_z}\right) u^2 du\right], \tag{321}$$

which concludes the derivation. $\square$

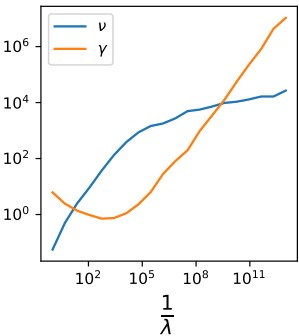

Figure 7: Parameters $\gamma, \nu$ involved in the derivation of Conjecture 1, for the attention model trained with the logistic loss, as a function of the regularization $\lambda$. The curves correspond to numerical experiments in dimension $d = 1000$, averaged over 3 trials. The problem parameters are $L = 2, R = 1, \theta = 2, \pi = 0.3$.

The theoretical prediction of Conjectures 2, 3 and 4 are contrasted with numerical experiments in Fig. 1, revealing a good agreement with the point where the training error – defined as the fraction of misclassified training samples – ceases to be zero. Note interestingly that the separability thresholds $\alpha^{\star}_{\text{vec,pool}}$ for the vectorized and pooled classifiers are related by a factor $L$. The latter can be rationalized by the fact that the vectorized classifiers operates in $\mathbb{R}^{Ld}$, while the pooled classifier acts on the smaller space $\mathbb{R}^{d}$. Moreover, observe that while the threshold $\alpha^{\star}_{\text{A}}$ for the attention model lies for large query/signal alignment $\gamma$ above $\alpha^{\star}_{\text{pool}}$, it becomes smaller for small values of $\gamma$ (see Fig.6, right). This temptingly suggests the intuitive interpretation that when the internal representation of the attention is misaligned with the signal, the attention model displays a smaller capacity than the simple pooled linear classifier. This conclusion echoes a similar observation at the level of the residual errors, see the discussion of Fig. 5 and its discussion in Appendix F.

