# OpenReview forum: "High-Dimensional Analysis of Single-Layer Attention for Sparse-Token Classification"
_ICLR.cc/2026/Conference — ICLR 2026 Poster_

### Official Review · Reviewer_sbqr · 2025-10-28

**Soundness:** 4
**Presentation:** 4
**Contribution:** 3
**Rating:** 8
**Confidence:** 4

**Summary:**

This work presents a theoretical analysis of a sparse-token classification task, in the high dimensional scaling limit of number of samples and embedding dimension, under the information theoretic limit of $n/d = \Theta(1)$. For a simple attention model, the precise asymptotics of the test error under an optimization procedure is shown, for a given finite sequence length $L$, as well as a fine-grained analysis of how small the signal-to-noise ratio can be. To complement this, a lower bound is given for linear feedforward models, matching intuitions from previous works.

**Strengths:**

- The results are very rigorous, and hold under the high-dimensional limit of $n,d$ for a given sequence length, and follow upon a line of works analyzing the fundamental task for attending to sparse token signals.
- The analysis shows that two gradient steps suffice for learning the hidden representation of the query matrix, which matches intuitions for weak recovery of the signal from other high dimensional settings such as multi-index models and such.
- Lower bounds are given for linear models, in the sense that the signal strength requires $\sqrt{L}$ scaling compared to $\log L$ in the attention model.

**Weaknesses:**

See questions below.

**Questions:**

- Finite sample/dimensional guarantees are given - what are the bottlenecks to extending the analysis to this setting?
- A natural question is whether or not the learned attention model length generalizes to longer length, as shown in the regression setting of Wang et al. (2024). I suspect that it does up to some reasonable upper bound - could the authors comment on this?

---

> ### Author Response · Authors · 2025-11-20
> **Reviewer 4 (sbqr) response**
>
> We thank the reviewer for their appreciation of our work and the insightful suggestions. We answer their questions below.
>
> Question 1: "Finite sample/dimensional guarantees are [not] given - what are the bottlenecks to extending the analysis to this setting?"
>
> Response: Our analyses suggest that the discrepancy between our asymptotic predictions for the test and training errors and their empirical values is of order $O(\mathrm{polyLog}(n_1)/\sqrt{n_1})$; see, for example, Lemmas 13, 15, and 16 (proof). From this, one may construct confidence intervals for the finite-sample estimates by taking a band of width $O(\mathrm{polyLog}(n_1)/\sqrt{n_1})$ around the asymptotically exact quantities $\mathcal{E}{\mathsf{test}}$ and $\mathcal{E}{\mathsf{train}}$ in Theorem 4. To streamline the presentation, we chose not to explicitly track these vanishing error terms in the proofs. Empirical results in Figure 1 indicate good agreement between theory and simulation even for moderate dimension $d = 1000$.
>
> Question 2: "A natural question is whether or not the learned attention model length generalizes to longer length, as shown in the regression setting of Wang et al. (2024). I suspect that it does up to some reasonable upper bound - could the authors comment on this?"
>
> Response:  We thank the reviewer for this insightful question. Although we did not explicitly discuss this point, the theoretical results extend with minimal modification to settings where test samples have a different sequence length $L’$ from the training length $L$. The only change is that the characterization in equation (82) now involves an expectation over an $L’$-dimensional Gaussian vector $g$, while all other parameters remain unchanged and are still characterized as in Theorem 4 and Lemma 1. We have summarized this observation in a new Remark 1 in Appendix E. In addition, we have added a new panel to Fig. 5 (right) showing the theoretical prediction for the test error of an attention model trained on sequences of length $L=4$ and evaluated on sequences of length $L’ = 4, 9, 16$. The plot shows that the model generalizes well to longer sequences at test time.

---

### Official Review · Reviewer_4vxC · 2025-11-01

**Soundness:** 3
**Presentation:** 3
**Contribution:** 3
**Rating:** 6
**Confidence:** 3

**Summary:**

This paper presents theoretical results regarding the learnability and representational capacity of a single attention layer on the task of sparse token classification. Compared to two linear model baselines, the attention model is shown to detect signals exponentially weaker in the limit of large sequence length, and is proved to require only two gradient steps for learning meaning representations that align with the hidden signal. Further, it provides a precise characterization of the test errors achieved by the attention model, highlighting a significant gap of the representation power of linear and the attention models. Overall, it presents solid theoretical evidence for the superior performance of the attention model in the sparse token classification task.

**Strengths:**

1. From my knowledge, it is a novel theory work that studies attention in the task of sparse token classification. It shows a clear advantage of the attention model in terms of representation capacity and learnability in this task.
2. It extends existing literature beyond square loss to general convex losses.
3. This paper is well-written and clearly presented.

**Weaknesses:**

1. The attention model considered is rather simplified, without multi-heads and separate K, V states. While this benefits theoretical analysis, there exists a clear gap between this simplified model and real practice.
2. While a minor weakness, the capacity result remains a conjecture rather than full proof.
3. Some experiments, even synthetic ones, could further strengthen the work and provide more evidence for the real-world applicability of the presented results.

**Questions:**

Please see the weaknesses section.

---

> ### Author Response · Authors · 2025-11-20
> **Reviewer 3 (4vxC) response**
>
> We thank the reviewer for their constructive questions, which we answer below.
>
> Weakness 1: "The attention model considered is rather simplified, without multi-heads and separate K, V states. While this benefits theoretical analysis, there exists a clear gap between this simplified model and real practice."
>
> Response: We acknowledge that these simplifications represent limitations of current theoretical analyses of transformers, shared by many contemporary works. Similar single-layer models have been the focus of several recent studies (e.g., Marion et al., 2024; Tarzanagh et al., 2023; Oymak et al., 2023). The ERM we analyze, while simplified, accommodates a finite number of samples and a broad class of convex losses, thereby complementing prior works that focus on the population setting and the square loss (e.g., Wang et al., 2024; Marion et al., 2024). Developing a theory that fully captures the complexity of modern transformer architectures remains an ambitious and long-term goal that the theory community will reach only through a sequence of progressive advances. We therefore believe our manuscript should be assessed in light of how it advances the theoretical understanding of attention-based mechanisms, rather than by its proximity to current empirical deep learning practice.
>
> Weakness 2: "While a minor weakness, the capacity result remains a conjecture rather than full proof."
>
> Response: The heuristic assumption $\gamma, \nu \to \infty$ as $\lambda \to 0$ in the derivation of Conjecture 1 is motivated by the following intuition. The logistic loss is minimized when its scalar argument tends to infinity, in which regime its second derivative vanishes. When the dataset is linearly separable and the regularization parameter $\lambda$ approaches zero, this implies that the norm $\nu$ of the estimator diverges. At the same time, as the second derivatives of the loss become small in this limit, the trace of the inverse Hessian—proportional to $\gamma$—also diverges. These assumptions are further corroborated by numerical experiments: as shown in the new Figure 7, both $\gamma$ and $\nu$ are observed to diverge as $\lambda \to 0$, consistent with the theory.
>
> Similar behavior has been rigorously established in simpler settings, for example by Mignacco et al. (2020), who analyzed logistic regression models where the feature vectors are homogeneous (in our model, the feature vectors are heterogeneous). Using the Convex Gaussian Min–Max Theorem (CGMT), they derived a variational characterization of the training loss and showed that taking $\nu, \gamma \to \infty$ indeed drives the loss to zero. In our work, we obtain a related characterization via a leave-one-out analysis leading to a system of self-consistent equations; studying this system in the $\lambda \to 0$ limit, however, poses additional technical challenges.
>
> Weakness 3: "Some experiments, even synthetic ones, could further strengthen the work and provide more evidence for the real-world applicability of the presented results."
>
> Response: We thank the reviewer for this suggestion. We have added a new section presenting synthetic experiments that compare the simplified attention model studied in our paper with more standard attention variants trained under typical optimization dynamics. These experiments reveal that these different models exhibit qualitatively similar learning curves, suggesting that our simplified setup can reflect the behavior of more complex attention architectures at the early phase of the training, for the sparse token classification task.

---

### Official Review · Reviewer_CDaj · 2025-11-01

**Soundness:** 2
**Presentation:** 2
**Contribution:** 3
**Rating:** 4
**Confidence:** 3

**Summary:**

This paper theoretically proves that a single-layer attention model can detect signals exponentially weaker ($\theta = O(\log L)$) than linear classifiers ($\theta = \Omega(\sqrt{L})$) in sparse-token classification tasks where weak signals are randomly scattered across sequences.

In a high-dimensional asymptotic regime ($d, n \to \infty$ with $n/d$ fixed), the authors show that just two gradient steps enable the attention mechanism to align with the hidden signal and provide exact characterizations of the resulting test error and model capacity.

**Strengths:**

The paper tackles a meaningful problem--detecting weak, rare, and sparse signals (e.g., lesions in scans)-- where attention’s adaptive token selection offers clear benefits over fixed pooling.

It establishes an exponential gap in representational power, showing attention detects signals at $\theta = O(\log L)$ vs. linear classifiers needing $\theta = \Omega(\sqrt{L})$, where $L$ is the sequence length. The analysis provides exact asymptotic characterizations of test error and shows that just two gradient steps suffice for meaningful query-signal alignment.

Experiments (Figures 1, 3) align closely with theoretical predictions for test/train errors, confirming the asymptotic analysis in moderately high-dimensional regimes.

**Weaknesses:**

W1.  The 4-stage training protocol (Section 3, lines 270–291) with data splitting $\mathcal{D} = \mathcal{D}_0 \cup \mathcal{D}_1$, only 2 gradient steps on $q$, then full ERM on $w, b$ does not reflect how transformers are actually trained.

W2.  The attention model (Eq. 5) is single-layer, single-head with no key/value matrices, feedforward layers, layer norms, or residual connections.

W3. Theorem 2 proves attention can detect signals at $\theta = O(\log L)$ by showing existence of optimal parameters $(q^* = \tau\xi, w^* = \xi)$, but provides no guarantee that gradient descent actually finds these parameters.


W4. Lemma 4 defines the clipping threshold $I$ using $R^2 = (2/\lambda) \mathbb{E}[\ell(\epsilon; y)]$, but this expectation depends on the solution $w^\star$ being bounded. The proof that $x^\star = x^\star_{\text{clip}}$ holds with probability $1-\delta$ relies on quantities that themselves depend on the solution’s properties, creating potential circularity that needs explicit resolution.


W5. Conjecture 1  provides the main capacity characterization via expressions in Eq. (15), but the derivation involves an unverified “heuristic step” (line 2356) in taking the $\gamma, \nu \to \infty$ limit.


W6. The high-dimensional limit takes $d, n \to \infty$ with $\alpha = n/d$ fixed while keeping $L, \theta, R$ finite (Assumption 1). This is opposite to practical transformer settings where sequence length $L$ often far exceeds embedding dimension $d$.

**Questions:**

Please refer to W1-W6.

---

> ### Author Response · Authors · 2025-11-20
> **Reviewer 2 (CDaj) response**
>
> We thank the reviewer for their comments, which we address below.
>
> W1, W2:
> We acknowledge that these simplifications represent limitations of current theoretical analyses of transformers, shared by many contemporary works. Similar single-layer models have been the focus of several recent studies (e.g., Marion et al., 2024; Tarzanagh et al., 2023; Oymak et al., 2023). The ERM we analyze, while simplified, accommodates a finite number of samples and a broad class of convex losses, thereby complementing prior works that focus on the population setting and the square loss (e.g., Wang et al., 2024; Marion et al., 2024). Developing a theory that fully captures the complexity of modern transformer architectures remains an ambitious and long-term goal that the theory community will reach only through a sequence of progressive advances. We therefore believe our manuscript should be assessed in light of how it advances the theoretical understanding of attention-based mechanisms, rather than by its proximity to current empirical deep learning practice.
>
> Moreover, in light of the reviewers' comments, we have added a new section containing synthetic experiments comparing the attention model studied in our paper with more familiar attention variants and under more complex training dynamics. These experiments demonstrate strong parallels  motivating the study of our simplified setup as a proxy for more complex variants. For instance, in Figure 2 (left), we see little difference between train and test errors under our training protocol with data splitting  $\mathcal{D} = \mathcal{D}_0 \cup \mathcal{D}_1$ versus reusing data $\mathcal{D}_0 = \mathcal{D}_1$ in the optimization of $w,b$.
>
> W3:
>  We note that while Section 2 focuses on statistical guarantees, Section 3 analyzes an explicit training procedure involving two gradient descent steps and shows that it achieves good generalization. In fact, Theorem 3 and Corollary 1 together imply that the weights $q$ and $w$ obtained after two gradient descent steps are colinear with $\xi$ in the limit of infinitely many samples ($\alpha_0 \to \infty$) followed by $L \to \infty$. Hence, gradient descent can indeed recover the parameters referenced by the reviewer.
>
> We expect that analyzing the full gradient descent dynamics should also be feasible for this simple model---both in the population case (along the lines of Marion et al., 2024) and in the finite-sample case (using ideas similar to those in Gerbelot et al., Rigorous Dynamical Mean Field Theory for Stochastic Gradient Descent Methods, 2022). However, such an investigation would require substantial additional technical work and we leave it to future research.
>
> W4:
> The expression for $R$ corresponds to the loss evaluated at zero weights and therefore does not involve the solution $x^\star$, and hence it does not lead to circularity. To elaborate, the proof strategy is to clip the first derivative of the loss outside the interval $(-I, I)$. We then show that, with high probability, all residuals $f_i^\top x^\star_{\mathrm{clip}}$ lie within $(-I, I)$, where the clipped and unclipped losses coincide. Consequently, $x^\star_{\mathrm{clip}}$ satisfies the optimality condition for the original minimization problem and, by uniqueness, coincides with $x^\ast$ under this high-probability event.
>
> W5:
> The heuristic assumption $\gamma, \nu \to \infty$ as $\lambda \to 0$ in the derivation of Conjecture 1 is motivated by the following intuition. The logistic loss is minimized when its scalar argument tends to infinity, in which regime its second derivative vanishes. When the dataset is linearly separable and the regularization parameter $\lambda$ approaches zero, this implies that the norm $\nu$ of the estimator diverges. At the same time, as the second derivatives of the loss become small in this limit, the trace of the inverse Hessian—proportional to $\gamma$—also diverges. These assumptions are further corroborated by numerical experiments: as shown in the new Figure 7, both $\gamma$ and $\nu$ are observed to diverge as $\lambda \to 0$, consistent with the theory. Similar behavior has been rigorously established in simpler settings, for example by Mignacco et al. (2020), where the feature vectors are homogeneous (our feature vectors are heterogeneous). We obtain a related characterization via a leave-one-out analysis leading to a system of self-consistent equations; studying this system in the $\lambda \to 0$ limit, however, poses additional technical challenges.
>
> W6:
> We share the reviewer’s view that extending the analysis of Section 3 to the regime $L \gg 1$ would be a very exciting research direction. Pursuing this extension would require the development of entirely new asymptotic analysis techniques and represents an ambitious goal for future work. We note, however, that the discussion in Section 2 regarding the representational power of the models is already formulated in the large-$L$ limit for arbitrary $d$.

---

### Official Review · Reviewer_sdvV · 2025-11-03

**Soundness:** 3
**Presentation:** 3
**Contribution:** 3
**Rating:** 8
**Confidence:** 4

**Summary:**

This paper introduces a sparse-token classification task, where positive samples demonstrate a weak signal in a random sparse subset of tokens, while negative tokens are i.i.d. Gaussian noise. The authors prove that by learning to recover the sparse subset, a single-layer of attention with a single learnable query token can achieve vanishing test error with the magnitude of signal only logarithmic in sequence length. On the other hand, the authors consider two baselines, fully vectorized linear classifiers and average-pooled linear classifiers, where they demonstrate that the signal magnitude requires to be at least of order $\sqrt{L}$ for sequence length $L$ for vanishing test error.

The authors additionally study the effect of two gradient descent updates on the attention model, which allows the query token to achieve non-trivial alignment with the signal vector. They further characterize the exact limiting train and test errors of the above algorithms, as well as their capacity to overfit.

**Strengths:**

While the ability of attention to recover a sparse subset of important tokens has been the subject of many recent theoretical works, this paper presents a strong case by studying a fundamental task in classification. A particularly interesting aspect of the paper is on the technical side to accommodate attention in high-dimensional asymptotic analyses of classification error.

**Weaknesses:**

* The problem introduced does not really need to use the full power of attention. This setting can be seen as having a fixed query vector $\mathbb{1}\_d$ of all ones and a trainable query projection $\mathrm{diag}(q)$. Recurrent/state-space models could also efficiently achieve vanishing test error by going through the input sequence and memorizing tokens that have a high correlation with $q$. It would be interesting, perhaps for future work, to extend this setting to the case where the number of queries in the attention mechanism would also grow, which would enable attention to outperform even more baselines.

* I think some additional details in the main text could provide valuable intuition to the readers. For example, how do the optimal weights for attention/vectorized/pooled models look like in Section 2? I also thing some details might be missing on the loss, e.g. Corollary 1 could not hold if the loss is constant, since $q$ would be left at $0$.

**Questions:**

* Isn't average pooling a specific case of the fully vectorized classifier? Would that imply that in Section 2, the optimal error of the pooled classifier is lower bounded by that of the fully vectorized?

* Minor typos:
    * Line 103: negative signals -> negative samples
    * Line 122: I believe the authors meant $CR/\sqrt{L}$, but it looks like $R$ is the exponent of $C$ in that expression.
    * Line 458: the capacity if linear classifiers -> of linear classifiers

---

> ### Author Response · Authors · 2025-11-20
> **Reviewer 1 (sdvV) response**
>
> We thank the reviewer for their appreciation of our manuscript and insightful questions, which we answer below.
>
> Weakness 1: "The problem introduced does not really need to use the full power of attention. This setting can be seen as having a fixed query vector $1_d$ of all ones and a trainable query projection ${\rm diag}(q)$. Recurrent/state-space models could also efficiently achieve vanishing test error by going through the input sequence and memorizing tokens that have a high correlation with $q$. It would be interesting, perhaps for future work, to extend this setting to the case where the number of queries in the attention mechanism would also grow, which would enable attention to outperform even more baselines."
>
> Response: The reviewer is correct that the model we study can be viewed as a simplified attention mechanism with a diagonal query projection. We demonstrate that even this simple model exhibits a clear advantage over non-adaptive linear classifiers in terms of the SNR required to achieve perfect classification in the studied task. It would be very interesting to investigate whether other architectures, such as RNNs, share similar SNR requirements for perfect classification.
>
> We also share the reviewer’s perspective that exploring scenarios requiring more complex attention mechanisms is an exciting direction for future work. In particular, we anticipate that settings involving multiple signals, $\xi_1, \ldots, \xi_m$, may need architectures with multiple queries (each column specializing in one signal) or multi-headed attention (with each head potentially specializing in a different signal). We leave the investigation of these promising extensions to future work.
>
> Weakness 2a: "I think some additional details in the main text could provide valuable intuition to the readers. For example, how do the optimal weights for attention/vectorized/pooled models look like in Section 2?"
>
> Response: We thank the reviewer for the suggestion. From equation (19), it follows that the optimal weight of the pooled classifier coincides with the signal $\xi$. For the vectorized classifier, the discussion around lines 770–800 establishes that the optimal weight corresponds to a vector formed by concatenating the signal vector $L$ times and rescaling by $1/\sqrt{L}$. Likewise, perfect generalization is achieved by the attention model when both weights $q$ and $w$ are colinear with the signal $\xi$, as discussed above equation (36). In the revised manuscript, we have added a discussion of this point under the statements of Proposition 1 and Theorem 1.
>
> Weakness 2b: "I also think some details might be missing on the loss, e.g. Corollary 1 could not hold if the loss is constant, since $q$ would be left at 0."
>
> Response: We thank the reviewer for pointing out this fact which we had implicitly assumed. A necessary and sufficient condition for Corollary 1 to hold is that $\xi^\top q^{(2)}$ is non-zero. This condition is in particular untrue for the constant loss, for which Corollary 1 therefore does not hold. We have explicitly added this condition in the statement of Corollary 1.
>
> Q1: Isn't average pooling a specific case of the fully vectorized classifier? Would that imply that in Section 2, the optimal error of the pooled classifier is lower bounded by that of the fully vectorized?
>
> Response: The reviewer is correct that the pooled classifier can be realized as a special case of the vectorized classifier with tied weights. Consequently, equation (8) in Proposition 1 provides an upper bound on the optimal error of the vectorized classifier analyzed in Theorem 1. We have added an explicit discussion of this point under the statement of Theorem 1 in the main text. In fact, this observation is already used implicitly in the proof of Theorem 1 (lines 836–838): the special case $a \propto 1_L$, which achieves perfect classification, corresponds to tying the weights of the vectorized classifier, thereby reducing it to the pooled case.
>
> Q2: Minor typos
>
> Response: We thank the reviewer for pointing out the typos in our submission which are now fixed.

---

### Meta-Review · Area_Chair_kwze · 2026-01-07

**Summary:**

This work theoretically analyzes a single attention layer for classification tasks with sparse signals and reveals both a dependency on the sequence length and signal recovery in a high-dimensional asymptotic regime. All results are rigorous and concrete.
While the reviewers commented on the validity of some simplified settings for theoretical tractability, the authors clarified the extensibility and limitations of their analysis in the rebuttal phase and the revised manuscript. I think that all reviewers would be sufficiently convinced of the significance of the current work as theoretical research. Thus, I recommend this work for acceptance.

**Reviewer Concerns:**

Reviewer CDaj is the only reviewer on the rejection side, but all of their concerns seem to be addressed by the rebuttal. In particular,

>W1. data splitting ... does not reflect how transformers are actually trained.

>W2. The attention model (Eq. 5) is single-layer, single-head with no key/value matrices, feedforward layers, layer norms, or residual connections.

seems resolved by numerical experiments in the revised manuscript.

**Reviewer Scores:**

Because of the resolved concerns as mentioned above, I expect that

*Reviewer CDaj would increase their score to the acceptance side.

---

### Decision · Program_Chairs · 2026-01-26

Accept (Poster)